# TROLL 4.0: representing water and carbon fluxes, leaf phenology, and intraspecific trait variation in a mixed-species individual-based forest dynamics model – Part 2: Model evaluation for two Amazonian sites

Sylvain Schmitt[1,2,3], Fabian J. Fischer[4], James G. C. Ball[5], Nicolas Barbier[3], Marion Boisseaux[6], Damien Bonal[7], Benoit Burban[8], Xiuzhi Chen[9], Géraldine Derroire[1,2,10], Jeremy W. Lichstein[11], Daniela Nemetschek[4], Natalia Restrepo-Coupe[12], Scott Saleska[12], Giacomo Sellan[10], Philippe Verley[3], Grégoire Vincent[3], Camille Ziegler[6], Jérôme Chave[13], Isabelle Maréchaux[3]

[1]CIRAD, UPR Forêts et Sociétés, F-34398, Montpellier, France
[2]Forêts et Sociétés, Univ Montpellier, CIRAD, Montpellier, France
[3]AMAP, Univ Montpellier, INRAE, IRD, CIRAD, CNRS, F-34000 Montpellier, France
[4]School of Biological Sciences, University of Bristol, 24 Tyndall Avenue, Bristol, BS8 1TQ, UK
[5]Department of Plant Sciences, University of Cambridge, Downing Street, Cambridge, CB2 3EA, UK
[6]Univ. Bordeaux, INRAE, BIOGECO, 33612 Pessac, France
[7]Université de Lorraine, AgroParisTech, INRAE, UMR Silva, 54000 Nancy, France
[8]INRAE, UMR EcoFoG (Agroparistech, Cirad, CNRS, Université des Antilles, Université de la Guyane), Campus Agronomique, 97310 Kourou, French Guiana
[9]School of Atmospheric Sciences, Sun Yat-sen University, Guangzhou 510275, China
[10]Cirad, UMR EcoFoG (Agroparistech, CNRS, INRAE, Université des Antilles, Université de la Guyane), Campus Agronomique, 97310 Kourou, French Guiana
[11]Department of Biology, University of Florida, Gainesville, Florida 32611, USA
[12]University of Arizona, Ecology & Evolutionary Biology, Tucson, Arizona, United States of America
[13]Centre de Recherche Biodiversité et Environnement, UMR5300, CNRS, Université Paul Sabatier, IRD, INPT, Toulouse Cedex 9, France

*Correspondence to*: Sylvain Schmitt (sylvain.schmitt@cirad.fr)

**Summary.** We evaluate the capability of TROLL 4.0, a simulator of forest dynamics, to represent tropical forest structure, diversity, dynamics and functioning in two Amazonian forests. Evaluation data include forest inventories, carbon and water fluxes between the forest and the atmosphere, and leaf area and canopy height from remote-sensing products. The model realistically predicts the structure and composition, and the seasonality of carbon and water fluxes at both sites.

**Abstract.** TROLL 4.0 is an individual-based forest dynamics model that jointly simulates the structure, diversity and functioning of tropical forests, including their water balance, carbon fluxes and leaf phenology, while accounting for intraspecific trait variation for a large number of species. In a companion paper, we describe how the model represents the physiological and demographic processes that control the tree life cycle in a one-metre-resolution spatially-explicit scene and uses plant functional traits measurable in the field to parameterize such processes across species and individuals (Maréchaux et al., submitted companion paper). Here we evaluate the performance of TROLL 4.0 for two Amazonian sites with contrasting soil and climate properties. We assessed the model's ability to represent forest structure, composition, and dynamics using lidar-derived spatial distribution of top canopy height and forest inventories combined with information on plant functional traits. We also evaluated the model's ability to represent carbon and water fluxes, as well as leaf area variation, at daily and fortnightly resolution over a decade, using detailed information from on-site eddy covariance towers, satellite data and ground-based or air-borne lidar data. We finally compared the responses of carbon and water fluxes to environmental drivers between simulated and observed data. Overall, TROLL 4.0 provided a realistic representation of forests at both sites. The simulated canopy height distribution showed a high correlation coefficient (CC) with observed aerial and satellite data (CC>0.92), while the species and functional composition were well represented (CC>0.75). TROLL 4.0 also realistically simulated the seasonal variability of carbon and water fluxes (CC>0.46) and their responses to environmental drivers, while capturing temporal variations in leaf area (CC>0.76) and its partitioning in leaf age cohorts. However, TROLL 4.0 overestimated annual gross primary productivity at both sites (mean RMSEP=0.94±0.67 kgC m$^{-2}$ yr$^{-1}$) and evapotranspiration at one site (mean RMSEP=0.75±0.63 mm day$^{-1}$), likely due to an underestimation of the soil water depletion and stomatal control during the dry season. This evaluation highlights the potential of TROLL 4.0 to represent ecosystem fluxes and the structure, diversity, and dynamics of plant communities at a fine resolution, paving the way for model predictions of the effects of climate change, fragmentation and forest management on forest structure and dynamics.

# 1 Introduction

Tropical forests cover just 7% of the Earth's land surface, yet they play a disproportionately large role in the biosphere, store around 25% of terrestrial carbon and contribute to more than a third of global terrestrial productivity (Bonan 2008). Regionally, tropical forests recycle around a third of precipitation through evapotranspiration, contributing to the generation and maintenance of a humid climate (Harper et al., 2013), effects that extend well beyond the tropics (Lawrence & Vandecar 2015). However, tropical forests remain a major source of uncertainty in simulations of global biogeochemical cycles (Fisher et al., 2014; Koch et al., 2020).

As an illustration, for light-limited tropical forests, dynamic global vegetation models (DGVMs, Prentice et al., 2007) typically simulate a decrease in productivity with a seasonal decline in precipitation (Restrepo-Coupe et al., 2017, Chen et al., 2020), while observations from eddy covariance data point to an increase in gross primary productivity during the dry season (Guan et al., 2015; Aguilos et al., 2018). Similarly, simulated forest responses to experimental and natural droughts have highlighted large model-data discrepancies and variation across models (Powell et al., 2013; Joetzjer et al., 2014; Yao et al., 2023; Paschalis et al., 2022). Improving the representation of tropical forest functioning in models is needed to enhance our understanding and ability to predict biogeochemical cycles.

One challenge is to better integrate the structure, diversity and functioning of forests into vegetation models (Purves and Pacala, 2008; McMahon et al., 2011; Evans, 2012; Mokany et al., 2016). In spite of progress (Fisher et al., 2018), most models still adopt a coarse grained representation of vegetation, which makes it difficult to use field data to parameterize and evaluate the models. Also, several processes driving the variation of tropical forest productivity and water fluxes remain incompletely represented in vegetation models. These include water uptake by the root system and seasonal variation of leaf quantity and quality at the ecosystem-level, which are driven by leaf phenology and allocation processes at the individual-level (Chen et al., 2020; Wu et al., 2021; Restrepo-Coupe et al., 2017, Cusak et al., 2024).

In a companion paper, we described the individual-based forest dynamics model TROLL 4.0 (Maréchaux et al., submitted companion paper). This model jointly simulates tropical forest structure, diversity and functioning, including forest water balance, carbon fluxes and leaf phenology, and accounts for intraspecific trait variation for a large number of species. TROLL 4.0 represents the processes underlying ecosystem fluxes, such as leaf gas exchanges and their responses to environmental variation, and is thus similar to DVGMs in that respect, with its outputs comparable with data from eddy covariance towers. However, unlike DGVMs that are designed for global applications and typically represent plant diversity with a few functional types, TROLL 4.0 represents diversity at the species level (e.g., 10s to 100s of tropical tree species). TROLL 4.0 is spatially-explicit and represents plant community structure and diversity at a spatial resolution of one metre, which is consistent with that used by field ecologists. Physiological and demographic processes are integrated using a parameterisation based on plant traits measurable in the field, relying on recent knowledge in plant physiology and functional ecology. The individual-based,

species-specific and spatially explicit representation of forest structure and composition enables TROLL 4.0 outputs to be directly compared with spatially explicit forest inventories, trait distributions or fine-scale remote sensing products.

In this paper, we evaluate TROLL 4.0 for two Amazonian sites with contrasting soil and climate properties. We parameterized the model using functional trait and soil data at both sites. We first calibrated three major forest structure parameters using inventory data, and then the three parameters of the phenological module that control leaf shedding as a function of soil water availability using litterfall data. We then ran simulations and evaluated the model's representation of forest structure, composition, and dynamics against independent data, including lidar-derived canopy height distribution, understory inventories and functional trait distribution. We also assessed the model ability to represent carbon and water fluxes at daily resolution, as well as leaf area variation at fortnightly resolution, against eddy covariance, satellite and terrestrial or drone lidar data. We finally compared the response of simulated and observed fluxes to incoming radiation, vapour pressure deficit, temperature, and wind speed. Finally, we discuss the potential model-data discrepancies and identify priorities for future developments.

## 2 Methods

TROLL represents individual trees explicitly in an aboveground voxelized space ($1 \, m^3$), in which light diffusion is modelled, and in a belowground space, which consists of several layers with user-defined thickness and horizontal resolution (here 25 $m^2$). Belowground water flow is simulated using a bucket model. We assign a species label to each simulated tree and provide as input species-specific mean plant trait values and intraspecific trait variances and covariances. New trees appear in the community through the process of tree recruitment, which is only possible in empty cells and with favourable light and water availability. Trees of a given species are recruited if there is at least one seed of that species in the local seed bank. Individual trait values of each recruited tree are randomly drawn from the intraspecific trait distribution. These traits parameterize the physiological and demographic processes that govern the life cycle of trees, from recruitment to growth, seed dispersal, and finally death. Carbon assimilation by trees is computed using the photosynthesis model of Farquhar, von Caemmerer and Berry (1980), coupled to the stomatal conductance model of Medlyn et al. (2011), as a function of leaf micro-environmental conditions, tree access to water, and leaf photosynthetic capacity and leaf respiration rate. Sugars produced during photosynthesis are used for tree respiration and allocation to plant tissues, including foliar production, carbon storage and woody growth.

We conducted model calibration and evaluation at two lowland Amazon forest sites: the Paracou research station in French Guiana (5°28'N, 52°92'W), hereafter Paracou (Gourlet-Fleury et al., 2004; Bonal et al., 2008), and the Tapajos National Forest in Brazil in the K67 site also named BR-Sa1 (2°86'S, 54°96'W), hereafter Tapajos (Silver et al., 2000; Saleska et al., 2003). Both sites are covered by a high biomass and species rich lowland moist tropical forest, and they present contrasting soil characteristics and climate (Table 1), with a longer dry season in Tapajos than in Paracou resulting in 2,075 mm per year

against 3,041 in Paracou. They thus differ in water regimes and resulting plant water stress and phenology. In addition, the two sites have been intensively monitored for several decades, mainly through repeated forest inventories and eddy flux tower measurements, fulfilling the requirement for in-depth model evaluation as previously used for such applications (Longo et al., 2019b). Additionally, we assumed forest dynamics to be at equilibrium, as both sites are characterised by old-growth forests.

To provide a conservative assessment of the model's performance and its transferability to multiple sites, we restricted the number of site-specific calibrated parameters to the ones that are currently poorly informed by available data, or to which the model is known to be sensitive based on sensitivity analyses performed on previous versions of the model (Maréchaux & Chave 2017; Fischer et al. 2019). At each site, we calibrated six parameters. These include three parameters related to forest structure: the reference background mortality rate $m$, and the intercept $a_{CR}$ and slope $b_{CR}$ of the crown radius scaling relationship (Table A1; Maréchaux and Chave, 2017; Fischer et al., 2019). $m$ can be site-specific as it is used to simulate tree mortality events that are triggered by processes not explicitly represented in the model, such as site-specific disturbance regimes (e.g. Rau et al. 2022). Novel developments in TROLL 4.0 were based on known or measurable ecological parameters and physical constants but the three parameters of the new leaf phenology module $a_{T,o}$, $b_{T,o}$ and $\delta_o$ (Table A1) are more empirical and not ecologically measurable. In TROLL 4.0, the shedding of old leaves is accelerated as soil water availability decreases (Maréchaux et al., submitted companion paper). When the leaf predawn water potential ($\psi_{pd}$, MPa) falls below a threshold $\psi_{T,o}$ (MPa), the residence time of old leaves is decreased using a multiplicative factor $f_0 < 1$. The parameter $\psi_{T,o}$ varies with the tree leaf drought tolerance and its size as follows:

$$\psi_{T,o} = min(a_{T,o} \times \pi_{tlp}, -0.01 \times h - b_{T,o})$$

where $\pi_{tlp}$ is the leaf water potential at turgor loss point in MPa and $h$ is the tree height in m. $f_0$ is decremented (resp. incremented) by $\delta_o$ when $\psi_{pd} < \psi_{T,o}$ (resp. $\psi_{pd} > \psi_{T,o}$). The first term accounts for a decline in leaf drought tolerance with age, i.e. a reduced ability of old leaves to maintain turgor when the soil dries, where $a_{T,o}$ controls the ratio of the turgor loss point of old to mature leaves. The second term accounts for the height dependence of this susceptibility to decreasing water availability: it makes large trees susceptible to a (small) decrease in soil water availability $b_{T,o}$, while preventing them from constantly shedding their old leaves at a fast rate. Finally, $\delta_o$ controls the rate of leaf shedding in old leaves as they begin to lose turgor, but in the absence of water depletion. Overall, the parameters $a_{T,o}$, $b_{T,o}$ and $\delta_o$ control the intensity and timing of the peak of litterfall under drying soil conditions. This scheme is consistent with field observations (Maréchaux et al. submitted companion paper), uncertainties remain on the values of $a_{T,o}$, $b_{T,o}$ and $\delta_o$ however, and they need to be calibrated. After calibration, we compared model outputs with site-specific data for evaluation at each site.

**Table 1: Site overview with climate, vegetation and soil properties. Soil properties are those used as input of the pedotransfer functions implemented in TROLL 4.0.**

| Variables | Units | Paracou | Tapajos | References |
| --- | --- | --- | --- | --- |

| | | | | |
|---|---|---|---|---|
| **Climate** | | | | |
| Annual rainfall | mm | 3,041 | 2,075 | P: Aguilos et al., 2018; T: Silver et al., 2000 |
| Average air temperature | °C | 25.7 | 26.1 | |
| **Vegetation** | | | | |
| Aboveground biomass ($DBH \geq 10$) | Mg ha$^{-1}$ | 419 | 287 | P: Rutishauser et al., 2010; T: Rice et al., 2004 |
| Number of stems ($DBH \geq 10$) | ha$^{-1}$ | 612 | 470 | P: Derroire et al., 2023; T: Rice et al., 2004 |
| Basal area ($DBH \geq 10$) | m$^2$ ha$^{-1}$ | 31 | 24 | P: Derroire et al., 2023; T: Goncalves et al., 2018 |
| **Soil** | | | | |
| Type | - | Sandy clay loam | Clay | - |
| Depth | m | 2.50 | 16.10 | P: Hiltner et al., 2021; T: Nepstad et al., 2002 |
| Layer thickness (top to bottom) | m | 0.10 / 0.23 / 0.40 / 0.80 / 0.97 | 0.10 / 0.40 / 1.00 / 2.50 / 12.10 | - |
| Sand | % | 65.25 | 37.27 | P: Van Langenhove et al., 2021; T: Silver et al., 2000 |
| Clay | % | 21.50 | 60.09 | |
| Silt | % | 13.25 | 2.64 | |
| Soil Organic Content | % | 2.37 | 2.54 | P: Van Langenhove et al., 2021; T:Quesada et al., 2010 |
| Dry Bulk Density | g cm$^{-3}$ | 1.040 | 1.125 | P: Van Langenhove et al., 2021; T: Silver et al., 2000 |
| Cation Exchange Capacity | mEq 100g$^{-1}$ | 2.98 | 2.97 | P: Sabatier et al., 1997; T:Quesada et al., 2010 |

| | | | |
|---|---|---|---|
| pH | 4.34 | 3.84 | P: Sabatier et al., 1997; T:Quesada et al., 2010 |

## 2.1 Simulation inputs and climatic drivers

TROLL 4.0 uses 35 global parameters defined by the user and provided as inputs. These parameters relate to atmospheric
constants, light transmission, leaf carbon acquisition, leaf shedding, tree carbon allocation, tree shape, reproduction, and death,
and intraspecific trait variability (Table S1). Except for the three parameters of forest structure mentioned above and the three
parameters of the leaf shedding module that have been calibrated at each site, all values are assumed site independent.

TROLL 4.0 requires trait parameters for each species: values need to be provided as input for six functional traits and three
scaling parameters. The scaling parameters are species maximum diameter at breast height ($dbh_{thres}$, cm), and parameters
defining the relationship between height and diameter at breast height (dbh), which are the asymptotic height ($h_{lim}$, m) and the
parameter $a_h$ (see Maréchaux et al. submitted companion paper, Eqs (16) and (62)). We used forest inventories from Paracou
(Derroire et al., 2023) and Tapajos (Goncalves et al., 2018) to create a species list for each site, and computed $dbh_{thres}$ as the
95[th] quantile of species diameter at breast height for species including more than 10 individuals. We used the TALLO global
database of height and diameter measurements (Jucker et al., 2022) to infer species-specific values of $h_{lim}$ and $a_h$ for the 496
species of the database that are present in Amazonia (latitude between 10°N and 18°S and longitude between 39°W and 78°W;
n = 24,609 trees with a mean of 49.62 ±730 trees per species). Parameters $a_h$ and $h_{lim}$ were inferred using Bayesian inference
as follows:

$$log(h) \sim N[log(h_{lim} \times \frac{dbh}{a_h + dbh}), \sigma^2] \mid h_{lim} \sim N(h_{lim,0}, \sigma^2_h), a_h \sim N(a_{h,0}, \sigma^2_a)$$

with the logarithm of height ($h$, in m) following a normal distribution centred on the log of a Michaelis-Menten model with
asymptotic height $h_{lim}$, height-dbh scaling parameter $a_h$, and variance $\sigma^2$. We used a Michaelis-Menten model form for tree
height $h$, which grows with diameter $dbh$ towards a plateau value $h_{lim}$ at a rate $a_h$ (Molto *et al.,* 2014). The two species-
specific parameters $h_{lim}$ and $a_h$ are random parameters following a normal distribution centred respectively on $h_{lim,0}$ and $a_{h,0}$
with variances $\sigma^2_h$ and $\sigma^2_a$.

The functional traits used in the parameterization include leaf area (LA, in cm$^2$), leaf mass per area (LMA, g m$^{-2}$), leaf nitrogen
content per dry mass (N, g g$^{-1}$), leaf phosphorus content per dry mass (P, g g$^{-1}$), leaf water potential at turgor loss point ($\pi_{tlp}$,
MPa), and wood specific gravity (wsg, g cm$^{-3}$). We used several datasets to retrieve species-specific mean values for these
traits (Vleminckx et al. 2021, Boisseaux et al., submitted; Kattge, Bönisch, and al., 2020; Maréchaux et al., 2015; Maréchaux
et al., 2019; Ziegler et al., 2019). Finally, we used predictive mean matching (Van Buuren and Groothuis-Oudshoorn, 2011)
to impute missing trait values for $a_h$, $h_{lim}$, $dbh_{thres}$, and $\pi_{tlp}$. Overall, this procedure leads to a parameterization of 114 species
for Paracou and 113 species for Tapajos, with imputed values for 4 to 34 species for $a_h$, $h_{lim}$, $dbh_{thres}$, and $\pi_{tlp}$ (Fig. A1).

TROLL 4.0 requires nine soil parameters to describe the texture, depth and chemistry. These were gathered from the literature,
assuming a single soil type and depth per site for simplicity and setting the number of soil layers to five (Table 1). Testing the
influence of horizontal and vertical soil heterogeneity on model outputs is left for future work.

TROLL 4.0 simulations are forced with six climatic drivers. Two of them are daily: cumulative rainfall (mm), and average
nighttime temperature (°C). The remaining four drivers are provided every half hour during the daytime (defined below):
incoming shortwave radiation (SW, W m$^{-2}$), temperature (T, °C), vapour pressure deficit (VPD, kPa), and wind speed (WS, m
s$^{-1}$). Historical time series for these climatic variables have been retrieved from the FLUXNET 2015 dataset (Pastorello et al.,
2020), which provides standardised data from eddy flux towers located at each site (2004-2014 for Paracou, and 2002-2011
for Tapajos). However, at Tapajos, rainfall data from FLUXNET 2015 is not reliable due to issues with rain gauges (Restrepo-
Coupe et al., 2017). Instead, we used rainfall data from the ERA5-Land reanalysis dataset (Muñoz-Sabater et al., 2021)
available at hourly resolution between 2002 and 2011. For other climatic variables, data from ERA5-Land showed high
correlation with FLUXNET 2015 data and ERA5-Land showed a better agreement with on-site precipitation data from
FLUXNET 2015 at Paracou when compared to other products, like CHIRPS (Funk et al. 2015; Fig. A2). We used spline
interpolation to derive half-hourly time series from the hourly FLUXNET 2015 data in Tapajos. The half-hourly net radiation
time series was used to define daytime hours (i.e. with $S_{net} > 0$) which were set from 6 a.m. to 6 p.m. in Paracou, and from 7
a.m. to 7 p.m. in Tapajos. The dry season was defined as a period with fortnightly rainfall below 50 mm on average across
years, consistent with the 100 mm per month used by Bonal et al. (2008). This leads to a 4-month dry season in Paracou
(August 1st to December 1st), and a 4.5-month dry season in Tapajos (June 15 to November 1st). Dry seasons were defined
for illustration purposes only and have no effect on the model behaviour, which is driven by the meteorological inputs described
above.
**2.2 Calibration and simulation set-up**
As opposed to fine-tuning the model, we opted for minimum calibration to assess the model's behaviour with a minimum of
information per site, and tuning to assess its transferability, at least across Amazonian sites. We calibrated the three forest
structure parameters ($m$, $a_{CR}$ and $b_{CR}$) for each site. $a_{CR}$ and $b_{CR}$ are not independent, and we used the TALLO global database
of crown radius (CR) and diameter (dbh) measurements (Jucker et al., 2022) to infer their relationship. To do so, we restricted
the TALLO database to observations located within 10 km around sites from which we generated a thousand pairs of ($a_{CR}, b_{CR}$)
values. Each pair of values was determined by randomly drawing 10 individuals per 10-cm diameter class to generate a size-
balanced dataset to which the following model was fitted: $log(CR) \sim N[a_{CR} + b_{CR} \times log(dbh), \sigma^2]$. This resulted in the
following linear relationship between the two parameters: $b_{CR} = -0.39 + 0.59 \times a_{CR} + \epsilon_{b_{CR}}$, with $\epsilon_{b_{CR}}$ the error around
the relation. This relationship constrained the exploration of the three-dimensional parameter space, so we only had to calibrate
$a_{CR}$, $\epsilon_{b_{CR}}$, and $m$. Based on preliminary exploratory analyses with the previous version of TROLL, we defined the range of

calibration for each parameter and site as follows: $a_{CR}$ varied from 1.60 to 2.00 in Paracou and from 2.3 to 2.7 in Tapajos with a step of 0.05, $\epsilon_{b_{CR}}$ from -0.30 to 0.10 in both sites with a step of 0.05, and $m$ from 0.030 to 0.050 in both sites with a step of 0.0025. This resulted in $9\ a_{CR} \times 5\ \epsilon_{b_{CR}} \times 9\ m \times 2\ site = 810$ triplets of parameter values.

For each set of three parameter values, we performed a 600-year simulation from bare ground over a 4-ha area. Simulations were run with an external seed rain uniformly distributed across species, so that the simulated community structure is an emergent property resulting from the community assembly mechanisms embedded in the model. As succession unfolds and the number of mature trees increases in the simulation, internal seed production increases according to the assumed relationships between individual size and fecundity. An alternative to uniform seed rain across species would be to prescribe non-uniform seed rain based on species' regional abundances. This approach would tend to make the simulated species abundances more closely resemble the observed regional abundances. In contrast, uniform seed rain as simulated here, biases the simulated abundances towards evenness across species, and differences in simulated abundances reflect differences in demographic performance controlled by the model trait-based parameterization rather than prescribed differences in the seed rain. Each simulation was forced each year by randomly drawing a year among the ten years of climatic data. In doing so, we avoided applying a periodic climatic forcing or any potential trend linked to global warming.

To evaluate the forest structure simulated with each triplet of parameter values, we compared simulated to observed total aboveground biomass (AGB$^{tot}$, Mg ha$^{-1}$), total number of stems (N$^{tot}$, ha$^{-1}$), and number of stems per 5-cm diameter class (N$^i$, ha$^{-1}$ for *dbh* class *i*) at the end of the 600-year regeneration. The Paracou reference dataset was a 2015 inventory of trees with dbh >10 cm in six 6-ha plots (Derroire et al., 2023). The Tapajos reference dataset was a 1999 inventory of trees with dbh > 10 cm in 19.75 ha along four 1-km transects (Rice et al., 2004). At both sites, we calculated the relative root mean squared error defined as:

$$RRMSEP = \frac{AGB^{tot}_o - AGB^{tot}_s}{AGB^{tot}_o} + \frac{N^{tot}_o - N^{tot}_s}{N^{tot}_o} + \frac{\sqrt{\frac{1}{n} \times \sum_{i=1}^{n} (N^i_o - N^i_s)^2}}{|N^i_o|}$$

where AGB$^{tot}_o$, N$^{tot}_o$ and N$^i_o$ are observed values, and AGB$^{tot}_s$, N$^{tot}_s$ and N$^i_s$ are the simulated values. $n$ is the number of dbh classes and $|N^i_o|$ is the mean number of stems among dbh classes. We extracted the simulation with the lowest *RRMSEP* at each site and used the corresponding values for $m$, $a_{CR}$ and $b_{CR}$ in all subsequent simulations.

After 600 simulated years of forest dynamics the system reached a mature forest stage with stable forest structure (Fig. A3), composition, and functioning at both sites. This is referred to as the 'spin-up phase'. We then used this mature forest stage to calibrate the three parameters of the phenological module. We performed an exhaustive search in the parameter space for combinations of $a_{T,o}$ in [0.01, 0.025, 0.05, 0.075, 0.1, 0.2, 0.3, 0.4, 0.5], $b_{T,o}$ in [0.01, 0.015, 0.02, 0.05, 0.04, 0.06, 0.08, 0.10], and $\delta_o$ in [0.1, 0.2, 0.3, 0.4, 0.5] resulting in $9\ a_{T,o} \times 8\ b_{T,o} \times 5\ \delta_o \times 2\ sites = 720$ simulations. For each triplet, we ran a

20-year simulation with historical weather repeating the 10 years of data twice with the mature forest as an initial condition.
Only the last 10 years were used for the calibration to allow the leaf dynamics to adjust to new parameter values.

To evaluate each simulation, we used leaf litter data from litter traps at both sites (unpublished data at Paracou, Rice et al.,
2008 at Tapajos). Litter traps were typically collected fortnightly (although time intervals between consecutive litter trap
collections were sometimes higher and up to 80 days in Paracou) between 2004 and 2023 in Paracou, and between 2000 and
2005 in Tapajos. The litter collected from the traps was oven-dried until the mass stabilised, partitioned between leaves, fruits
and woody debris, and then the fractions were weighed. We computed observed leaf litterfall flux in Mg ha$^{-1}$ year$^{-1}$ as the mean
across traps converted from trap surface to hectare and time interval in days to year. We also recorded the time interval between
consecutive trap collections to account for the smoothing effect of the longer time intervals in simulated data. Simulated leaf
litterfall fluxes over the last 10 years of simulation for each triplet of parameter values were compared to the observed fluxes
using the same observation dates and corresponding time intervals.

To compare simulations against observations, we defined two yearly indices that quantify the timing and intensity of the
litterfall peak. The two indices are (i) the day of the litterfall peak as the Julian day of the maximum annual litterfall flux value
(*day*), and (ii) the ratio between the maximum value (computed as the average of litterfall flux over the two consecutive time
intervals before and after the peak day) divided by the basal flux (computed as the yearly average between January and April)
(*ratio*). Both indices are key features of litterfall patterns in tropical rainforests (Chave et al., 2010; Yang et al., 2021). For
each simulation we calculated the root mean squared error defined as:
$$RMSEP = \sqrt{\frac{\sum_{y=y_0}^{y=y_{max}} (ratio_{y,o} - ratio_{y,s})^2}{N_{year}} + \frac{\sum_{y=y_0}^{y=y_{max}} (day_{y,o} - day_{y,s})^2}{N_{year}}}$$

where $day_{y,o}$ and $ratio_{y,o}$ are observed z-scores (i.e., standard deviations from the mean) for year y, and $day_{y,s}$ and $ratio_{y,s}$ are
simulated z-scores for year y. Thus a unit *RMSEP* corresponds to a ratio error of one standard deviation, *i.e.* 7.6 folds, or to a
day error of one standard deviation, *i.e.* 45.5 days. The best-fit parameters were those corresponding to the lowest *RMSEP* at
each site.

To assess the model sensitivity to the chosen parameters, we used the calibration parameter spaces and measured response
variable sensitivity to each parameter with partial correlation coefficients (PCC). Moreover, we used a sequential calibration
scheme to reduce computation load based on the hypothesis that the second calibration of litterfall parameters does not interfere
with the first of forest structure parameters. To assess this assumption, we explored the sensitivity of forest structure variables
to forest litterfall parameters.

Finally, to quantify the envelopes of stochastic simulation outputs, we ran ten replicates of 600-year simulations starting from
bare ground with the six calibrated parameter values.

**2.3 Evaluation of forest structure, composition and dynamics**

To assess the model's ability to simulate forest structure and dynamics, as well as species and functional composition, we used airborne lidar scanning (ALS) and satellite data, as well as forest inventories combined with functional traits. Independently from the calibration, we evaluated the diameter distribution of the forest understory at Paracou using an independent 9-ha inventory of trees with dbh between 1 and 10 cm from 2020-2023 (unpublished data). We evaluated the structure of the simulated forest at the end of the 600-year replicates against observed basal area (BA, $m^2$ $ha^{-1}$) and logarithm of number of stems ($ha^{-1}$) per 1-cm diameter class below 10 cm. We evaluated tree height distributions using ALS data from 2015 at Paracou (unpublished data) and from 2012 at Tapajos (dos-Santos et al., 2019), which were processed into canopy height models with a standardised pipeline (Fischer et al., 2024). From both simulated and ALS-derived canopy height models, we derived the distribution of top canopy height, expressed in proportion of 1-$m^2$ pixels per 1-m height class. We evaluated the species composition after the 600-year replicates against the observed rank-abundance curve of the 114 most abundant species at both sites, and the functional composition against the observed density distribution of each trait for each site and each plot. Due to a lower taxonomic resolution of botanical identification at the Tapajos site, we used genus level functional trait data at Tapajos and species level functional trait data at Paracou. Finally, we evaluated forest dynamics by retrieving the simulated individual-tree growth rates (cm $yr^{-1}$) and death rates (% $yr^{-1}$) over 10 years per 5-cm diameter class and comparing them to the ones estimated from field inventories of six 6.25-ha plots in Paracou from 2003 to 2013.

**2.4 Evaluation of total leaf area dynamics**

We assessed the model's ability to represent the dynamics of total leaf area and its partitioning into three leaf age cohorts (Maréchaux et al., submitted companion paper). For evaluation, we gathered leaf area index (LAI) datasets as follows: LAI from MODIS satellites at both sites, LAI from terrestrial lidar at Tapajos (Smith et al., 2019), and LAI from UAV-borne lidar at Paracou (unpublished data; Vincent et al., 2017). The MODIS LAI product was at 8 day and 500 m resolution, and pre-processed in PLUMBER2 (Ukkoloa et al., 2020). At Tapajos, plant area index (PAI) was derived from terrestrial lidar scanning (TLS) performed every 1-2 months in 2010, 2012, 2015 and 2017 along four 1-km long transects representing 0.4 ha with a spatial resolution of about 3 m to characterise canopy porosity (Smith et al., 2019). PAI was derived from lidar hits following Stark et al. (2012) and based on the MacArthur–Horn transformation (MacArthur & Horn, 1969). This PAI was then converted to LAI using an annual mean LAI of 5.7 (Stark et al., 2012). In Paracou, the PAI was derived from repeated UAV-borne lidar surveys, resulting in PAI mapping at 21 day and 1 m resolution between 2020 and 2022 over a 2.5 ha forest area. This PAI derived from UAV lidar was obtained by vertical integration of Plant Area Density (PAD) profiles previously recalibrated to match a TLS-derived PAD profile of a common 1-ha plot scanned in October 2019. This was required because the limited penetration of the UAV lidar yielded overestimation of raw PAD values (Vincent et al., 2023). This PAI was converted to LAI variation with a factor of 0.68, where the conversion factor is derived from other products.

Simulated LAI variation per leaf age cohort (Eqs 56-57, Maréchaux et al. submitted companion paper) were compared
qualitatively against the one derived from phenological cameras by Wu et al., (2016) at Tapajos and from the reanalysis of
Yang et al. (2023) at both sites. Wu et al. (2016) analysed 478 images collected over 24 months from 65 tree crowns and fitted
the transition from young to mature and from mature to old leaf pools, assumed to occur at 1 and 3 months, respectively. Yang
et al. (2023) used global satellite observations of the TROPOMI satellite Solar Induced Fluorescence (SIF) sensor as an
indicator of leaf photosynthesis variation, validated by *in situ* measurements, and set the transition from young to mature and
from mature to old leaf pools, occuring at 1.71 and 5.14 months, respectively. By comparison, simulated leaf age per cohort
depends on the individual leaf lifespan in TROLL 4.0 (see Maréchaux et al. submitted companion paper).

### 2.5 Evaluation of carbon and water fluxes

To assess the model's ability to simulate carbon and water fluxes, we evaluated gross primary productivity (GPP, kgC m$^{-2}$
year$^{-1}$) and evapotranspiration (ET, mm day$^{-1}$). We extracted GPP and latent heat flux (LE, W m$^{-2}$ half-hour$^{-1}$) from the
FLUXNET 2015 dataset (Pastorello et al., 2020). ET was derived from LE and temperature (T, in °C) using $ET =$
$\frac{LE \times 60 \times 30 \times 10^{-6}}{\lambda(T)}$ $with$ $\lambda(T) = 2.501 - (2.361 \times 10^{-3}) \times T$ (Allen et al., 1998). GPP was obtained from net ecosystem
exchange with the nighttime partitioning method (Reichstein et al., 2005). We summarised half-hourly GPP and ET into daily
values by calculating the daily mean and sum. TROLL 4.0 carbon fluxes were also compared with a remotely sensed product
of GPP derived from TROPOMI SIF using the formula $GPP = 15.343 \times SIF$ (Chen et al. 2022). We additionally computed
the light use efficiency (LUE in mol$_C$ mol$_{photons}$$^{-1}$) normalizing GPP by photosynthetic photon flux density (PPFD) and the
fraction of absorbed photosynthetically active radiation (fAPAR) derived from leaf area index (LAI) to explore carbon flux
environmental drivers independently of the overriding effect of light as in Bloomfield et al. (2023). We compared how the
fluxes depended on environmental drivers in both simulated and observed data. Using the FLUXNET 2015 dataset (Pastorello
et al., 2020), daily values of cumulative photosynthetically active radiation (PAR, mol m$^{-2}$), maximum vapour pressure deficit
(VPD, kPa), mean temperature (T, °C), and mean wind speed (WS, m s$^{-1}$) were calculated, and simulated and observed
responses of GPP, LUE, and ET to PAR, VPD, T and WS were compared. TROLL 4.0 water fluxes were assessed using the
relative variation of soil water content (RSWC, %) of the top horizon from the Paracou eddy flux tower (Bonal et al., 2008)
and the relative variation of soil water content of the top horizon reanalysed against the climatic water deficit at Tapajos
(Restrepo-Coupe et al., 2024). RSWC is defined as the daily mean of soil water content (m$^3$ m$^{-3}$) divided by the annual 95th
quantile of the daily mean.

All simulations were run using TROLL 4.0 (Maréchaux et al., submitted companion paper) wrapped in the R package *rcontroll*
(Schmitt et al., 2023) and encapsulated in a Singularity image (Kurtzer et al., 2017) leveraging a Python Snakemake workflow
(Köster et al., 2012) on a high performance computing platform using 100 cores. To compare simulations and observations,
we used the same metrics for all variables, regardless of their type, origin, spatial or temporal resolution: the goodness of fit
$R^2$ from linear regression with null intercept, the Pearson's correlation coefficient CC, the root mean square error of prediction
RMSEP, the standard deviation of the error of prediction SD.

## 3 Results

### 3.1 Forest structure, composition and dynamics

We calibrated background mortality rate ($m$) and crown radius scaling parameters ($a_{CR}$ and $b_{CR}$) at Paracou and Tapajos against
observed aboveground biomass, total number of stems and number of stems per 5-cm dbh classes, and found $m$=0.035,
$a_{CR}$=1.80 and $b_{CR}$=0.3860 at Paracou, and $m$=0.040, $a_{CR}$=2.45 and $b_{CR}$=0.7565 at Tapajos. For tree with 10-cm dbh, the calibrated
crown radius - dbh allometry (equ. 17 in Maréchaux et al., submitted companion paper) predicts a crown radius of 2.49 m at
Paracou and 2.03 m at Tapajos, a variation that falls well within the one reported globally (Jucker et al., 2024). The modelled
aboveground biomass, total number of stems and number of stems per 5-cm dbh classes were in good agreement with
observations (correlation coefficient, CC>0.99 at both sites, Fig. 1). The three parameter values were very similar across the
five best simulations, i.e. the ones minimising RRMSEP ($m$±0.0025, $a_{CR}$±0.1 and $b_{CR}$±0.057 at Paracou and $m$±0.01, $a_{CR}$±0.1
and $b_{CR}$±0.0285 at Tapajos, Table A3 & A4), and we used the values of the best simulation in all subsequent simulations.
Finally, in agreement with results on previous versions of the model, forest structure showed high sensitivity to the explored
parameters. Partial correlation coefficients (PCC) were around -0.4 for $a_{CR}$ and around 0.4 for $b_{CR}$ with number of stems,
aboveground biomass, and basal area. The background mortality rate $m$ also had a strong effect on aboveground biomass and
basal area with a PCC around -0.2 but little to no effect on number of stems (Fig. A4). The sensitivity of forest structure to $a_{CR}$,
$b_{CR}$, and $m$ was illustrated by a high variation of simulated forest structure when varying these parameters, for instance a basal
area variation of 3.9, 2.9, and 1.7 m2m-2 per standard unit of $a_{CR}$, $b_{CR}$ and $m$, respectively (Fig. A13).

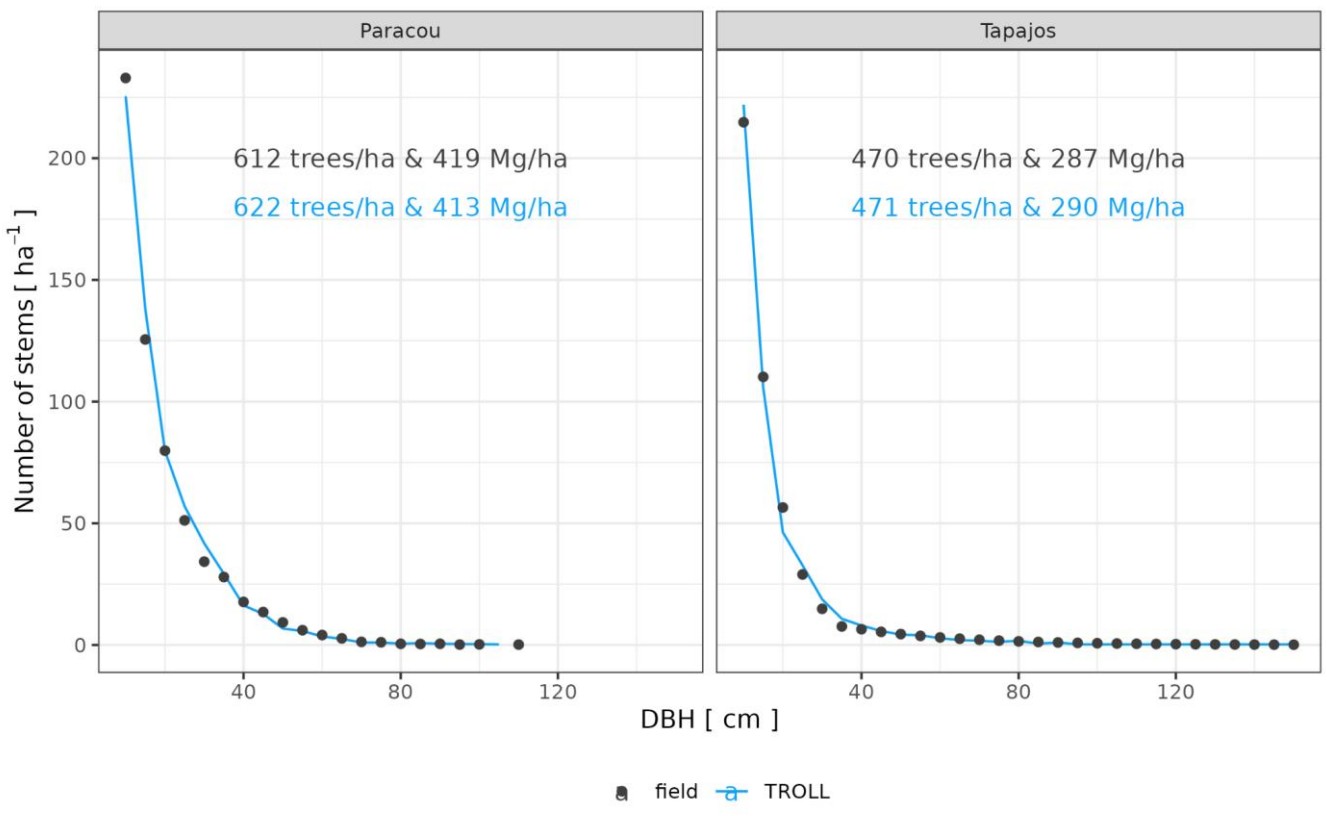


**Figure 1: Tree size structure at Paracou and Tapajos, expressed in terms of number of stems per 5 cm-dbh classes. Comparison between distributions simulated by TROLL 4.0 after calibration of $m$, $a_{CR}$ and $b_{CR}$ in blue and the ones derived from field inventories of trees with dbh >10 cm in black, at Paracou (left) and Tapajos (right). Observed (black) and simulated (blue) densities of trees with dbh > 10 cm, and aboveground biomass are also provided. All simulated values correspond to the end-state of a 600-year regeneration from bare ground with calibrated values for $m$, $a_{CR}$ and $b_{CR}$ at each site.**

After calibration, the top canopy height distribution simulated by TROLL 4.0 matched that measured by lidar aerial scanning (ALS), with a root mean square error of prediction (RMSEP) of the proportion of 1-m² pixels per 1-m height class below 0.8% and a correlation coefficient (CC) above 0.91, despite a slight overestimation of low canopy areas in Paracou, at heights below 20 m, and a slight underestimation of high canopy areas, above 40 m in Tapajos (Fig. 2). For example, in Paracou, 4% of the 1-m² pixels scanned by ALS had a canopy height around 25m. An RMSEP of 0.8% means that TROLL simulations could lead to 3.2 or 4.8% of pixels with a canopy height of 25m. TROLL 4.0 simulations also reproduced the forest understory structure characterised by basal area (BA) and number of stems distribution per 1-cm diameter classes for trees < 10 cm dbh at Paracou (Fig. 3). However, TROLL 4.0 underestimated the number of small trees (2,139 vs. 3,787 trees ha⁻¹), resulting in an underestimation of basal area (BA = 2.9 vs. 3.7 m² ha⁻¹).

382

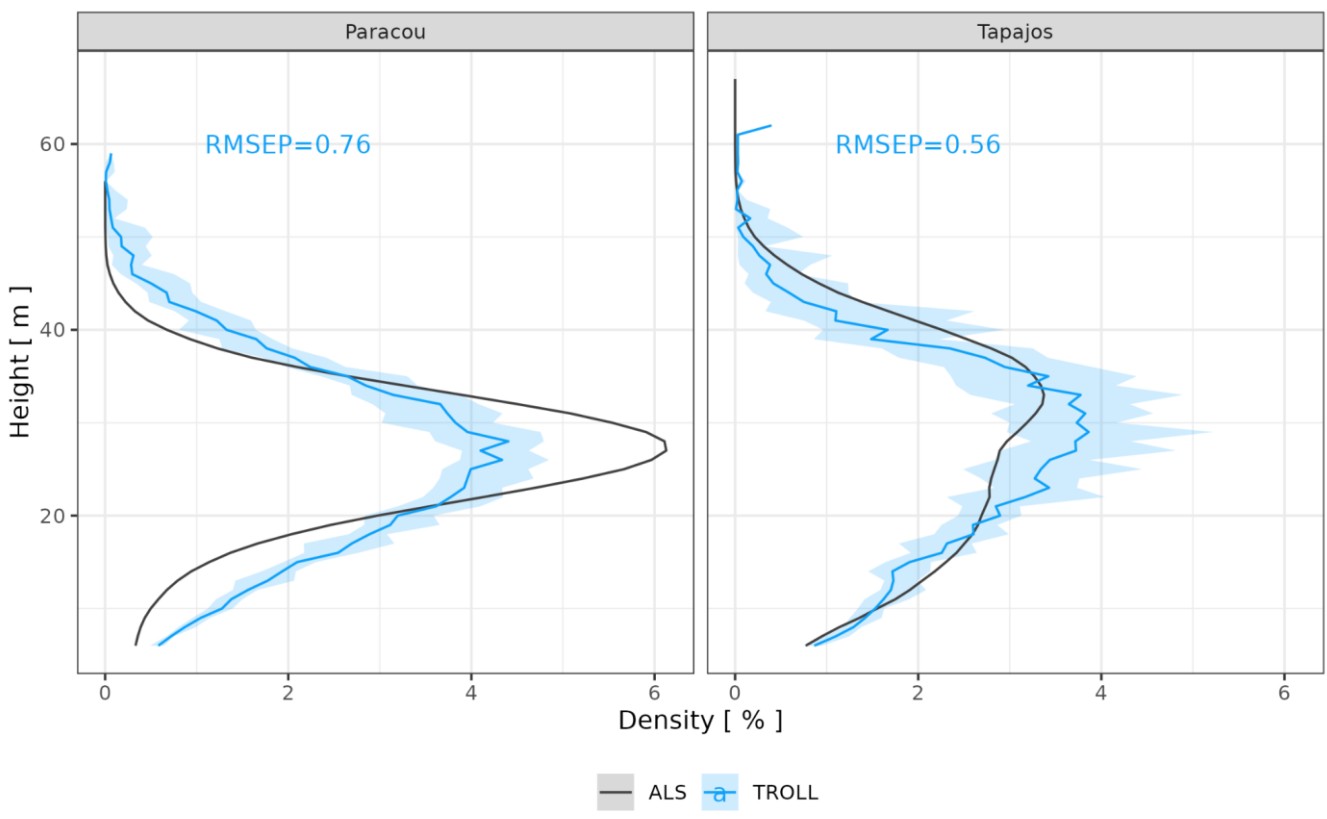

**Figure 2: Canopy height distribution at Paracou and Tapajos. For each 1-m² pixel of the ground, the top canopy height in that pixel (i.e. the height of the highest voxel with positive plant area density, or PAD, and located above this ground pixel) was determined, and its distribution across 1-m² pixels plotted as the proportion of 1-m² ground pixels (%, x-axis) with a given canopy height (m, y-axis, at 1-m resolution). The figure shows a comparison between distributions derived from PAD fields simulated by TROLL 4.0 (blue lines), and the ones derived from airborne laser scanning point clouds (black lines). Simulated values and their confidence intervals correspond to the end-state of simulations of ten 4-ha 600-year regeneration from bare ground for each site.**

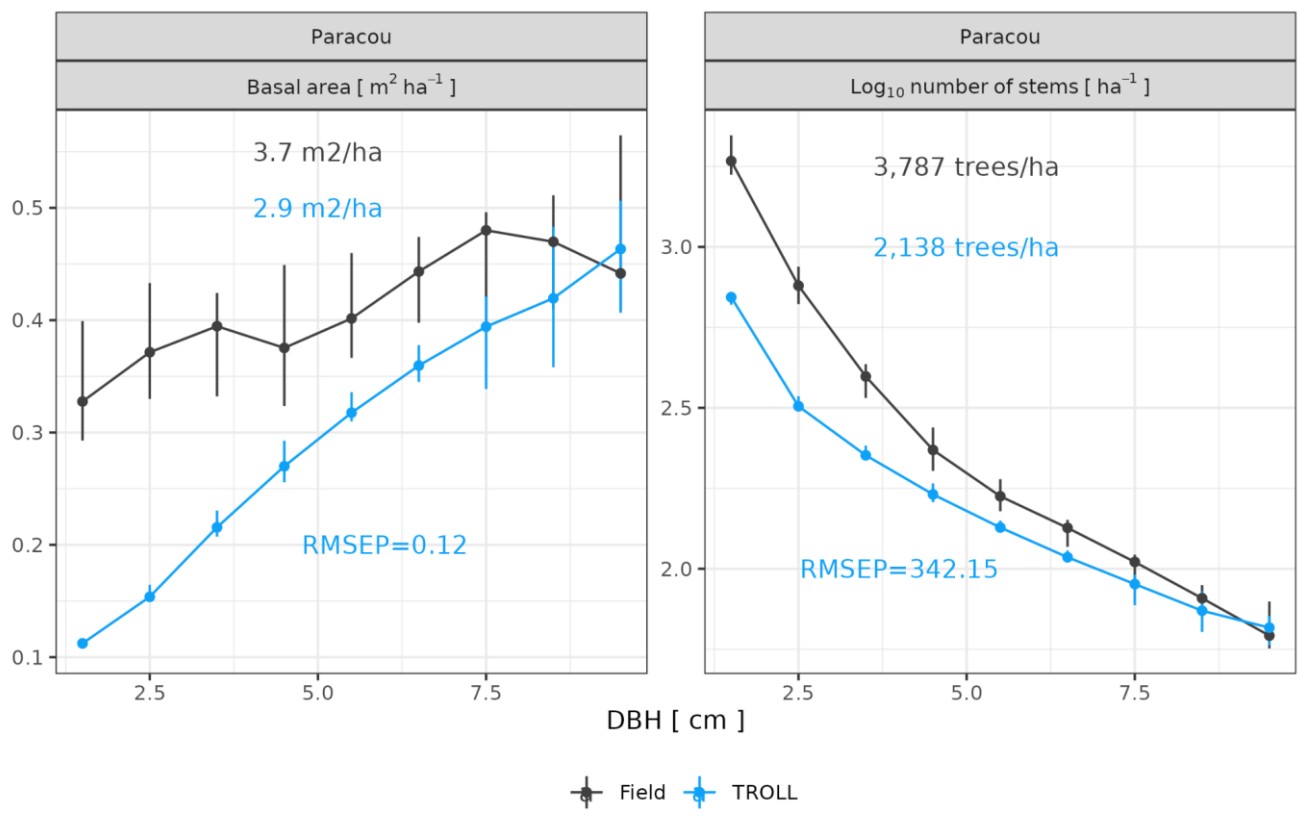

**Figure 3: Understory tree size structure at Paracou, expressed in terms of basal area distributions (left) and number of stems (right) per 1 cm-dbh classes. The figures compare distributions simulated by TROLL 4.0 in blue and field inventory observations in black. Simulated values and their confidence intervals correspond to the end-state of simulations of ten 4-ha 600-year regeneration from bare ground. Confidence intervals at 95 % are shown with error bars and are based on variations among plots (9 plots of 1 ha) for the observations. Simulated (blue) and observed (black) total basal area (left) and densities (right) for trees with dbh >1 cm and < 10 cm are also provided. To the best of our knowledge, similar data was not available in Tapajos.**

At Paracou, the simulated and observed species rank-abundance curves were similar (Fig. 4), with a RMSEP of 3.67 trees ha$^{-1}$ and a CC of 0.93, but with an underestimation in the abundance of dominant species and an overestimation in the abundance of rare species resulting in a higher evenness overall. At Tapajos, the simulated and observed rank-abundance curves displayed similar patterns as at Paracou (RMSEP=3.62 trees ha$^{-1}$ and CC=0.94) but amplified , with a strong underestimation of the abundance of dominant species and an overestimation of the abundance of rare species.

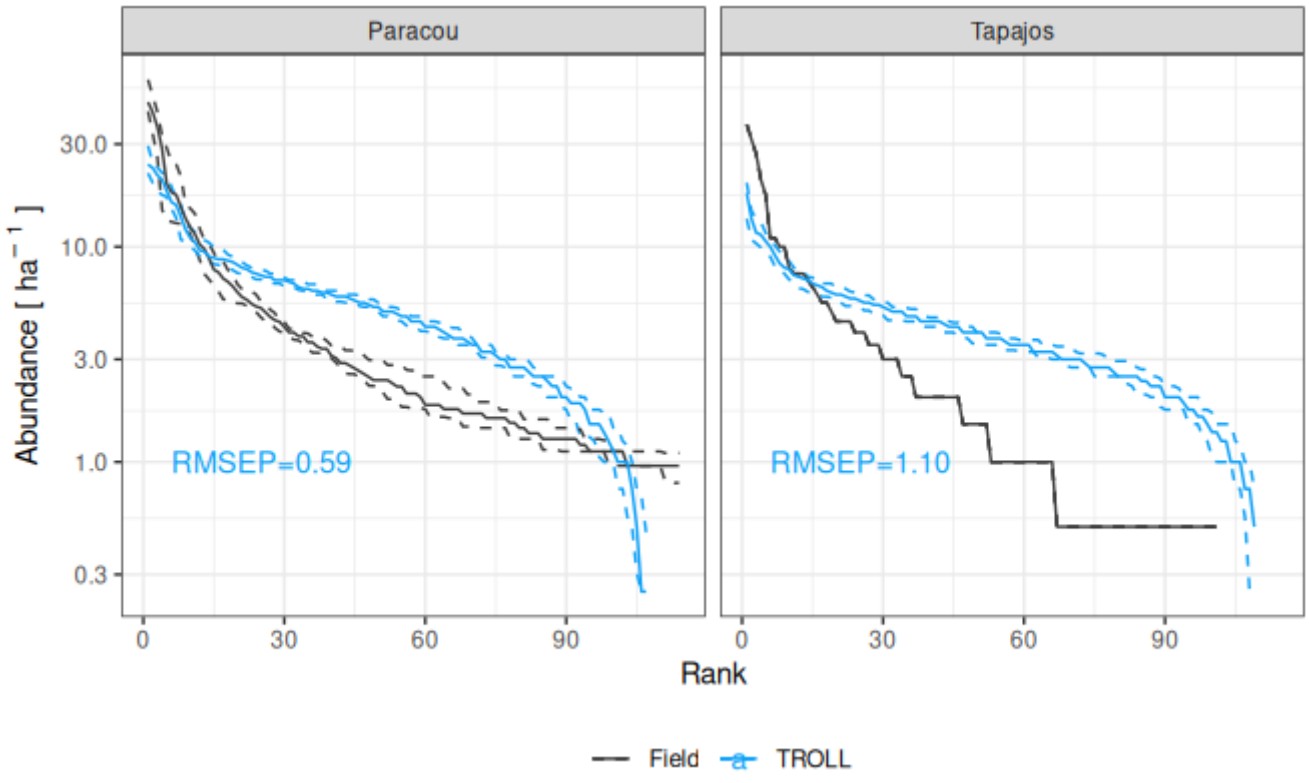


**Figure 4: Species-rank abundance curves at Paracou and Tapajos. Comparisons between curves simulated by TROLL 4.0 (blue)**
**and derived from field inventories at both sites. Simulations included 114 and 113 species at Paracou and Tapajos respectively.**
**Curves derived from inventories were cut at the 114th species. Simulated values and their confidence intervals correspond to the**
**end-state of ten 4-ha 600-year regeneration from bare ground. Confidence intervals at 95 % are shown with error bars and are**
**based on variations among plots for observations.**
Functional trait distributions simulated by TROLL 4.0 were consistent with empirical ones at Paracou and Tapajos (Fig. 5),
with a CC from 0.91 to 1.00 for all traits at both sites, except for leaf area at Paracou (CC=0.74) and Tapajos (CC=0.87).
However, abundances of low wood density trees, high LA trees, and high LMA trees were underestimated in simulations
when compared to observations at Paracou.

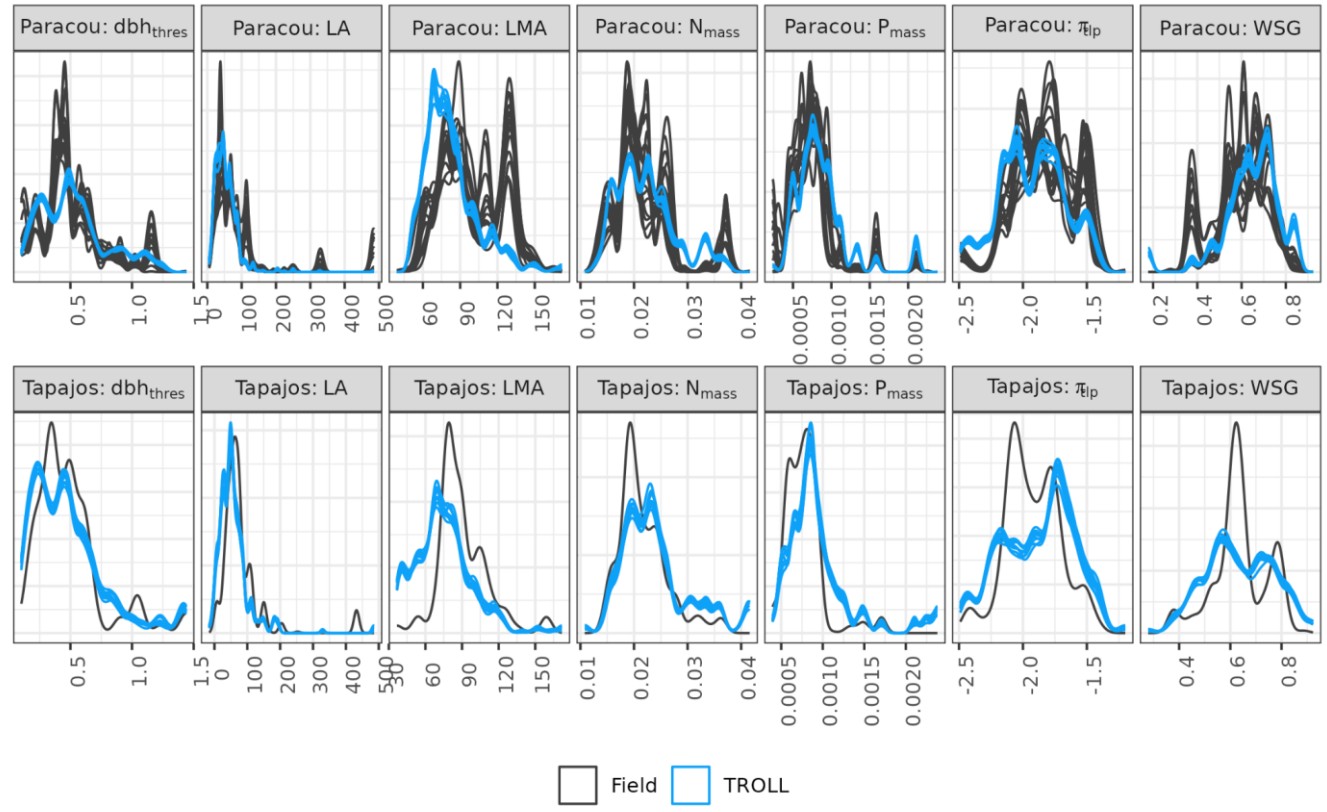


**Figure 5: Functional trait distributions at Paracou and Tapajos. Distributions derived from field inventories (black) were based on botanical identification at the species level in Paracou and the genus level in Tapajos. Simulated distributions (blue) were based on the final stage of ten 4-ha 600-year regeneration from bare ground. Confidence intervals are shown with repeated lines and are based on variations among plots for observations and among repetitions for simulations. dbh$_{thres}$: maximum diameter in m, LA: leaf area in cm$^2$, LMA: leaf mass per area in g cm$^{-3}$, N$_{mass}$: leaf nitrogen content per dry mass in g g$^{-1}$, P$_{mass}$: leaf phosphorus content per dry mass in g g$^{-1}$, $\pi_{tlp}$: leaf water potential at turgor loss point in MPa, WSG: wood specific gravity in g cm$^{-3}$.**

Forest dynamics simulated by TROLL 4.0 were consistent with the ones estimated from field inventories (Fig. 6).
Simulated individual-tree growth-size relationship were comparable to the ones retrieved from inventories (simulated mean
of 0.18 cm yr$^{-1}$ against 0.13 cm yr$^{-1}$) with an expected bell-shaped relationship (Hérault et al. 2011) and similar high
variation (Schmitt, Hérault & Derroire 2023). Simulated death rates also showed magnitude and variation similar to observed
ones (simulated mean of 1.73 % yr$^{-1}$ against 2.60 % yr$^{-1}$ observed at Paracou but with consistent and overlapping ranges).
Despite overlapping confidence intervals between simulated and observed death rate variation across size, simulated mean
death rates tended to be lower for medium to large trees, especially between 30 and 75 cm dbh, than observed.

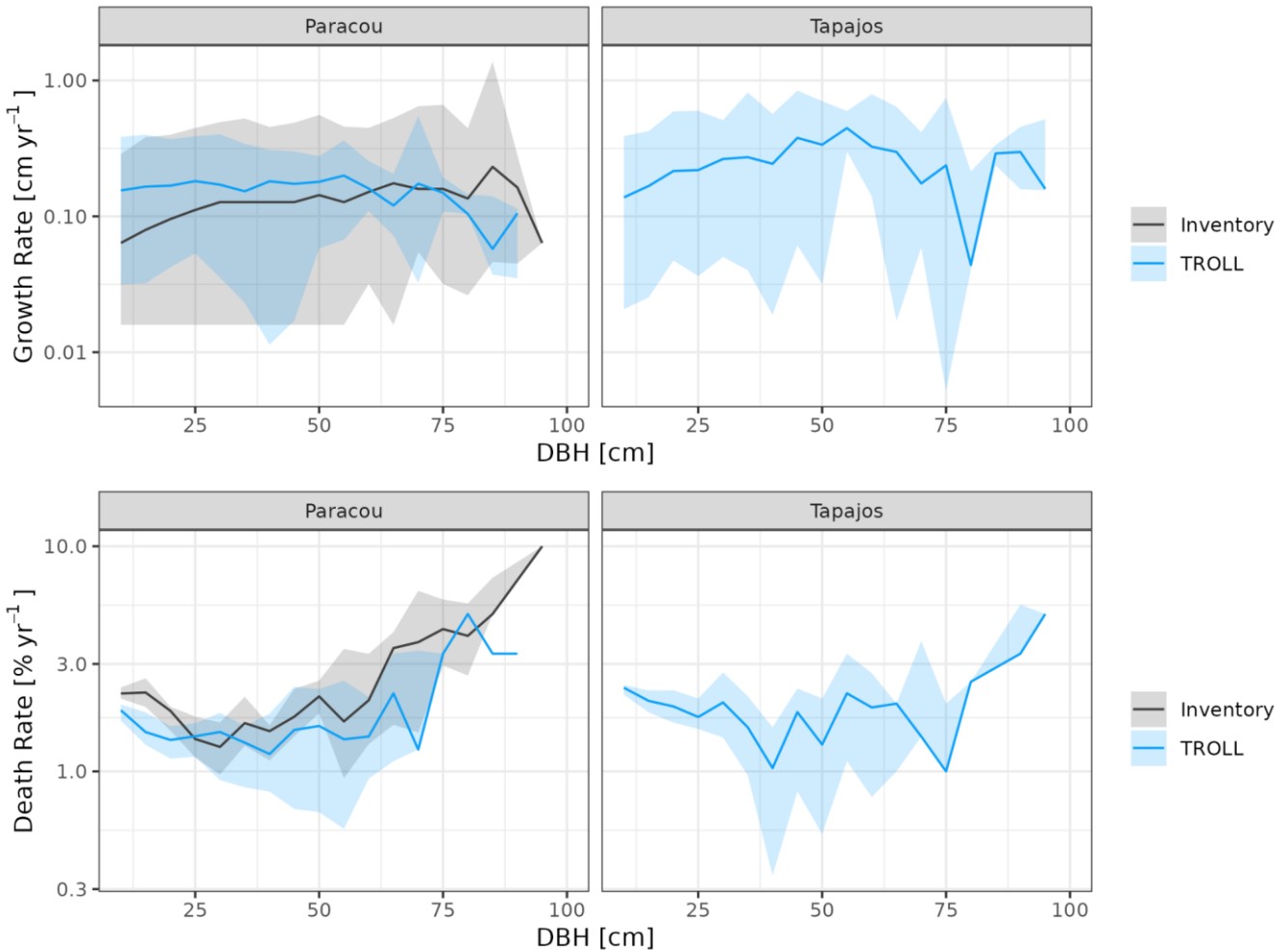


**Figure 6: Forest dynamics at Paracou and Tapajos, expressed in terms of individual-tree growth rate (top) and death rate (bottom) both per 5 cm-dbh classes across 10 years. The figures compare distributions simulated by TROLL 4.0 in blue and multiple field inventory observations from six 6.25-ha plots in Paracou from 2003 to 2013 in black. Simulated values and their confidence intervals correspond to ten repetitions of 10-year simulations starting from the end-state of 600-year regeneration from bare ground with calibrated parameters at each site. Confidence intervals at 95 % are shown with shaded areas and are based on variations among plots (6 plots of 6.25 ha) for the observations.**

### 3.2 Leaf phenology

The calibration of the three parameters of the leaf shedding module against observed litterfall illustrated how each parameter affects the simulated timing and intensity of the litterfall peak during the dry season, with no or little effect on the background litterfall rate (Fig. A5) but revealing a strong positive effect of $a_{T,o}$ and $b_{T,o}$ on the peak day of litterfall and negative effect on the ratio of the peak of litterfall, and a weak effect of $\delta_o$ on the peak of litterfall. As anticipated, litterfall calibration was independent of the forest structure calibration (Fig. A4). Calibration resulted in a best-fit $a_{T,o}$ value of 0.2, and a $b_{T,o}$ value of 0.015 at both sites. The calibrated $\delta_o$ differed across sites ( $\delta_o$=0.1 at Paracou and $\delta_o$=0.2 at Tapajos). The simulated seasonal

variation of litterfall at Paracou and Tapajos shows qualitative agreement with the observed data (Fig. 7). Both empirical and
simulated data showed a marked peak in litterfall during the dry season, despite a clear under-estimation of simulated litterfall
flux during both wet and dry seasons, particularly at Tapajos, and a delayed peak during the dry season, particularly at Paracou,
in comparison to observations.

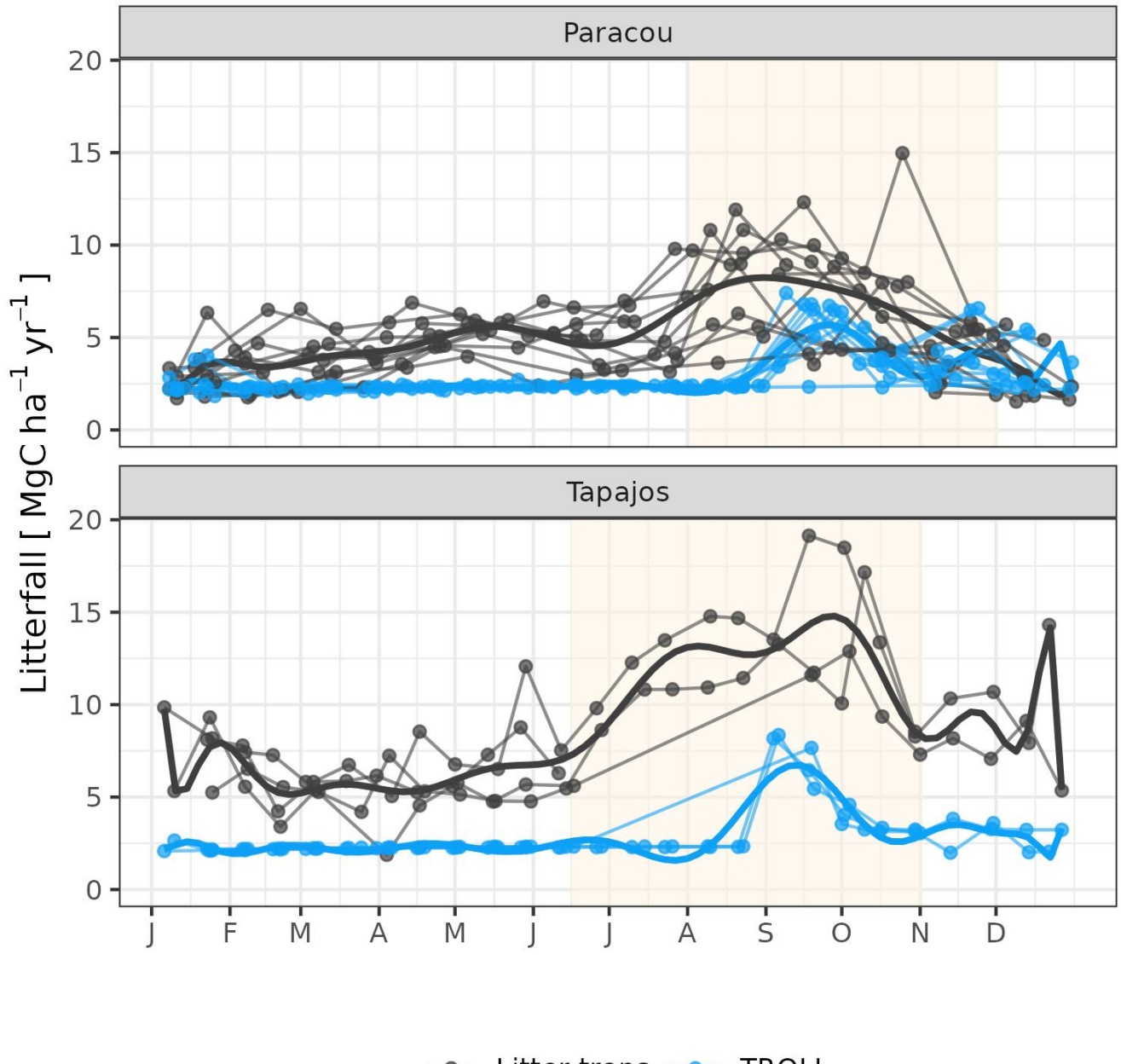


**Figure 7: Litterfall annual cycle from fortnightly litterfall fluxes at Paracou and Tapajos. Each thin line represents one year with points showing values at sampling dates, the thick lines represent polynomial smoothing among years, and the vertical yellow bands in the background correspond to the site's climatological dry season. Simulated values correspond to the last 10 years of 20-year simulations starting from the end-state of 600-year regeneration from bare ground with calibrated parameters at each site.**

The empirical LAI datasets displayed strikingly different results, illustrating the challenge of estimating LAI with confidence in dense tropical forests (Fig. 8, Tab. S2). MODIS-derived LAI displayed almost no seasonality with mean LAI values around 6.0 $m^2\,m^{-2}$ at both sites. At Paracou, LAI derived from UAV-borne lidar showed a clear seasonality, with lowest values around 5.5 $m^2\,m^{-2}$ from April to June and highest values of almost 6.0 $m^2\,m^{-2}$ in December, at the end of the dry season. At Tapajos, LAI derived from terrestrial lidar showed no seasonality, around 5.8 $m^2\,m^{-2}$ throughout the year, but LAI derived from phenological cameras (PhenoCams) did display some seasonality, with lowest values at 5.5 $m^2\,m^{-2}$ in June and highest values above 6.0 $m^2\,m^{-2}$ in December, at the end of the dry season. These observations were compared with simulations. At Paracou, simulated LAI matched the one derived from UAV-borne lidar, both showing an increase during the dry season (CC=0.84, RMSEP=0.11 $m^2\,m^{-2}$). At Tapajos, simulated LAI matched the empirical LAI derived from PhenoCams (CC=0.91, RMSEP=0.15 $m^2\,m^{-2}$; Table S2).

The different datasets gathered to estimate LAI dynamics per cohorts also showed contrasted patterns (Fig. 9 and Fig. A6). At Tapajos, PhenoCams indicate a maximum young leaf LAI reached during the dry season and a minimum during the wet season, with inverse patterns for old leaf LAI. TROLL 4.0 simulations yielded patterns consistent with these observations (Fig. 9). However, Yang et al.'s (2023) reanalysis predicts the exact opposite trends for young and old leaves, with a maximum young leaf LAI during the wet season and a minimum during the dry season. At Paracou, we could only compare simulated trends against Yang et al. (2023)'s reanalysis and the match was relatively poor (Fig 8).

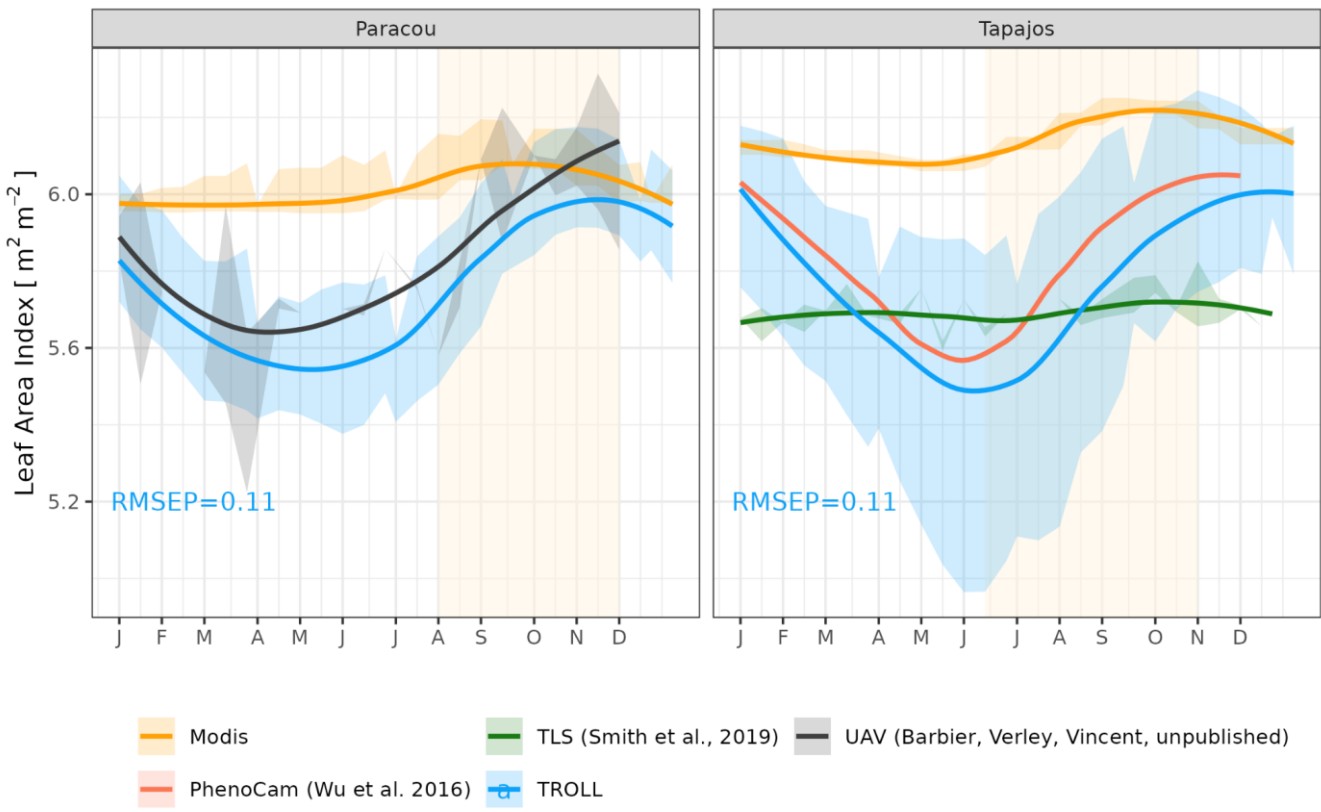

467

**Figure 8: Mean annual cycle of leaf area index (LAI) at Paracou and Tapajos, derived from fortnightly means, from different sources (see methods). Bands are the intervals of means across years, and the vertical yellow bands in the background correspond to the site's climatological dry season. Simulated values correspond to 10 years of simulations starting from the end-state of 600-year regeneration from bare ground with calibrated parameters at each site.**

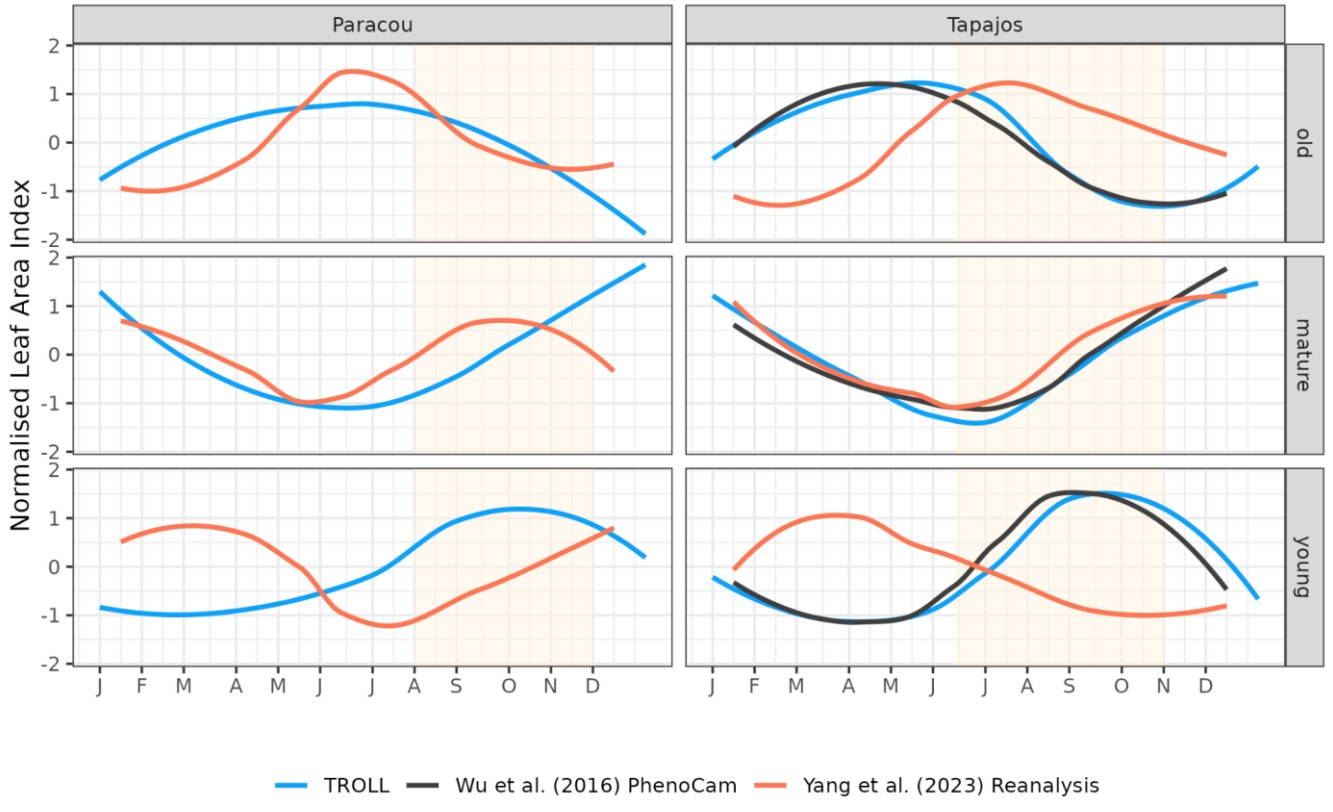

472

**Figure 9: Mean annual cycle of normalised leaf area index per leaf age cohorts, derived from fortnightly means, at Paracou and Tapajos. Note that the three leaf age cohorts (young, mature and old leaves) are not defined the same way in the three independent sources. Leaf age per cohort depends on the individual leaf lifespan in TROLL 4.0 (see Maréchaux et al., submitted companion paper), while the transition from young to mature and mature to old are respectively fixed to 1.71 and 5.14 months in Yang et al. (2023) and fitted to 1 and 3 months in Wu et al. (2016). The vertical yellow bands in the background correspond to the site's climatological dry season. See figure A6 for absolute variation per cohort, site and dataset. Simulated values correspond to 10 years of simulations starting from the end-state of 600-year regeneration from bare ground with calibrated parameters at each site.**

### 3.3 Water and carbon fluxes

TROLL 4.0 captured the seasonality of gross primary productivity (GPP) observed at the two sites, with an increase before the onset of the dry season, reaching its maximum during the dry season, and a decrease starting before or at the onset of the wet season (Fig. 10 and see Fig. A7 for interannual variations, Tab. S2). Comparison with eddy flux estimates with simulations were high both at Paracou (CC=0.60) and Tapajos (CC=0.46). TROLL 4.0 overestimated GPP at both sites, particularly during the dry season, with a RMSEP of 0.75 and 1.12 kgC $m^{-2}$ $year^{-1}$ when compared with both eddy flux and TROPOMI SIF estimates at Paracou and Tapajos, respectively.

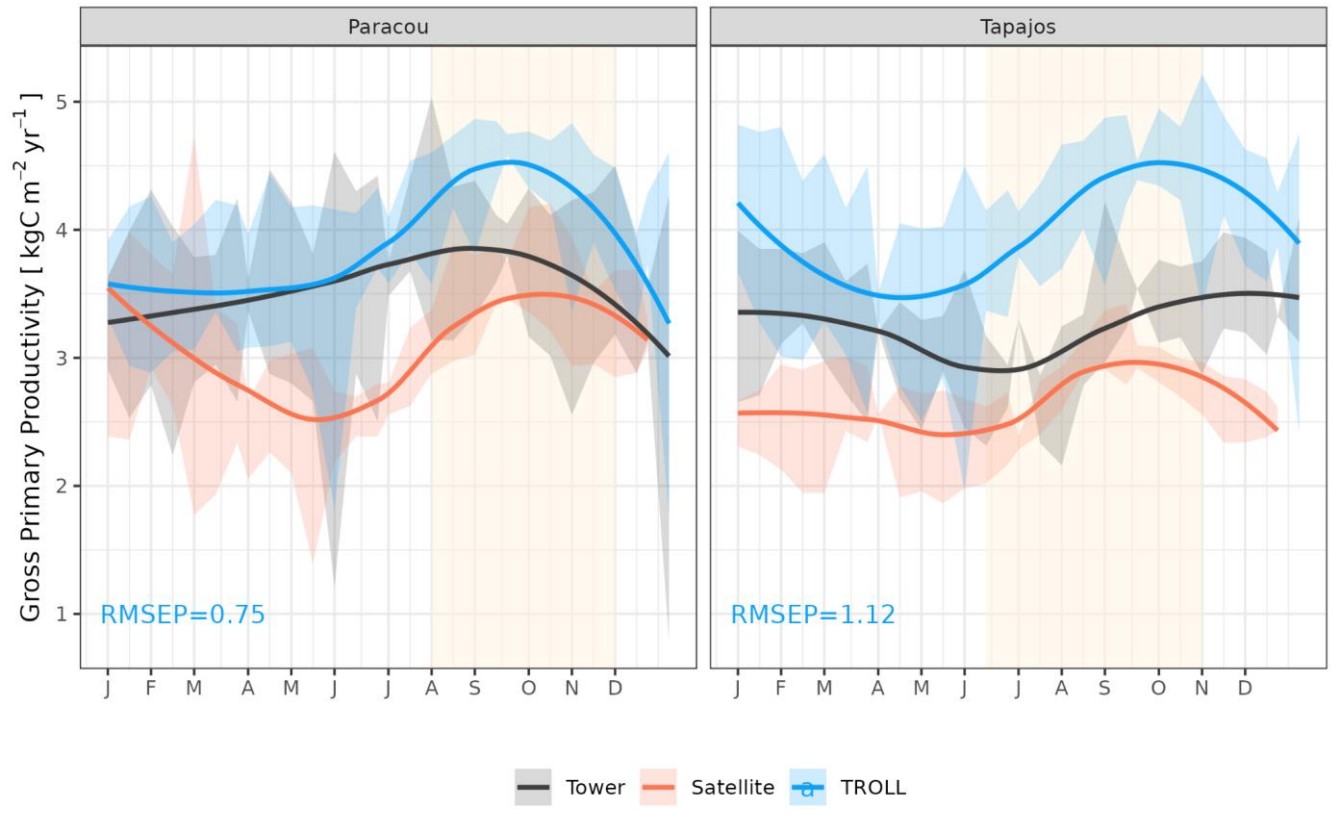

487

**Figure 10: Mean annual cycle of gross primary productivity for Paracou and Tapajos, derived from fortnightly means. The red lines represent the gross primary productivity estimated from TROPOMI SIF while the black lines represent the one derived from eddy flux measurements, and the blue lines the simulated gross primary productivity with TROLL 4.0. Bands are the intervals of means across ten years, and the vertical yellow bands in the background correspond to the site's climatological dry season. Simulated values correspond to 10 years of simulations starting from the end-state of 600-year regeneration from bare ground with calibrated parameters at each site. Inter-annual variations are shown in Figure A7.**

The seasonality of water flux was captured by TROLL 4.0 (Fig. 11 and see Fig. A8 for interannual variations, Tab. S2), with a pronounced increase in evapotranspiration (ET) during the dry season at both sites, and leading to CC of 0.66 and 0.70 when compared with eddy flux estimates at Tapajos and Paracou respectively. Although intra-annual variations of simulated and observed values overlapped, TROLL 4.0 tended to overestimate ET in Tapajos during the dry season, leading to RMSEP values of 0.60 and 0.75 mm day$^{-1}$ when compared with eddy flux estimates at Paracou and Tapajos respectively. The partitioning of evapotranspiration between canopy evaporation, soil evaporation and tree transpiration (Fig. A9) showed that most of the evapotranspiration is due to tree transpiration in the dry season, while canopy evaporation is an important part of the total evapotranspiration in the wet season (Kunert et al., 2017). TROLL 4.0 also captured the seasonality in RSWC of the top soil layer at Paracou and Tapajos (Fig. A10, Table A2, see Fig. A11 for absolute variation with varying depth), with a high RSWC in the wet season close to 100% and a sharp decrease in RSWC in the dry season, although overall smoother in simulations than field estimates.

505

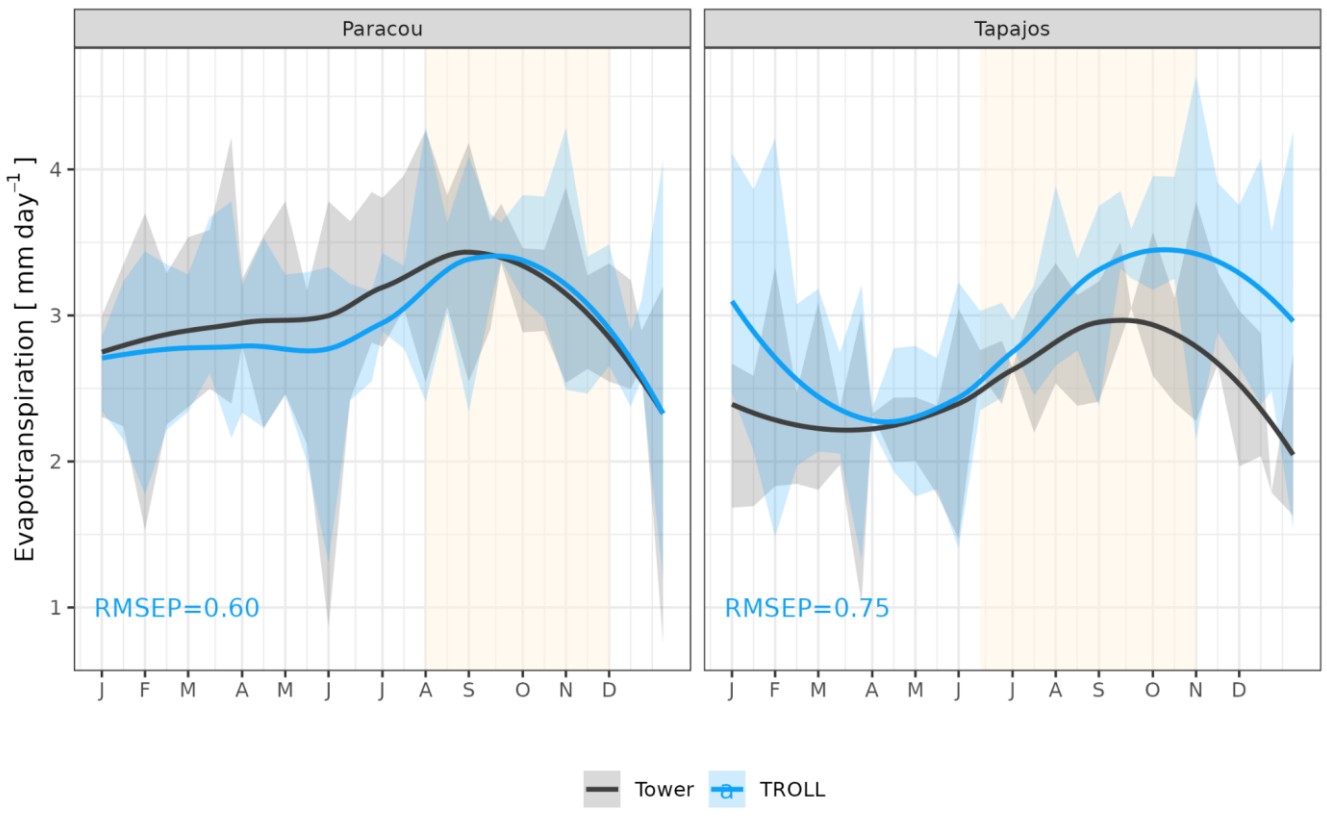

506

**Figure 11: Mean annual cycle of evapotranspiration for Paracou and Tapajos, derived from fortnightly means. The black lines represent the evapotranspiration derived from eddy flux measurements and the blue lines the evapotranspiration simulated with TROLL 4.0. Bands are the intervals of means across years, and the yellow vertical bands in the background correspond to the site's climatological dry season. Simulated values correspond to 10 years of simulations starting from the end-state of 600-year regeneration from bare ground with calibrated parameters at each site. Inter-annual variations are shown in Figure A8.**

Both eddy flux-derived and simulated GPP showed a positive logarithmic relationship with cumulative incoming PAR and maximum VPD, and a positive linear relationship with mean temperature at daily scale (Fig. 12). Similarly, controlling for absorbed light, both eddy flux-derived and simulated LUE showed a negative logarithmic relationship with maximum VPD and a negative linear relationship with mean temperature at daily scale (Fig. A12). Limitations of LUE at high VPD and T values were however lower in simulations than in eddy flux- or SIF-derived estimates. TROLL 4.0 predicted a higher PAR conversion to carbon under high irradiance, high VPD and high temperature conditions when compared to eddy flux estimates, consistent with the higher dry-season GPP in simulations (Fig. 10). Responses of SIF-derived GPP to climatic variables were weak in comparison to simulated and eddy flux derived GPP. Simulated ET was positively correlated with maximum VPD, cumulative PAR and mean temperature, similarly to eddy flux derived ET (Fig. 13). At Paracou, the relationships between environmental drivers and simulated ET, closely aligned with the ones obtained from eddy flux estimates. However, at Tapajos, simulated ET was overestimated under high irradiance, VPD, temperature and windy conditions in comparison to

523  eddy flux estimates. Simulated GPP and ET at both sites were more strongly controlled by environmental variables (higher $R^2$
524  in Figs. 12-13) than eddy flux derived GPP and ET.

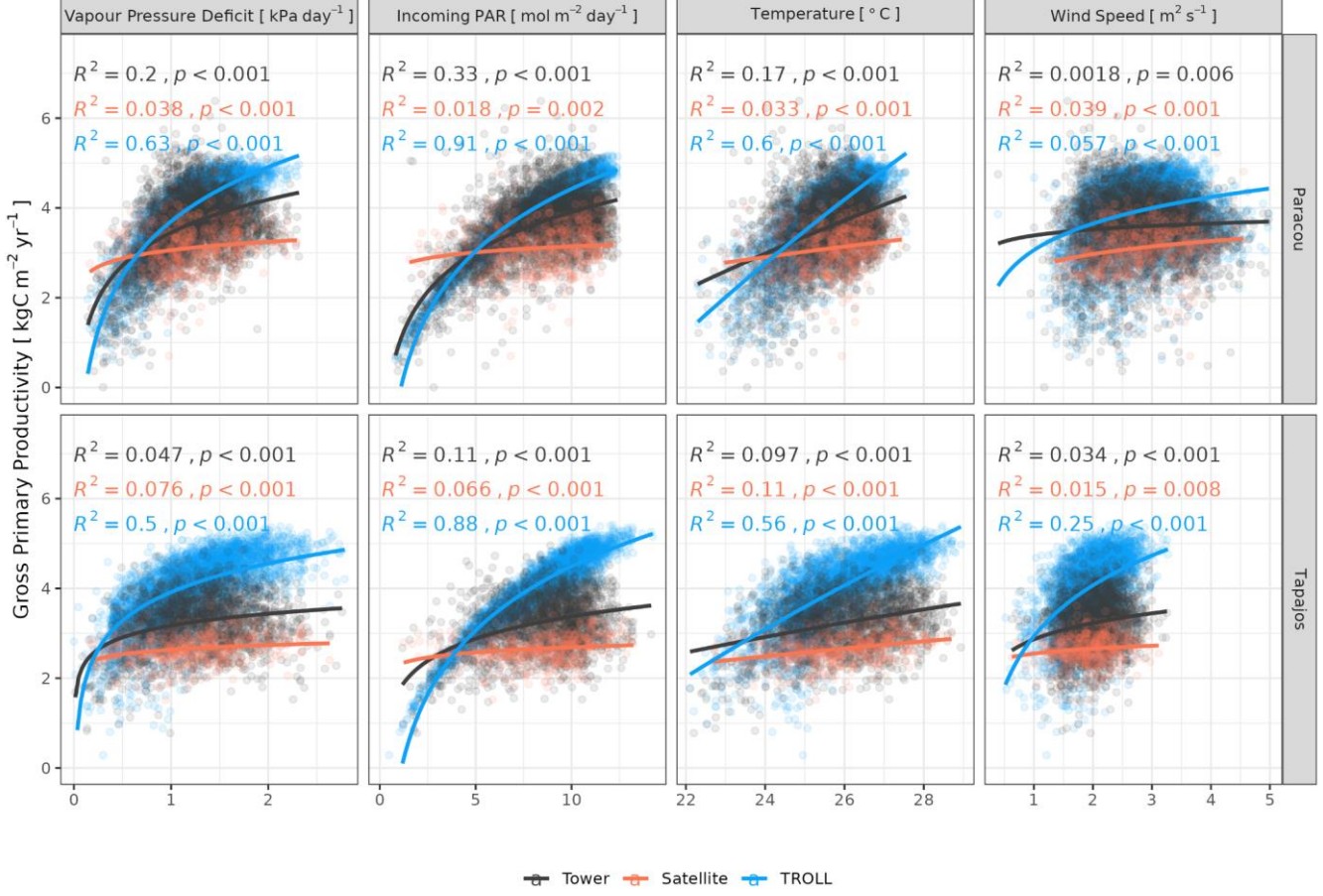

525

**Figure 12: Daily averages of gross primary productivity as a function of daily maximum vapour pressure deficit, total incoming photosynthetically active radiation, average temperature, and average wind speed for model-, satellite- and eddy flux-based estimates at Paracou (top) and Tapajos (bottom). Lines illustrate the linear regression of form y ~ log(x), and text the squared Pearson's R correlation coefficient.**

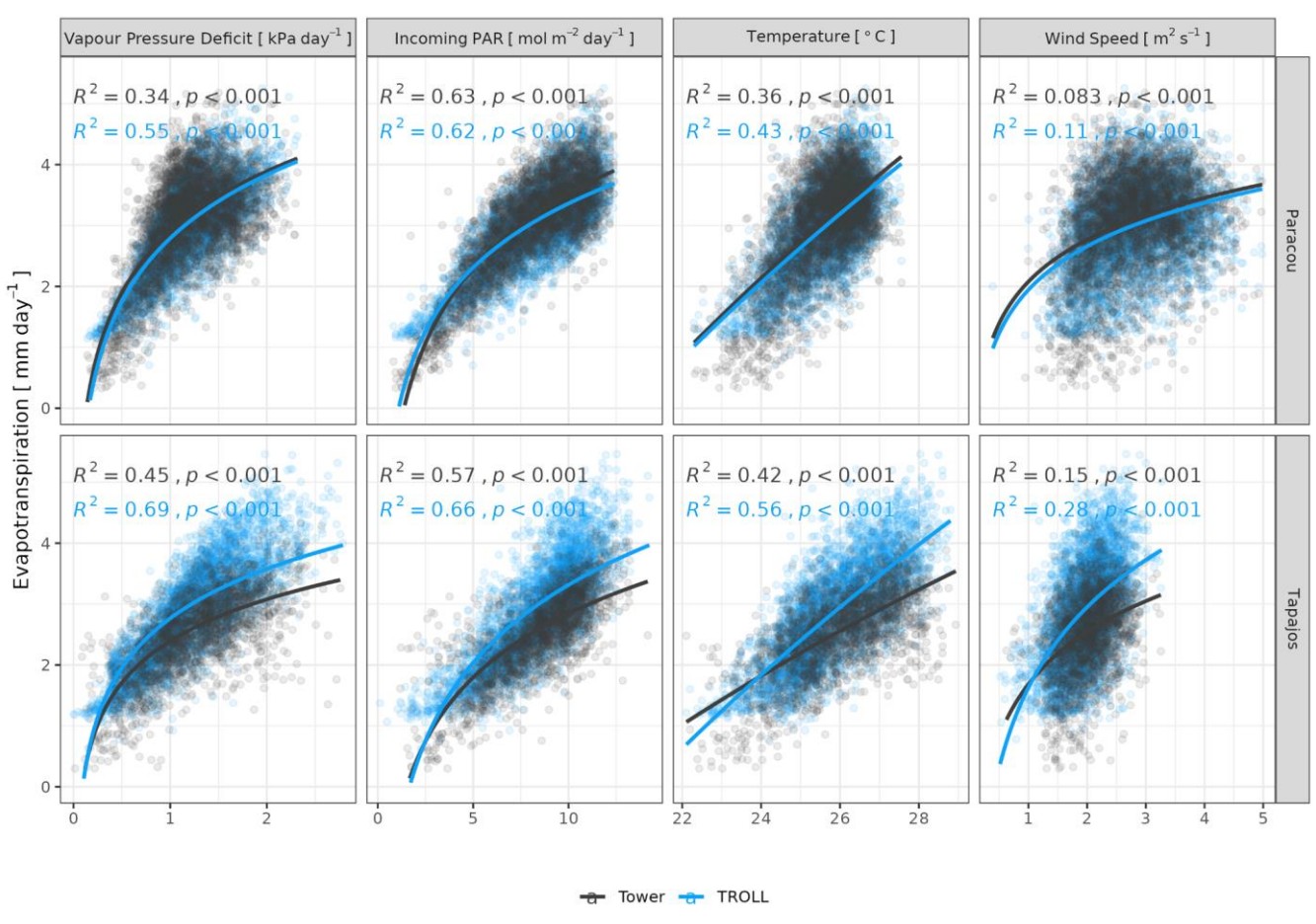

**Figure 13: Daily total evapotranspiration as a function of daily maximum vapour pressure deficit, total incoming photosynthetically active radiation, average temperature, and average wind speed for model- and eddy flux estimates at Paracou and Tapajos. Lines illustrate the linear regression of form y ~ log(x), and text the squared Pearson's R correlation coefficient.**

## 4 Discussion

Here we tested the performance of TROLL 4.0 in reproducing observed forest structure and diversity, water and carbon fluxes, and leaf dynamics. We conducted a detailed model evaluation for two Amazonian rainforest sites, Paracou and Tapajos, presenting contrasting climate and soil properties. Both sites have been intensively monitored over the past decades, and we compared the model outputs with available data. We now discuss the consistencies and discrepancies between simulated and observed patterns, potential uncertainties in our results, and the advantages and possible improvements of TROLL 4.0.

### 4.1 Forest structure, composition and dynamics

TROLL 4.0 was found to jointly simulate realistic forest structure and species composition (Maréchaux et Chave, 2017). The calibration of three global parameters led to simulated number of stems across size classes and basal area or aboveground

biomass in good agreement with observations from forest inventories above 10-cm dbh. Also, aerial lidar data allowed forest structure to be assessed independently of calibration data. This revealed a good ability of TROLL 4.0 to simulate the horizontal and vertical structure of both forests, which is promising for various applications, including biomass estimation (Knapp et al., 2018). Similarly, the multiple inventories at Paracou from 2003 to 2013 revealed a good ability of TROLL 4.0 to simulate forest dynamics with both bell-shaped growth-size relationship and tree mortality. Comparing the different sources of mortality with tree size between observations and simulations would be useful to assess the representation of mortality processes, although documenting mortality sources is often challenging (McDowell et al., 2018). Understory inventories at Paracou also allowed us to independently evaluate TROLL 4.0's ability to simulate tree community structure in the 1 to 10-cm tree diameter range. TROLL 4.0 simulated the distribution of smaller trees reasonably well, although it underestimated individuals from the smallest cohorts. This underestimation of the density of small trees may be partly explained by the fact that the one-metre resolution of the voxel grid used in TROLL 4.0 only allows for one tree per square metre of ground, whereas smaller trees may be squeezed into certain areas of the understorey. However the number of simulated small stems remains lower than the maximal potential number in simulations. Another explanation could be a lack of light heterogeneity and associated trait variation in the understorey in simulations in comparison with observations (Montgomery and Chazdon, 2001), thus limiting the opportunities for recruitment and survival of small stems. Explorations of simulated micro-environmental variations within the canopy (de Frenne et al., 2019) and inclusion of trait ontogenetic shifts (Fortunel et al., 2019) and trait plasticity (Xu et al., 2017; Lamour et al., 2023) could further help understand and improve TROLL's ability to simulate forest structure and composition in the understory.

TROLL 4.0 attributes individual trees to botanical species and it permits tree functional traits to vary within species. It thus provides a finer-grained description of biodiversity compared to models based on plant functional types (e.g. Longo et al., 2018), and uses a description matching the one of ecologists, in contrast with taxonomy-free continuous trait spectrum approaches (e.g. Sakschewski et al., 2015). The simulated species composition presented classically observed L-shaped profile of species rank abundance distribution in the two sites, but with an over-estimated species evenness resulting in under-abundant dominant species and over-abundant rare species, as already observed in previous versions of the model (Maréchaux and Chave, 2017). Several simulation factors could have resulted in the overestimation of species evenness. The species trait values were extracted from global databases and partially imputed and may therefore not represent the true trait values for the region concerned, which could affect the behaviour of individual species in the model. However, as this noise is random, it seems unlikely that the global values and imputation have led to the skewed species abundance. More likely, the simulations used an external seed rain representing immigration from a continuous forest matrix. We here implemented a homogeneous seed rain, in which all species are equally-abundant, as a conservative test of the model's ability to represent community assembly. Here, the simulated composition after regeneration from bare ground is determined by species traits and their simulated effect on demographic processes and species fitness, rather than prescribed differences in seed rain. However, this homogeneous, and therefore unrealistic, seed rain maintains diversity in the simulated forest with a rescue effect, and can dampen species dominance by promoting less dominant species through a high immigration. The effects of the representation of seed

production, dispersal and recruitment on simulated communities should be further explored in the future, especially for projections under disturbance scenarios where forest regeneration is key (Diaz-Yanez et al., 2024, Hanbury-Brown et al., 2022).

TROLL 4.0 also explicitly simulates forest functional diversity in the community. Simulated functional trait distributions matched well the observed distributions at both sites, as already observed in previous versions of the model (Maréchaux and Chave, 2017). In Paracou, the main discrepancies were the lack of individuals with high LMA (between 120 and 150 g m$^{-2}$), low wood specific gravity (below 0.4 g cm$^{-3}$) and/or high leaf area (above 100 cm$^2$). In contrast, in Tapajos, the model tended to simulate lower LMA and less negative turgor loss points on average. Since trait combinations are structured at the species level, and trait integration is high dimensional in tropical forests, with decoupled leaf and wood economic spectra (Baraloto et al., 2010) and weak associations between leaf turgor loss point and other leaf traits (Maréchaux et al., 2019), these discrepancies can be more easily interpreted at Paracou where the trait distributions are built on species-level (and not genus-level) information. Regarding the lack of high LMA individuals, TROLL 4.0 underestimated the abundance of common species such as *Lecythis persistens* or *Licania alba*, which present high LMA. These species come from genera that are hyperdominant across the Amazon basin (ter Steege et al., 2013) but may be underrepresented in the simulations due to the overestimation of species evenness in TROLL 4.0 as discussed above. The lack of light wood and high leaf area individuals can be related to the underestimated abundances of light demanding and pioneer species with fast growth (Chave et al., 2009), such as the ones of the genus *Cecropia*. These species are known to quickly colonise forest gaps under high light conditions, thanks to fast carbon assimilation and growth, and the dispersal of a high number of small, potentially dormant, seeds, leading to an omnipresence of these species in the forest seed bank (Holthuijzen and Boerboom, 1982; Alvarez-Buylla and Martínez-Ramos, 1990). In TROLL 4.0, the seed-size mediated tolerance-fecundity trade-off (Muller-Landau et al., 2010) is assumed to be perfectly equalising, and all species present in the local seed bank and able to thrive under the local light availability have the same probability of being  recruited per seed. However, this assumption likely disadvantages gap-affiliated species with a colonisation strategy, and could easily be revisited in future model developments.

## 4.2 Leaf phenology

We calibrated and evaluated the new phenology module of TROLL 4.0. The calibration of the three module parameters ($a_{T,o}$, $b_{T,o}$ and $\delta_o$), which together control the variation of old leaf fall under drying conditions, was conducted using litterfall trap data. This resulted in a realistic litterfall seasonality with a peak during the dry season as already documented (Manoli et al., 2018, Chave et al., 2010, van Langenhove et al., 2020). Interestingly, the calibration resulted in the same values for two parameters at the two sites ($a_{T,o}$, $b_{T,o}$) and close values for the third one ($\delta_o$) to which the simulated litterfall pattern is less sensitive (Fig. A5). At both sites, simulations with the mean value of the third parameter resulted in similar evaluations (not shown). This suggests a good transferability of the phenology module across sites without the need for site-specific calibration, although this remains to be further tested at additional sites and in contrasted conditions (e.g. Restrepo-Coupe et al., 2017). A faster shedding of old leaves was assumed to depend on soil water potential in the root zone, rather than soil water content, on

individual leaf water potential at turgor loss point, and on tree size. These are biologically reasonable hypotheses and this supports a good generality of the module. However, the current implementation of leaf dynamics in TROLL 4.0 leads to an underestimation of the flux of litterfall in wet and dry seasons and, as a result, of total annual litterfall at both sites. In TROLL 4.0, leaf lifespan was parameterized based on an empirical relationships with leaf structure (leaf mass per area; Maréchaux et al., companion paper). Previous relationships provided in the literature (Reich et al., 1991; Reich et al., 1997; Wright et al., 2004) provided contrasting leaf lifespan estimates, with the one implemented in TROLL 4.0 providing among the highest values, calling for a more in-depth exploration of the reliability and transferability of these empirical relationship. Alternative representations, such as the ones based on optimality principles (Kikuzawa 1991, Franklin et al., 2020, Manzoni et al., 2015), and their combination with the environmentally-driven old leaf shedding acceleration implemented in the new module could be explored in the future.

The evaluation of leaf area index (LAI) and its dynamics was difficult due to the number of products that yield inconsistent time series. Remotely sensed MODIS LAI showed a very small seasonal variation with a slight increase of LAI starting at the beginning of the dry season at both sites. However, MODIS LAI data products are known to be susceptible to the uncertainty affecting the bidirectional reflectance, and to saturate at high LAI values (Petri and Galvão, 2019). Local measurements of LAI through UAV-borne lidar in Paracou showed a stronger increase of total LAI of 0.5 $m^2$ $m^{-2}$ starting at the beginning of the dry season, and leading to a maximum in the dry season. This pattern of variation was in strong agreement with that simulated for LAI by TROLL 4.0. Similarly, local measurements of top canopy LAI derived from phenological cameras in Tapajos (Wu et al., 2016) also showed a high increase of total LAI in the dry season, above 0.5 $m^2$ $m^{-2}$, also in good agreement with the seasonal LAI variation simulated by TROLL 4.0 at that site. By contrast, the LAI derived from terrestrial vertical lidar in Tapajos showed almost no variations (Smith et al., 2019), and such differences with both the patterns derived from phenological cameras and simulations need to be further scrutinised. Among potential explanations, LAI from TLS in Tapajos was adjusted to the annual mean of 5.7 (Stark et al., 2012), leading to lower absolute variations than what was obtained elsewhere, and used coarse spatial and temporal resolutions over small spatial and temporal extents (see material and methods). The discrepancy with simulated patterns could also be linked to uncertainties in LAI variations in the understory in our simulations. Recent studies have suggested opposite variations in LAI between the canopy and the understorey (Nunes et al., 2022), which should be further explored with TROLL 4.0. Overall, while obtaining a robust estimate of LAI temporal variation in tropical forests remains a challenge (Vincent et al., 2023; Bai et al., 2023), the relative variation of LAI simulated by TROLL 4.0 matched the most reliable products at each site, providing an encouraging assessment of this model's ability. Importantly, while total LAI variation remains limited on average within a year in tropical rainforests, this hides important turnover across leaf ages and species, and to ensure robust predictions models should endeavour to represent such turnover and its underlying processes (Wu et al., 2017).

The dry-season increase in total LAI simulated in TROLL 4.0 corresponds to a rejuvenation of the canopy leaf cover associated with a decrease in the LAI of old leaves at the beginning of the dry season, directly followed by an increase in the LAI of

young leaves during the dry season. This turnover is in very good agreement with the one captured by phenological cameras at Tapajos (Wu et al., 2016) and documented in other studies (Yang et al., 2021; Doughty and Goulden, 2008), while the SIF-derived young LAI pattern (Yang et al., 2023) showed an opposite pattern at this site. The main difference in simulated cohorts between the two sites is the continuous dominance of old LAI in Tapajos while mature leaves dominated at the end of the dry season in Paracou. This dominance of older (and less efficient) leaves in Tapajos simulations may be linked to the underestimated litterfall flux and soil water depletion during the dry season at this site. However, the relative proportion of leaf area across the different leaf age pools within and across datasets strongly depends on the definition of the leaf age pools themselves. These pools depend on the individual leaf lifespan in TROLL 4.0 (see section 2.6.2 in Maréchaux et al., submitted companion paper), while the transition from young to mature and mature to old are respectively fixed to 1.71 and 5.14 months in Yang et al. (2023) and fitted to 1 and 3 months in Wu et al. (2016). These contrasting approaches may explain the higher relative importance of old leaves in Wu et al. (2016) compared to Yang et al. (2023) and the intermediate values of TROLL 4.0 (Fig. 9). The seasonal dynamics of leaf cohorts remains poorly known in tropical forests and additional high-resolution optical imagery, *e.g.* by drones or phenological cameras, would be extremely useful to better document these patterns.

## 4.3 Water and carbon fluxes

At Tapajos, DGVMs simulated opposite seasonal trends in carbon and water fluxes compared to the observed ones (e.g. Fig. 1 in Chen et al., 2020; Fig. 5 in Longo et al., 2019b; Fig. 3 in Restrepo-Coupe et al., 2017). In contrast, TROLL 4.0 showed a good ability to represent the dynamics of both carbon and water fluxes estimated with eddy covariance data. In particular, TROLL 4.0 captures the dry season increase in gross primary productivity (GPP) and evapotranspiration (ET) documented for light-limited forests (Guan et al. 2017, Wagner et al. 2016, Aguilos et al. 2018). Simulated GPP and ET also presented realistic daily responses to environmental drivers, namely vapour pressure deficit (VPD), temperature, incident radiation and wind speed, both in direction and relative magnitude.

However, at Tapajos, we found that TROLL 4.0 overestimated ET during the dry season in comparison to eddy flux-derived ET values, under high irradiance, high VPD and high temperature. Simulated ET consists in tree transpiration summed over simulated individuals, water evaporation from the topsoil layer, and the direct evaporation of the rainfall intercepted by the canopy (Kunert et al., 2017). TROLL 4.0 may underestimate the stomatal control of transpiration during the dry season at Tapajos. Accordingly, the control of ET by atmospheric conditions in Tapajos was overestimated in simulated data in comparison to observations, suggesting a stronger coupling of vegetation and atmosphere at that site than simulated (de Kauwe et al., 2017). Underestimation of stomatal control can result from the representation of stomatal conductance and its responses to atmospheric dryness and soil water availability. In particular, the use of daily leaf predawn water potential to control leaf-level gas exchange, and not hourly variation of leaf water water potential (see equ. 39 and 40 in Maréchaux et al. submitted companion paper) can explain the overestimated ecosystem-level fluxes during the dry season. More generally, the understanding of leaf- to ecosystem-level fluxes are active areas of research and alternative representations could be considered in the future as availability of data increases (Wolf et al. 2016; Anderegg et al. 2018; Sabot et al., 2022, Lamour et al., 2022;

see sections 2.5.2 and 2.5.3 and 4.1, and Appendix B in Maréchaux et al. submitted companion paper). Alternatively and/or concurrently, during the dry season, a lack of stomatal control can be due to an overestimation of soil water availability in the model. Soil water content dynamics depend on both the soil depth (Fig. A11) and on the soil hydraulic properties. The two sites are known to present heterogeneity in soil properties but we here performed simulations with homogenous soil properties, both horizontally and vertically. For instance in Paracou, the topsoil layer is sandier than the 15-30 cm layer (Van Langenhove et al., 2021). Although TROLL 4.0 quantitatively captures the soil water depletion observed during the dry season, it appears to underestimate this depletion compared to empirical estimates at both sites (Fig. A10). This underestimation occurs in spite of the agreement between simulated and eddy covariance-derived ET during the dry season in Paracou, and of the higher simulated than eddy-covariance-derived ET during the dry season at Tapajos. Testing the model's sensitivity to soil layer thickness and properties will be important to perform prior to forest projections under drier future conditions and model spatial up-scaling (Meunier et al., 2022). For example, simulations with the ED2 model suggested that forest responses to drier conditions at Tapajos strongly depended on soil texture (Longo et al., 2018). Overall, it would be valuable to evaluate the model under drier conditions than the natural climate variation at the two sites we focused on in this study, such as under throughfall exclusion experiment (Powell et al. 2013; Yao et al. 2022). This would allow us to tease out potential model limitations and further test its forecasting capacity, and we hope to address this in a future contribution. Finally, the greater disagreement between simulated and eddy-covariance-derived ET at Tapajos than Paracou also calls for an in-depth evaluation of the global reanalysis precipitation data at this site. More generally, climate of the Amazon is notoriously challenging for models and it is important to further explore climate forcings in vegetation models.

TROLL 4.0 tended to overestimate empirical GPP estimates, particularly during the dry season, in comparison to both eddy covariance- and SIF-derived GPP. GPP is driven by the photosynthetic activity of the canopy, which depends on multiple processes (Diao et al., 2023; Slot et al., 2024) and further work would be needed to discriminate among them, while accounting for eddy covariance uncertainties (Cui and Chui, 2019). Absorbed light typically has an overriding effect on the variation of GPP across seasons in these light-limited rainforests (Yang et al. 2022, Guan et al. 2015), and simulated GPP is sensitive to the parameters that control light transmission and absorbance (light extinction coefficient, apparent quantum yield; Maréchaux & Chave, 2017). Both are assumed fixed and constant in simulations, but are known to vary with leaf angle distribution and leaf optical properties, depending on micro-environmental conditions and species (Long et al., 1993; Poorter et al., 1995; Meir et al., 2000; Kitajima et al., 2005). In addition, after removing the effect of absorbed light, simulated GPP showed less limitation to high values of VPD and temperature compared to eddy flux- or SIF-derived estimates. The response of leaf-level gas exchanges to the joint effect of atmospheric dryness and soil water availability shows no clear consensus across models (Powell et al, 2013; Trugman et al., 2018), and could be underestimated during the dry season in TROLL 4.0 simulations as discussed above for transpiration. Simulated GPP was higher than inferred from eddy covariance data, which was itself higher than GPP inferred from SIF satellite data (Chen et al., 2022). The eddy covariance-derived GPP were obtained from the net ecosystem exchanges using the nighttime partitioning method (Reichstein et al., 2005). This method was developed for temperate forests with greater temperature variations than tropical forests, which could therefore bias the empirical estimates.

In addition, the eddy flux method has long been reported to underestimate $CO_2$ fluxes (Baldocchi, 2003; Gao et al., 2019). Similarly, even though solar induced fluorescence offers a great potential for the evaluation or the calibration of seasonal carbon fluxes in vegetation models, especially as the tropics are underrepresented by eddy flux tower networks (Villarreal et Vargas, 2021), current SIF products should be used with care (Marrs et al., 2020).

## 5 Conclusions

Here we evaluated the TROLL 4.0 individual-based forest dynamics model, which is capable of jointly simulating forest structure, diversity, dynamics and functioning. To this end, we assembled data from forest inventories, eddy flux towers, litterfall traps, UAV-borne and terrestrial lidar, phenological cameras, and satellite products at two Amazonian forest sites and found that TROLL 4.0 was able to realistically simulate the forest structure, composition, and dynamics, water and carbon fluxes, and leaf area dynamics. In using data of different nature and under the control of different processes, we limited the emergence of equi-finality issues (Medlyn et al., 2005), suggesting a good transferability and robustness of TROLL 4.0.

Comparison with field inventories, aerial and satellite data confirm TROLL 4.0's ability to realistically simulate the structure, composition, and dynamics of tropical forests, without imposing constraints beyond the species pool and calibrating more than three parameters. Discrepancies between observed and simulated number of stems in small size classes and abundance of trait values specific to colonising species suggest further developments of regeneration processes are needed, a worthy endeavour in the context of increased disturbance regimes. TROLL 4.0 was further able to simultaneously simulate the seasonality of productivity, evapotranspiration and leaf area in these two light-limited forests, as opposed to many current DGVMs (Chen et al., 2020; Restrepo-Coupe et al., 2017; Longo et al., 2019). The model's ability to simulate ecosystem fluxes is further shown by the responses of carbon and water fluxes to environmental drivers, whose direction and relative importance were well aligned with observations at both sites despite contrasting climate and soil properties. Additionally, the dynamics of total leaf area appeared realistically partitioned into different leaf pools, as shown by the leaf rejuvenation during the dry season in these systems (Wu et al., 2016; Yang et al., 2021). However, further inspection of the leaf area dynamics across the canopy vertical profile would be useful. Also, the model overestimation of productivity and evapotranspiration during the dry season calls for a more in-depth exploration of the model representation of respiration, plant hydraulics (e.g., stomatal control), and soil hydrology.

Overall, our analyses establish the suitability of TROLL 4.0 for simulating forest structure, diversity, dynamics and ecosystem functioning in short- and long-term studies of tropical forest dynamics, paving the way for multiple applications (Maréchaux et al., 2021). TROLL 4.0 could thus be used for projections of the effects of climate change on tropical forests, and exploration of the effect of biodiversity on forest resilience to these changes (Sakschewski et al., 2016). Similarly, as TROLL 4.0 retains the species-level taxonomic description, it can also help explore the effects of management practices such as timber production, for which half of tropical forests are designated (Blaser et al., 2011). While the development of TROLL 4.0 will continue, in

light of knowledge improvement, novel data collection and identification of uncertainties and discrepancies, we believe it represents a valuable tool for addressing the major challenges tropical forests are currently facing.

## Code and data availability

The TROLL version 4.0 and further developments are publicly available on GitHub as a C++ standalone at https://github.com/TROLL-code/TROLL (Maréchaux et al., 2024) or wrapped into an R package at https://github.com/sylvainschmitt/rcontroll/ (Schmitt et al., 2024). All the code associated with the analyses described in this paper are available at https://github.com/sylvainschmitt/troll_eval (Schmitt, 2024) with corresponding analyses notebook at https://sylvainschmitt.github.io/troll_eval/. Inventories data for Paracou trees over 10 cm are available through request on the CIRAD dataverse: https://dataverse.cirad.fr/dataverse/paracou. Paracou trees understory trees are available through request, PI: GS, GD, JC. Aerial Lidar Scanning from Paracou are available through request (PI: GV) and from dos-Santos et al. (2019) for Tapajos. Species data are available from Jucker et al., (2022), Maréchaux et al., (2015), Guillemot et al., (2022), Vleminckx et al., (2021), Maréchaux et al., (2019), Schmitt and Boisseaux (2023), Boisseaux et al., (submitted), Ziegler et al., (2019), Baraloto et al., (2010), and from TRY (Kattge, Bönisch, et al., 2020). Soil data have been collected from Van Langenhove et al., (2021), Silver et al., (2000), Quesada et al., (2010), Sabatier et al., (1997), and Nepstad et al., (2002). Eddy covariance data from Paracou and Tapajos sites are available on FLUXNET at https://fluxnet.fluxdata.org (last access: 6 September 2023). ERA5-Land data are available on the Climate Data Store: https://cds.climate.copernicus.eu/cdsapp#!/dataset/reanalysis-era5-land?tab=overview. TROPOMI SIF satellite data are available in Chen et al., (2022). Litterfall data at Tapajos are available online through the Oak Ridge National Laboratory (ORNL) Distributed Active Archive Center (DAAC): https://daac.ornl.gov/LBA/guides/CD10_Litter_Tapajos.html and upon-request at Paracou, PI: DB. MODIS LAI data are available online and were extracted from PLUMBER2 on Research Data Australia: https://researchdata.edu.au/plumber2-forcing-evaluation-surface-models/1656048. Terrestrial LAD data from Tapajos are available in Smith et al., (2019). Lidar PAD data from Paracou are available upon-request, PIs: NB and GV. LAI variations among young, mature and leaf cohorts are available from the reanalysis of Yang et al. (2023) at: https://figshare.com/articles/dataset/Leaf_age-dependent_LAI_seasonality_product_Lad-LAI_over_tropical_and_subtropical_evergreen_broadleaved_forests/21700955/4 and from the phenological camera of Wu et al., (2016) at: https://datadryad.org/stash/dataset/doi:10.5061/dryad.8fb47. Tapajos soil moisture data from Restrepo-Coupe et al. (2024) are available at: https://datadryad.org/stash/dataset/doi:10.5061/dryad.d51c5b08g.

## Supplement

The supplement related to this article is available below.

## Author contributions

SS and IM designed the model assessment and carried out the TROLL 4.0 simulations. SS, FJF, JC and IM developed TROLL 4.0. SS, FJF, NB, MB, DB, BB, XC, GD, JL, NRC, ScS, GS, PV, GV, CZ, JC, IM contributed to the data collection and compilation. SS and IM wrote the paper. All authors contributed to the writing.

## Competing interests

The authors declare that they have no conflict of interest.

## Acknowledgements

We are particularly grateful to all the ground workers and data collectors (forest inventories, eddy flux measurements, litter traps, lidar acquisition, sampling and measurement of functional traits, and more) who are not named here but who contributed to the vast knowledge base without which the evaluation of TROLL 4.0 would have been impossible. We are grateful to the GenoToul bioinformatics facility (Castanet-Tolosan, Toulouse, Occitanie, France, doi:10.15454/1.5572369328961167E12) for providing computing resources.

## Financial support

This research has been supported by fundings from ANR (the French National Research Agency) under the "Investissements d'avenir" program with the references ANR-16-IDEX-0006, ANR-10-LABX-25-01, ANR-10-LABX-0041, the Amazonian Landscapes in Transition ANR project (ALT), CNES Biomass-Valo project, and ESA CCI-BIOMASS.

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

**Appendix**
**Table A1: TROLL 4.0 global parameters.**

| Abbreviation | Definition | Units | Value | Nature* | Reference |
|---|---|---|---|---|---|
| $c_a$ | Carbon free air concentration | $\mu$mol mol$^{-1}$ | 375 | Constant | |
| $P_{ress}$ | Atmospheric pressure | kPa | 101 | Constant | |
| $k_{geom}$ | Light extinction coefficient, reflecting leaf geometric arrangement | unitless | 0.5 | Constant | Ross 1981 |
| absorptance$_{leaves}$ | leaves absorptance | unitless | 0.83 | Literature | Long et al., 1993; Poorter et al., 1995 |
| $\theta$ | Curvature factor (Farquhar model parameter) | unitless | 0.7 | Literature | Farquhar et al., 1980 |

| | | | | | | |
|---|---|---|---|---|---|---|
| $g_0$ | leaf minimum conductance for water vapor | mmol H$_2$0 m$^{-2}$ s$^{-1}$ | 5 | Literature | Duursma et al., 2019 |
| $a_{T,o}$ | Phenological parameter that modulates old leaf drought tolerance | unitless | | Calibrated | |
| $b_{T,o}$ | Phenological parameter that modulates the height dependence of leaf susceptibility to drought | MPa | | Calibrated | |
| $\delta_o$ | Phenological parameter that controls the pace of old leaf shedding acceleration | unitless | | Calibrated | |
| $f_{wood}$ | Fraction of carbon allocated to wood | unitless | 0.35 | Literature | Aragão et al., 2019; Malhi et al., 2011 |
| $f_{canopy}$ | Fraction of carbon allocated to canopy | | 0.25 | Literature | Aragão et al., 2019; Malhi et al., 2011 |
| $f_{gap}$ | Fraction of gaps in the tree crown | | 0.15 | Literature | Fischer et al., 2019 |
| a$_{CR}$ | Crown radius intercept | unitless | | Calibrated | |
| b$_{CR}$ | Crown radius slope | unitless | | Calibrated | |
| a$_{CD}$ | Crown depth intercept | m | 0 | Literature | Chave et al., 2005 |
| b$_{CD}$ | Crown depth slope | unitless | 0.2 | Literature | Chave et al., 2005 |
| $shape_{crown}$ | Crown shape parameter | | 0.72 | Calibrated | |
| N$_{tot}$ | Intensity of the external seed rain | seeds ha$^{-1}$ | 50,000 | Assumed | |
| n$_s$ | Number of reproduction opportunities per mature tree | seeds tree$^{-1}$ | 10 | Assumed | |
| $m$ | Reference background mortality | death year$^{-1}$ | | Calibrated | |
| $v_T$ | Variance of the flexion moment for treefall | | 0.021 | Calibrated | |

| | | | | | | |
|---|---|---|---|---|---|---|
| $\sigma_h$ | Intraspecific variation in height (log scale) | m | | 0.19 | Inferred | Baraloto et al., 2010 |
| $\sigma_{cr}$ | Intraspecific variation in crown radius (log scale) | m | | 0.29 | Calibrated | Fischer et al., 2019 |
| $\sigma_{cd}$ | Intraspecific variation in crown depth (log scale) | m | | 0 | | |
| $\sigma_{dbhthres}$ | Intraspecific variation in maximum diameters (log scale) | m | | 0.05 | Inferred | Baraloto et al., 2010 |
| $corr_{cr-h}$ | Intraspecific correlation between crown radius and height | | | 0 | | |
| $\sigma_P$ | Intraspecific variation in phosphorus (log scale) | g g$^{-1}$ | | 0.24 | Inferred | Baraloto et al., 2010 |
| $\sigma_N$ | Intraspecific variation in nitrogen (log scale) | g g$^{-1}$ | | 0.12 | Inferred | Baraloto et al., 2010 |
| $\sigma_{LMA}$ | Intraspecific variation in leaf mass per area (log scale) | g m$^{-2}$ | | 0.24 | Inferred | Baraloto et al., 2010 |
| $\sigma_{wsg}$ | Intraspecific variation in wood specific gravity | g cm$^{-3}$ | | 0.06 | Inferred | Baraloto et al., 2010 |
| $\sigma_{LA}$ | Intraspecific variation in leaf area (log scale) | cm$^2$ | | 0.48 | Inferred | Schmitt and Boisseaux 2023 |
| $\sigma_{tlp}$ | Intraspecific variation in turgor loss point (log scale) | MPa | | 0.10 | Inferred | Schmitt and Boisseaux 2023 |
| $corr_{N-P}$ | Intraspecific correlation between nitrogen and phosphorous | | | 0.65 | Inferred | Baraloto et al., 2010 |
| $corr_{N-LMA}$ | Intraspecific correlation between nitrogen and leaf mass per area | | | -0.43 | Inferred | Baraloto et al., 2010 |
| $corr_{P-LMA}$ | Intraspecific correlation between phosphorus and leaf mass per area | | | -0.39 | Inferred | Baraloto et al., 2010 |

*Assumed is a value that is supposed; Calibrated is a value that was previously calibrated; Constant is a physical constant; Inferred is a value
that has been derived from an existing dataset; Literature is a value prescribed from the literature.

**Table A2: Evaluation of forest structure, composition and fluxes explored at Paracou and Tapajos. Evaluations include the goodness-of-fit $R^2$ from the linear regression with a null intercept, the Pearson's r correlation coefficient CC, the root mean square error of prediction RMSEP, the standard deviation of the error of prediction SD.**

| Site | Variable | Unit | Observations | Temporal resolution | $R^2$ | CC | RMSEP | SD |
|------|----------|------|--------------|---------------------|-------|-----|-------|-----|
| Paracou | height | % | Plane | single | 0.93 | 0.95 | 0.76 | 0.76 |
| Tapajos | height | % | Plane | single | 0.94 | 0.94 | 0.56 | 0.55 |
| Paracou | height | % | Satellite | single | 0.95 | 0.96 | 0.55 | 0.55 |
| Tapajos | height | % | Satellite | single | 0.92 | 0.91 | 0.69 | 0.62 |
| Paracou | BA understory | $m^2$ $ha^{-1}$ | Inventory | single | 0.94 | 0.90 | 0.12 | 0.08 |
| Paracou | Number of stems in understory | $ha^{-1}$ | Inventory | single | 0.99 | 1.00 | 342.15 | 309.81 |
| Paracou | Rank-abundance | $ha^{-1}$ | Inventory | single | 0.89 | 0.88 | 0.59 | 0.44 |
| Tapajos | Rank-abundance | $ha^{-1}$ | Inventory | single | 0.47 | 0.96 | 1.10 | 0.68 |
| Paracou | GPP | kgC $m^{-2}$ $year^{-1}$ | eddy flux | day | 0.97 | 0.60 | 0.75 | 0.67 |
| Tapajos | GPP | kgC $m^{-2}$ $year^{-1}$ | eddy flux | day | 0.97 | 0.45 | 1.12 | 0.67 |
| Paracou | GPP | kgC $m^{-2}$ $year^{-1}$ | Satellite | day | 0.95 | 0.45 | 1.18 | 0.80 |
| Tapajos | GPP | kgC $m^{-2}$ $year^{-1}$ | Satellite | day | 0.96 | 0.22 | 1.54 | 0.28 |
| Paracou | LAI | $m^2$ $m^{-2}$ | Satellite | 15 days | 1.00 | 0.69 | 0.29 | 0.13 |
| Tapajos | LAI | $m^2$ $m^{-2}$ | Satellite | 15 days | 1.00 | 0.55 | 0.26 | 0.17 |
| Paracou | LAI | $m^2$ $m^{-2}$ | Drone | 15 days | 1.00 | 0.84 | 0.11 | 0.11 |
| Tapajos | LAI | $m^2$ $m^{-2}$ | Terrestrial | 15 days | 1.00 | 0.25 | 0.32 | 0.20 |
| Tapajos | LAI | $m^2$ $m^{-2}$ | Phenocam | 15 days | 1.00 | 0.91 | 0.11 | 0.08 |
| Paracou | ET | mm $day^{-1}$ | eddy flux | day | 0.96 | 0.69 | 0.60 | 0.60 |
| Tapajos | ET | mm $day^{-1}$ | eddy flux | day | 0.96 | 0.75 | 0.75 | 0.63 |
| Paracou | RSWC | % | eddy flux | day | 0.97 | 0.77 | 0.24 | 0.13 |
| Tapajos | RSWC | % | eddy flux | day | 0.99 | 0.39 | 0.20 | 0.11 |

**Table A3: Comparisons of forest structure and phenology parameter values from the five best fits, including minimum, maximum and median values, as well as the one of the best fit. Note that the median of the parameter values of the five best fits always equal the value of the best fit, except for $m$ at Paracou with a small difference of 0.0025 and $\delta_0$ in both sites.**

| Site | Parameter | minimum | median | best | maximum |
|------|-----------|---------|--------|------|---------|
| Paracou | $a_{CR}$ | 1.80 | 1.80 | 1.80 | 1.90 |
| Paracou | $b_{CR}$ | 0.386 | 0.386 | 0.386 | 0.443 |
| Paracou | $m$ | 0.0325 | 0.0325 | 0.0350 | 0.0375 |
| Paracou | $a_{T,0}$ | 0.2 | 0.2 | 0.2 | 0.2 |
| Paracou | $b_{T,0}$ | 0.015 | 0.02 | 0.015 | 0.02 |
| Paracou | $\delta_0$ | 0.1 | 0.4 | 0.1 | 0.5 |
| Tapajos | $a_{CR}$ | 2.35 | 2.45 | 2.45 | 2.50 |
| Tapajos | $b_{CR}$ | 0.6994 | 0.7565 | 0.7565 | 0.7850 |
| Tapajos | $m$ | 0.0300 | 0.0400 | 0.0400 | 0.0500 |
| Tapajos | $a_{T,0}$ | 0.2 | 0.2 | 0.2 | 0.3 |
| Tapajos | $b_{T,0}$ | 0.015 | 0.015 | 0.015 | 0.015 |
| Tapajos | $\delta_0$ | 0.2 | 0.3 | 0.2 | 0.5 |

**Table A4: Calibrated parameters intervals for the 5 best simulations for each of stem distribution, number of stems, basal area and**
**aboveground biomass, as well as the one with equal weighing among them. Values show median first followed by minimum and**
**maximum values in brackets.**

| Site | Metric | RMSEP | $a_{CR}$ | $b_{CR}$ | $m$ |
|---|---|---|---|---|---|
| Paracou | Number of stems | 5.75 [2-7.75] | 1.75 [1.75-1.8] | 0.3575 [0.3575-0.386] | 0.0475 [0.0375-0.05] |
| Paracou | Basal Area | 0.04 [0.03-0.07] | 1.85 [1.65-2] | 0.4715 [0.3505-0.5075] | 0.0325 [0.03-0.05] |
| Paracou | Stem distribution | 2.4 [1.38-2.7] | 1.85 [1.8-1.9] | 0.4145 [0.386-0.443] | 0.0425 [0.0325-0.05] |
| Paracou | All equally weighted | 0.16 [0.13-0.17] | 1.8 [1.8-1.9] | 0.386 [0.386-0.443] | 0.0325 [0.0325-0.0375] |
| Tapajos | Number of stems | 3 [0-3.5] | 2.5 [2.4-2.65] | 0.785 [0.728-0.9205] | 0.035 [0.03-0.04] |
| Tapajos | Aboveground Biomass | 0.13 [0.04-0.19] | 2.45 [2.35-2.5] | 0.835 [0.6495-0.885] | 0.045 [0.03-0.05] |
| Tapajos | Stem distribution | 2.54 [2.38-2.74] | 2.35 [2.3-2.35] | 0.6995 [0.671-0.6995] | 0.045 [0.0375-0.05] |
| Tapajos | All equally weighted | 0.25 [0.18-0.25] | 2.45 [2.35-2.5] | 0.7565 [0.6995-0.785] | 0.04 [0.03-0.05] |


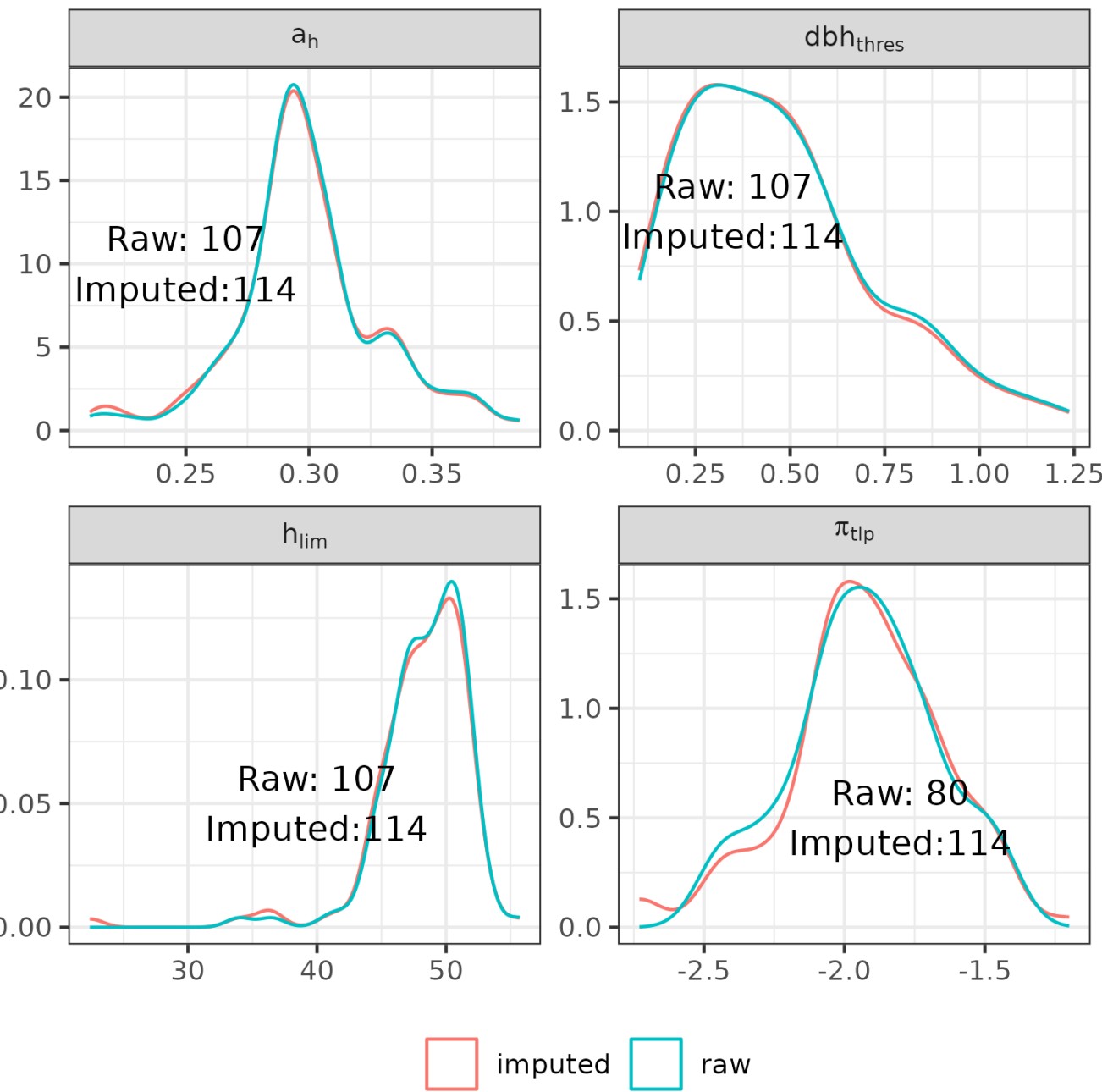

Figure A1: Representativity of imputed functional traits values (red) against raw functional trait values (blue) from various datasets (see methods). Traits were imputed using predictive means matching for $a_h$, $dbh_{thres}$, hlim, and $\pi_{tlp}$ only. The number in each subplots represents the number of species with a trait value in the raw data and after imputation composing respectively the blue and red curves.

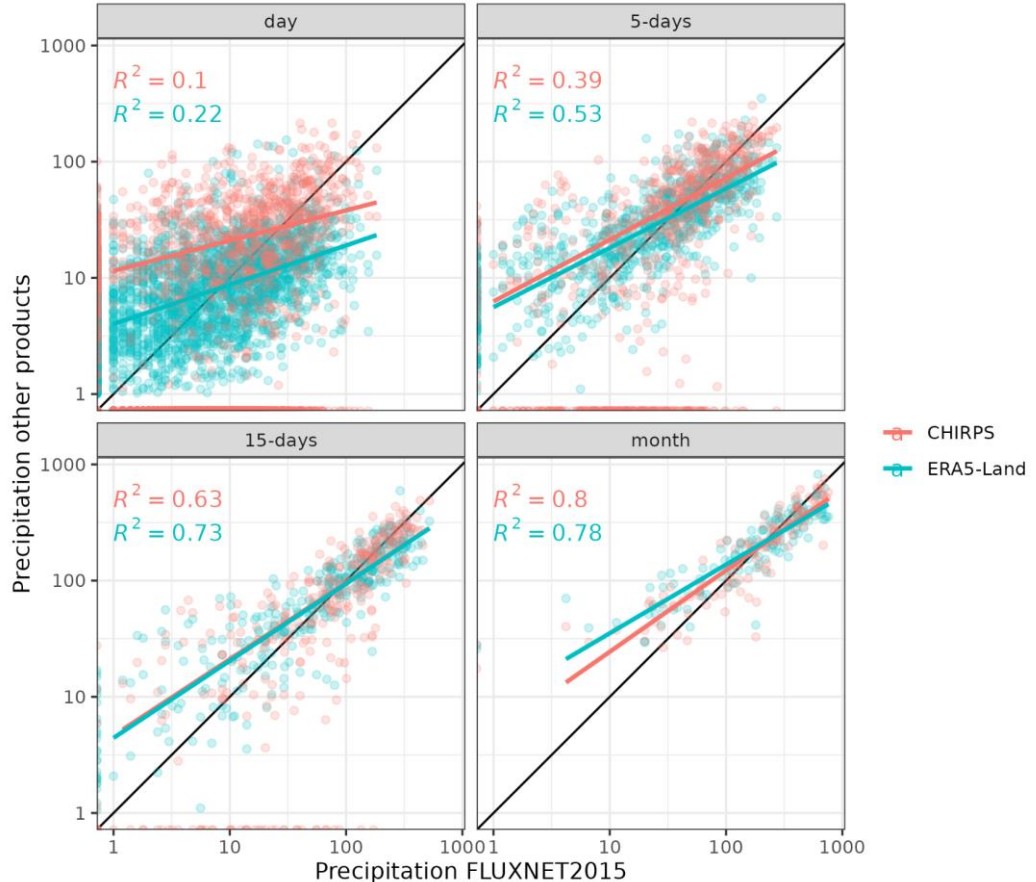

1291

**Figure A2: Comparisons of CHIRPS (red) and ERA5-Land (blue) precipitation products against local eddy-flux tower measurements retrieved from FLUXNET 2015 in Paracou at daily, 5-day, 15-day, and monthly resolutions. CHIRPS and ERA5-Land had similar agreement to locally measured precipitations, with even higher correlations for ERA5-Land than CHIRPS. However, they both overestimated low precipitation events and underestimated high precipitation events, resulting in low agreement for daily variations (R2 of 0.10 and 0.22), which quickly increases for 5-day (R2 of 0.39 and 0.53) and 15-day variations (R2 of 0.63 and 0.73). Although a similar assessment was not possible in Tapajos due to a lack of local reliable rainfall data, we therefore decided to keep ERA5-Land for filling the precipitation data gaps in Tapajos.**

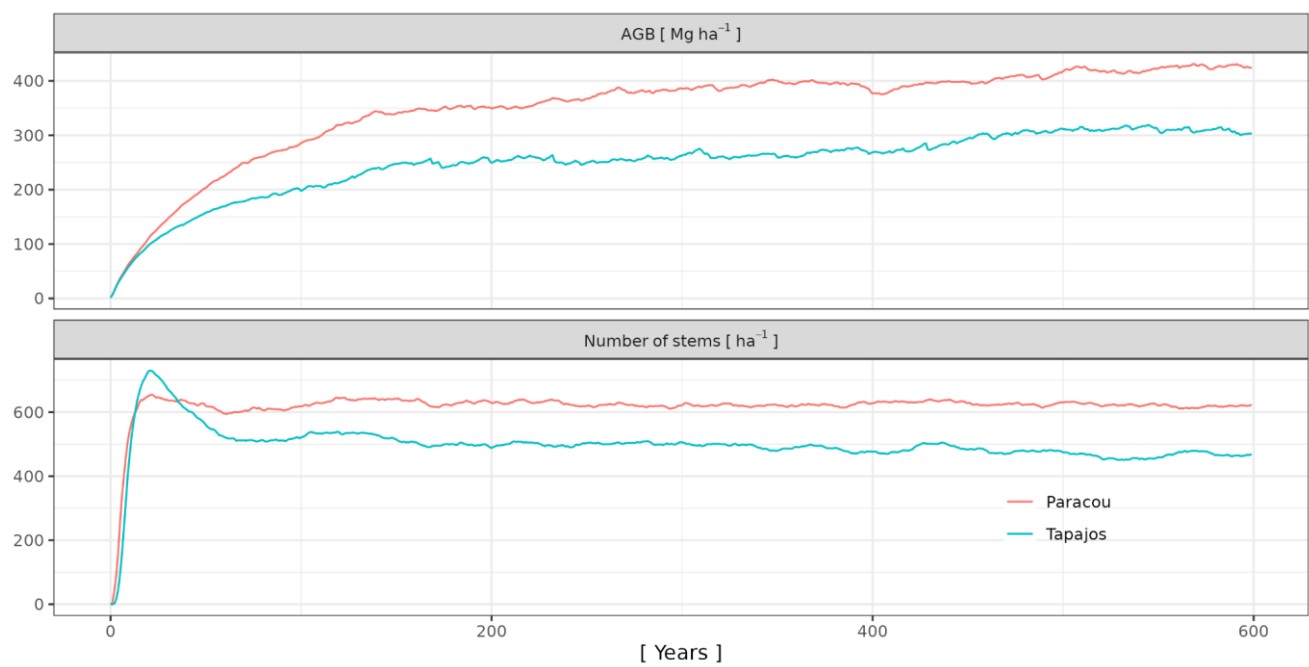

1299

**Figure A3: 600-year spin-up simulations from bareground with calibrated parameters showing equilibrium reached by number of**
**stems (bottom) aboveground biomass (AGB, top) at Paracou (red) and Tapajos (blue).**

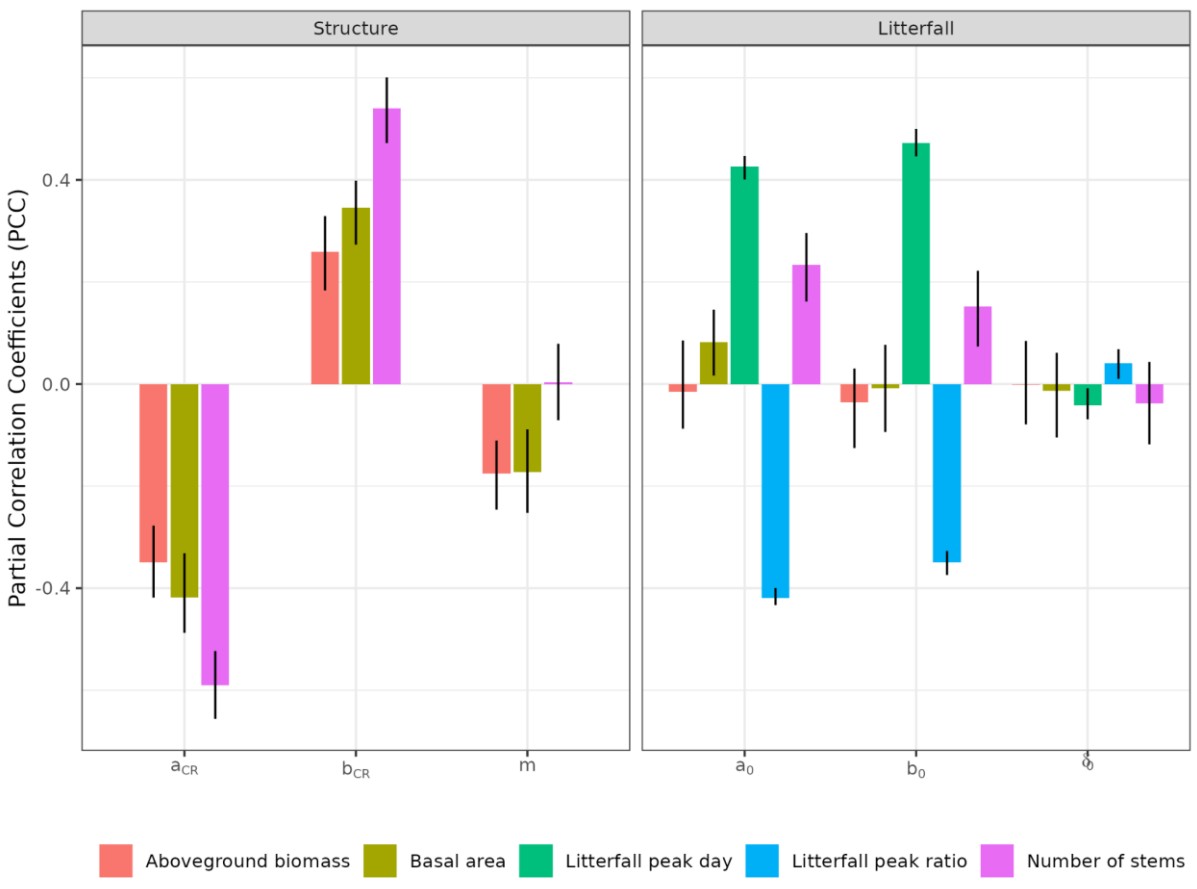

1302

Figure A4: Sensitivity of forest structure (left panel) and forest litterfall (right panel) to calibrated parameters. Forest structure and forest litterfall sensitivity to each parameter was assessed with partial correlation coefficients (PCC) using the function *pcc* of the R package *sensitivity* with a thousand bootstrap draws to assess confidence intervals. The intercept and slope of the crown radius allometry $a_{CR}$ and $b_{CR}$ had a strong effect on forest structure , i.e., number of stems (red), aboveground biomass (AGB, light green), and basal area (BA, green). Basal mortality $m$ also had a strong effect on aboveground biomass (AGB) and basal area (BA) but little to no effect on the number of stems. $a_{T,0}$ and $b_{T,0}$ had a strong positive effect on the peak day of litterfall (blue) and negative effect on the ratio of the peak of litterfall (purple), but $delta_0$ had only a weak effect on the peak of litterfall. The forest structure variables, namely number of stems, aboveground biomass (AGB), and basal area (BA) showed little to no partial correlations to $a_{T,0}$, $b_{T,0}$ and $delta_0$.

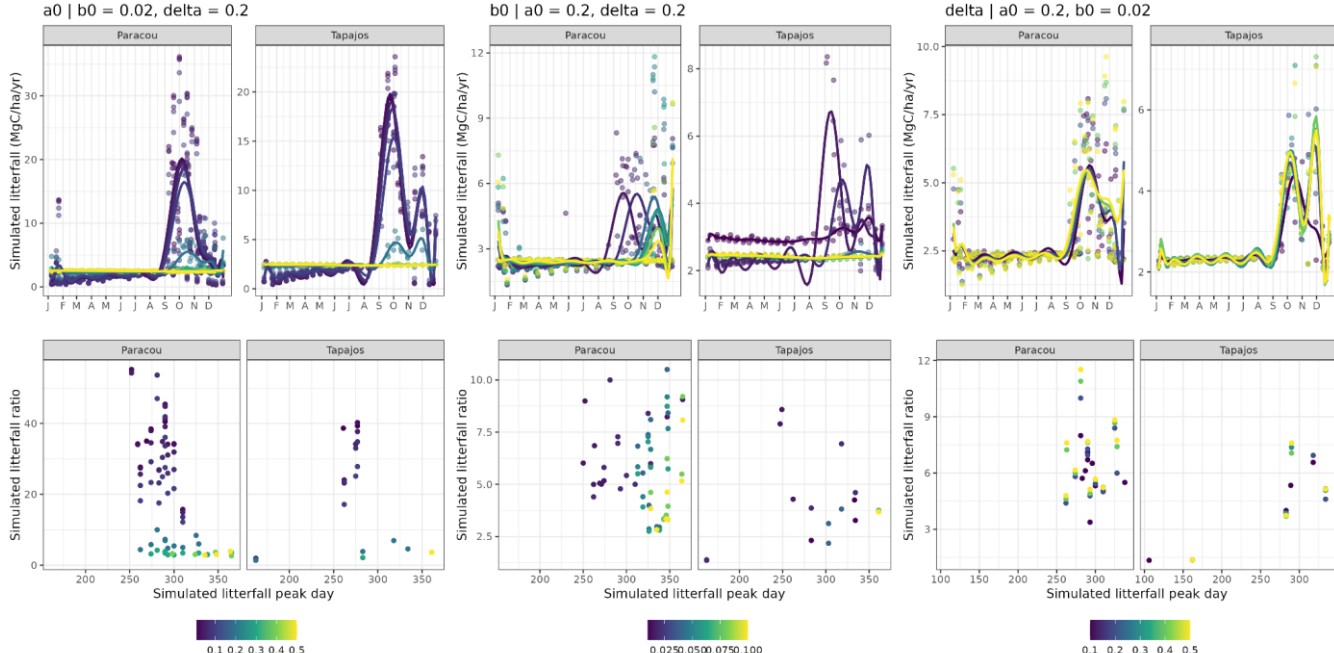

1312

Figure A5. Effect of each parameter of the new leaf shedding module on the simulated timing and intensity of the litterfall peak during the dry season. Top panels illustrate simulated variations of litterfall at both sites for varying $a_{T,0}$, $b_{T,0}$, and $\delta_0$ with the other parameters fixed to a calibrated value. Bottom panels illustrate the corresponding timing and intensity of the dry season litterfall peak: (i) the day of the litterfall peak as the julian day of the maximum annual value (day), and (ii) the ratio between the peak value (computed as the average of litterfall flux over the two consecutive time intervals before and after the peak day) divided by the basal flux (computed as the average between January and April) (ratio). $a_{T,0}$ mainly limited the intensity of the peak with a peak up to 60 times the wet season base litter flux with small parameter values close to 0.01 and no peak with values greater than 0.3, when $b_{T,0}$=0.02 and $\delta_0$ =0.2. Values of $a_{T,0}$ greater than 0.1 also resulted in a later peak during the dry season. $b_{T,0}$ mainly influenced the date of the simulated peak during the dry season, as well as the intensity of the simulated peak for values greater than 0.1. Indeed, low values of $b_{T,0}$ , close to 0.01, resulted in a peak starting in September, while high values showed a peak starting in December, when $a_{T,0}$=0.2 and $\delta_0$=0.2. Finally, $\delta_0$ appeared to have a smaller influence on the intensity and timing of the simulated litter peaks. Higher values of $\delta_0$ increased the duration of the simulated peaks or the litter flux between two peaks during the same dry season.

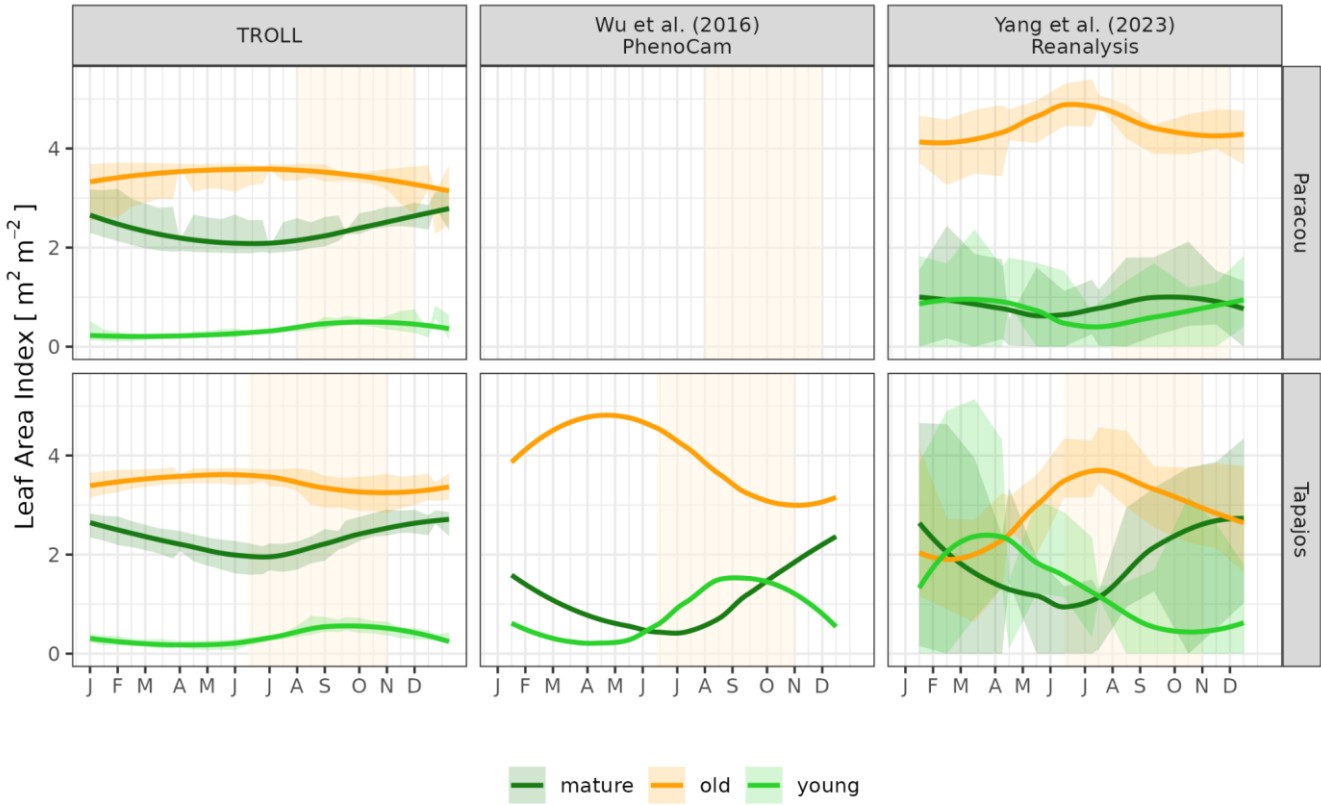

**Figure A6: Mean annual cycle of leaf area index per leaf age cohorts, derived from fortnightly means, at Paracou and Tapajos. Note**
**that the three leaf age cohorts (young, mature and old leaves) are not defined the same way in the three sources. Leaf age per cohort**
**depends on the individual leaf lifespan in TROLL 4.0 (see Maréchaux et al., submitted companion paper), while the transition from**
**young to mature and mature to old are respectively fixed to 1.71 and 5.14 months in Yang et al. (2023) and fitted to 1 and 3 months**
**in Wu et al. (2016). Bands are the intervals of means across years, and the vertical yellow bands in the background correspond to**
**the site's climatological dry season.**


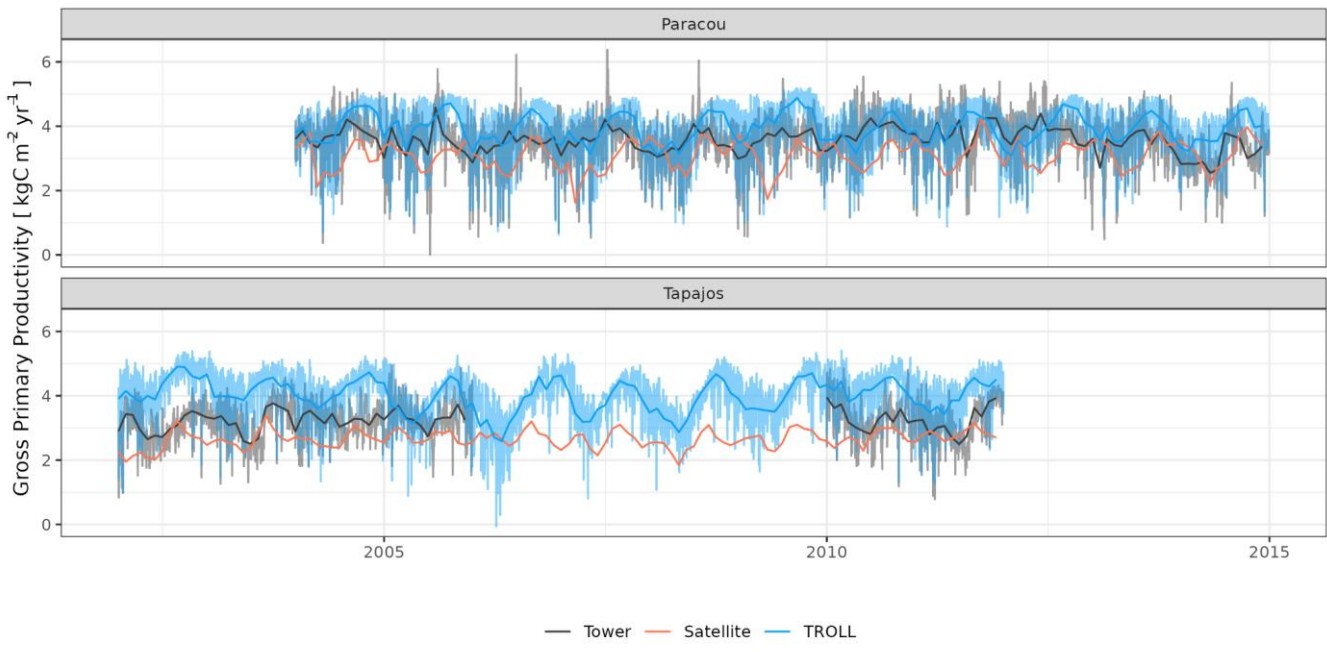


Figure A7: Daily and monthly means of gross primary productivity for Paracou and Tapajos. Dark lines are the monthly means,
semi-transparent lines are the daily means variations with the exception of satellite data for which data are available only every 8
days.


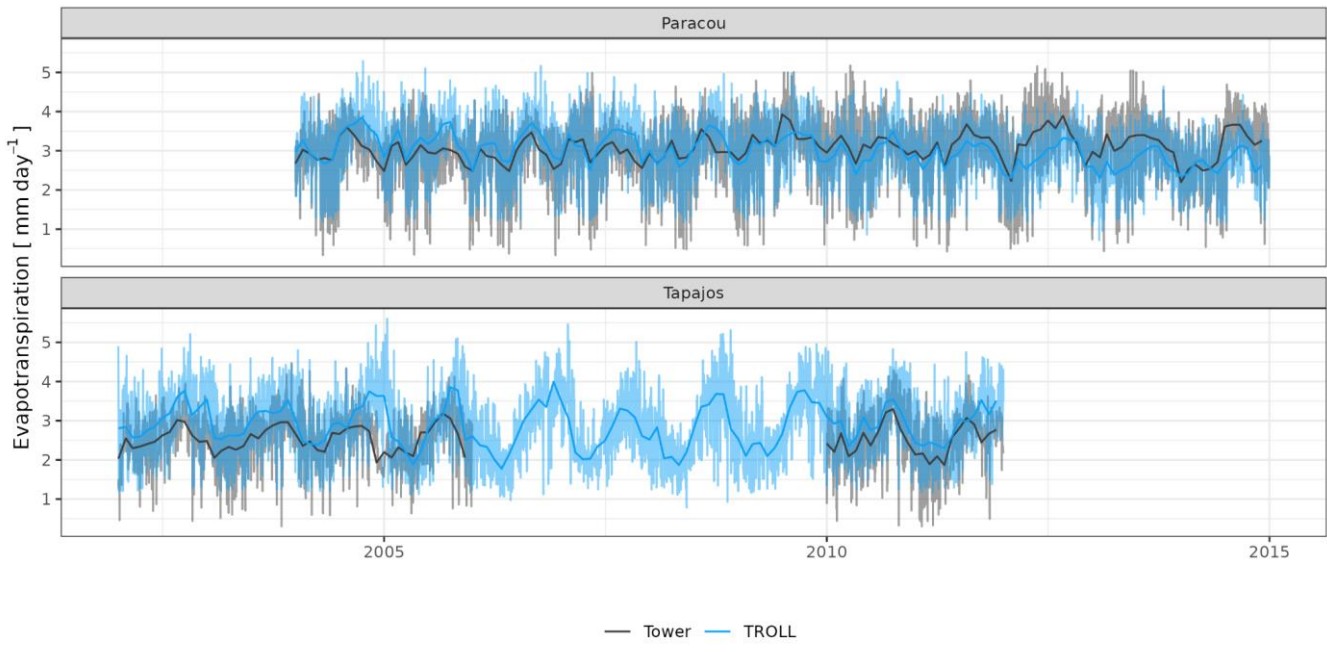


Figure A8: Daily and monthly total of evapotranspiration for Paracou and Tapajos. Dark lines are the monthly means, semi-transparent lines are the daily means variations.

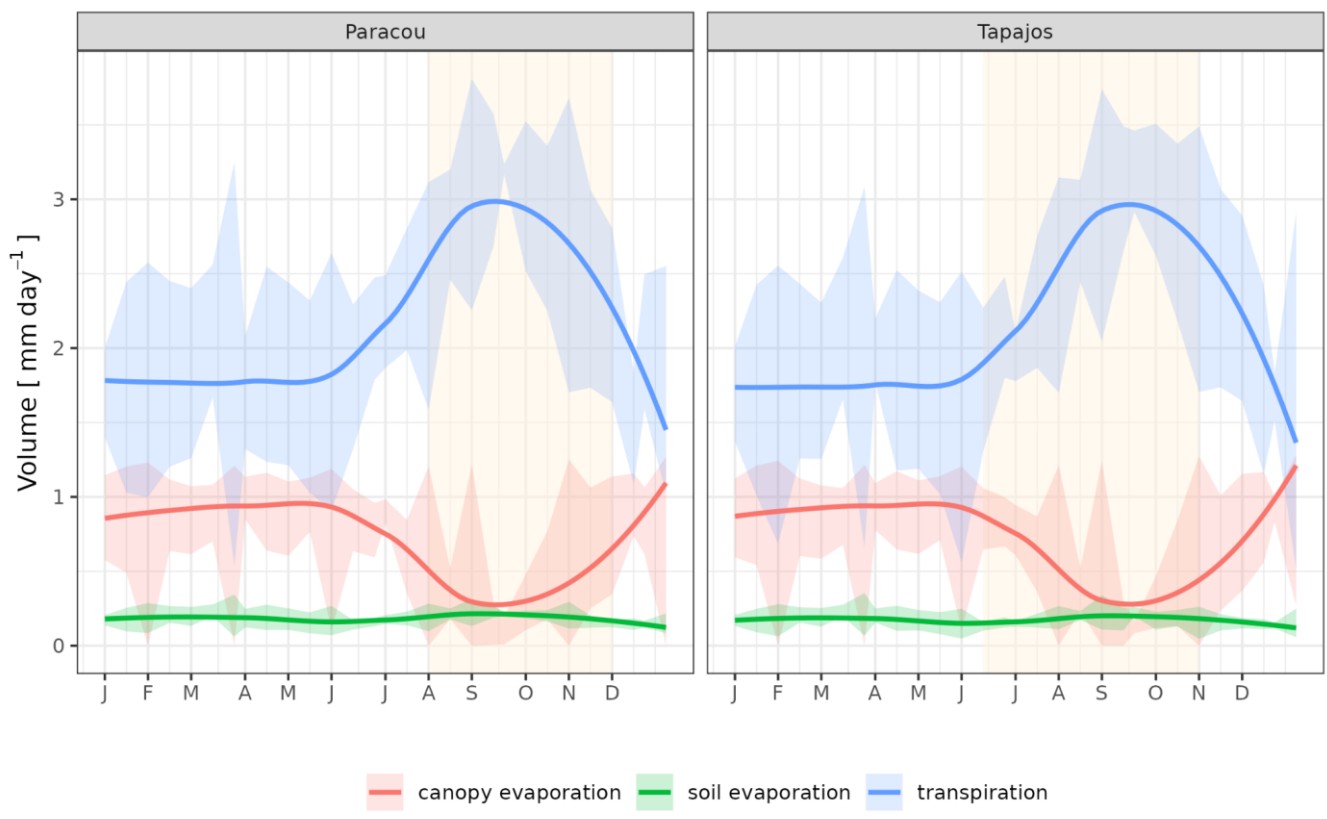


**Figure A09: Mean annual cycle of evapotranspiration partitioning between canopy evaporation (red), soil evaporation (green), and tree transpiration (blue) for Paracou (left) and Tapajos (right), derived from fortnightly means simulated with TROLL 4.0. Bands are the intervals of means across years, and the yellow vertical bands in the background correspond to the site's climatological dry season. Simulated values correspond to 10 years of simulations starting from the end-state of 600-year regeneration from bare ground with calibrated parameters at each site.**

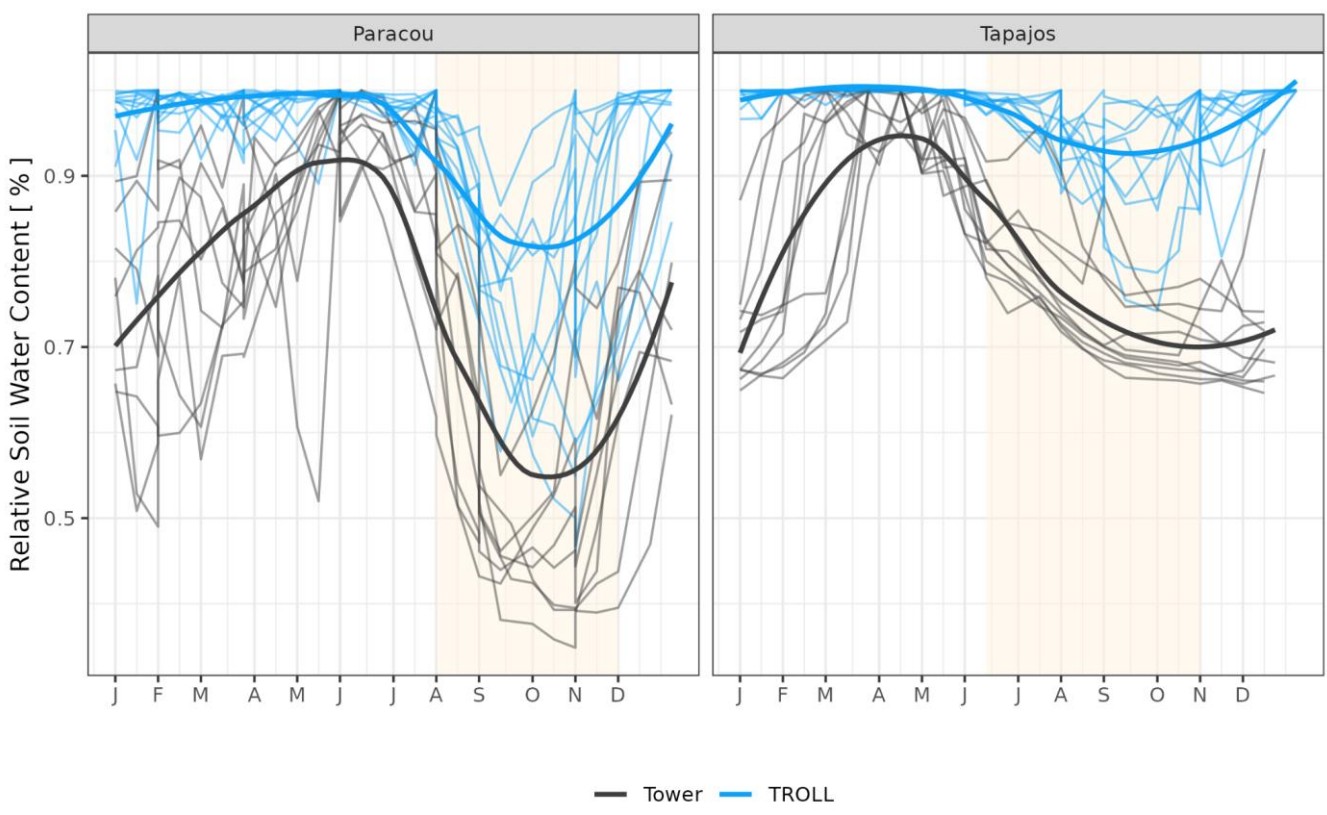

**Figure A10: Mean annual cycle from daily means of relative soil water content for Paracou and Tapajos for the topsoil layer up to**
**10 cm. Dark lines are the daily mean across years, semi-transparent lines are the daily means per year. The vertical yellow bands in**
**the background correspond to the site's climatological dry season.**

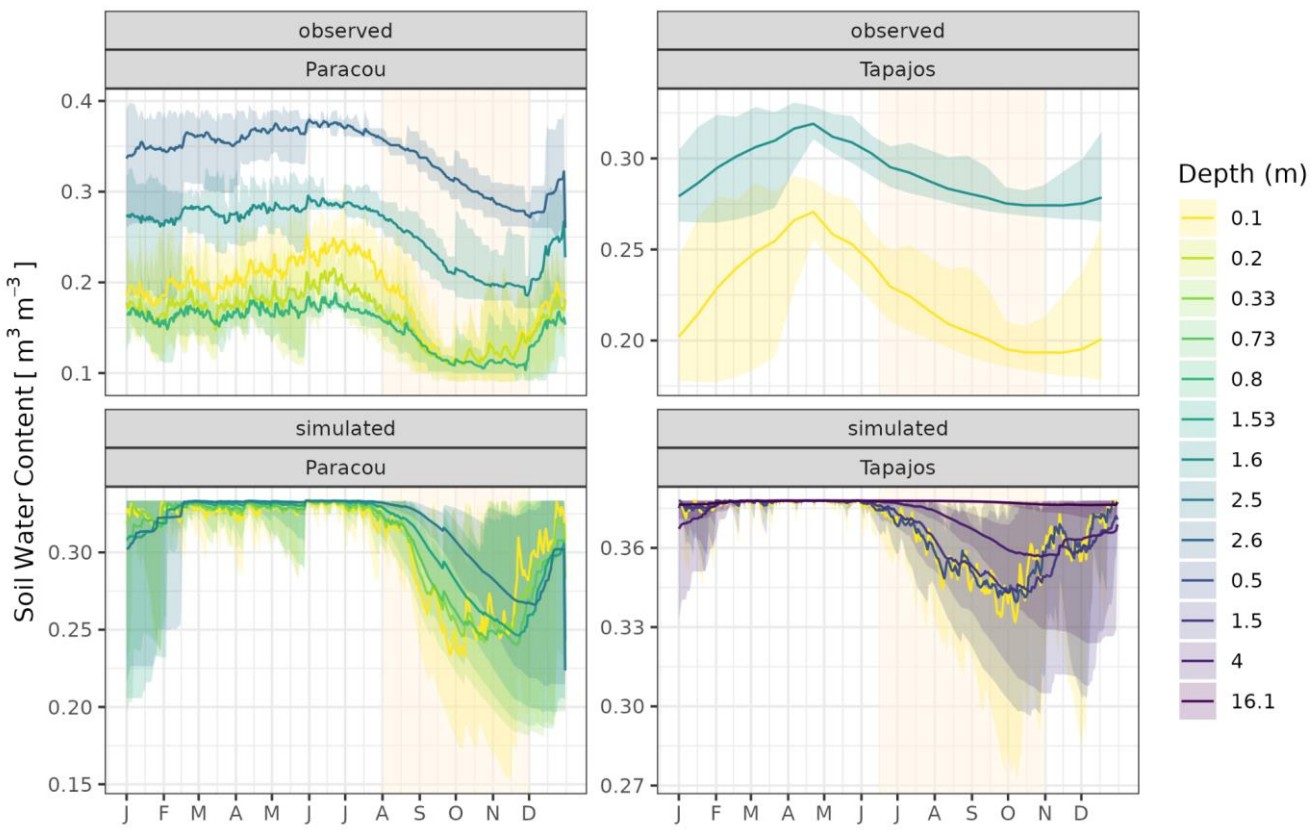


**Figure A11: Mean annual cycle from daily means of soil water content for Paracou and Tapajos at different depths. The depth value**
**indicates the maximum depth of the layer. Dark lines are the daily means across years, and bands are the intervals of means across**
**ten years The vertical yellow bands in the background correspond to the site's climatological dry season.**

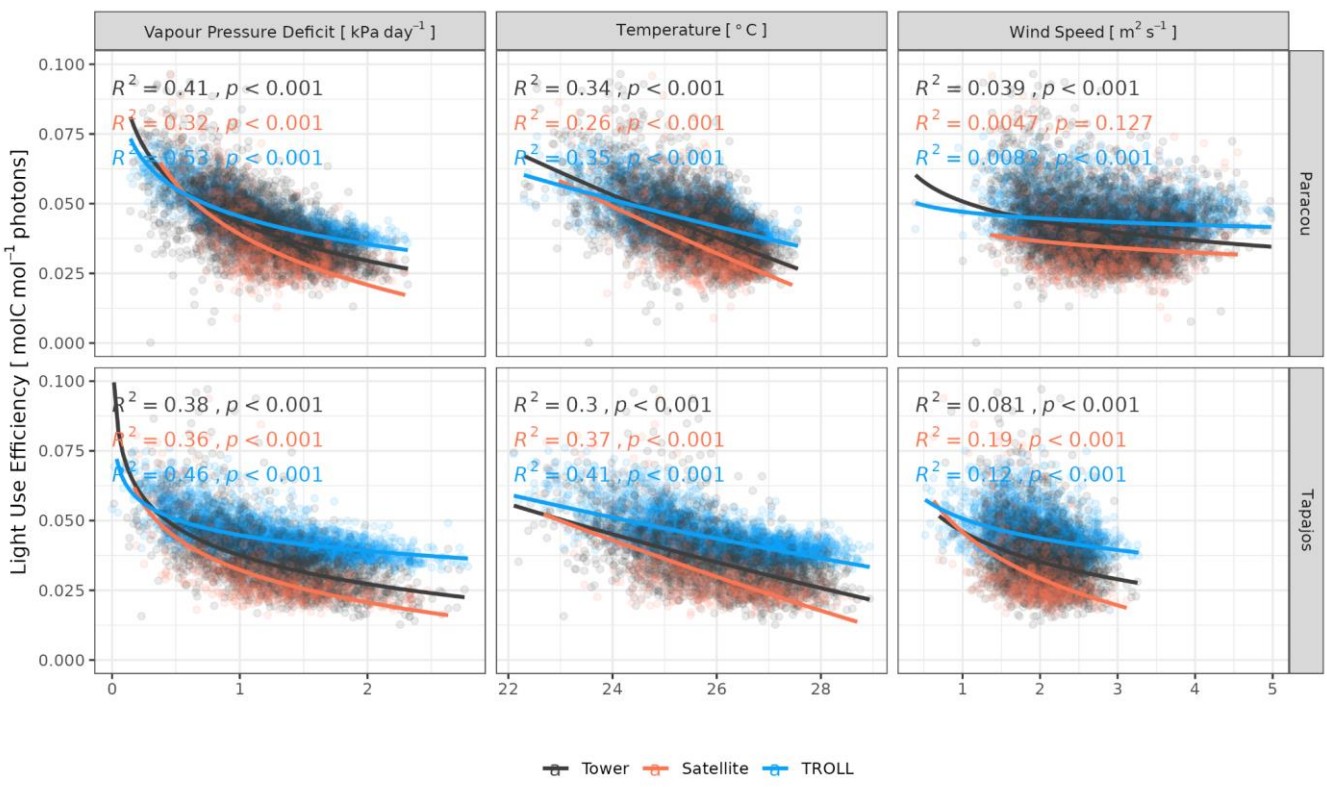


**Figure A12: Daily averages of light use efficiency as a function of daily maximum vapour pressure deficit, average temperature, and average wind speed for model-, satellite- and eddy flux-based estimates at Paracou (top) and Tapajos (bottom). Lines illustrate the linear regression of form y ~ log(x), and text the squared Pearson's R correlation coefficient. The light use efficiency (LUE) was obtained by normalizing gross primary productivity (GPP) by photosynthetic photon flux density (PPFD) and the fraction of absorbed photosynthetically active radiation (fAPAR) itself derived from leaf area index (LAI).**

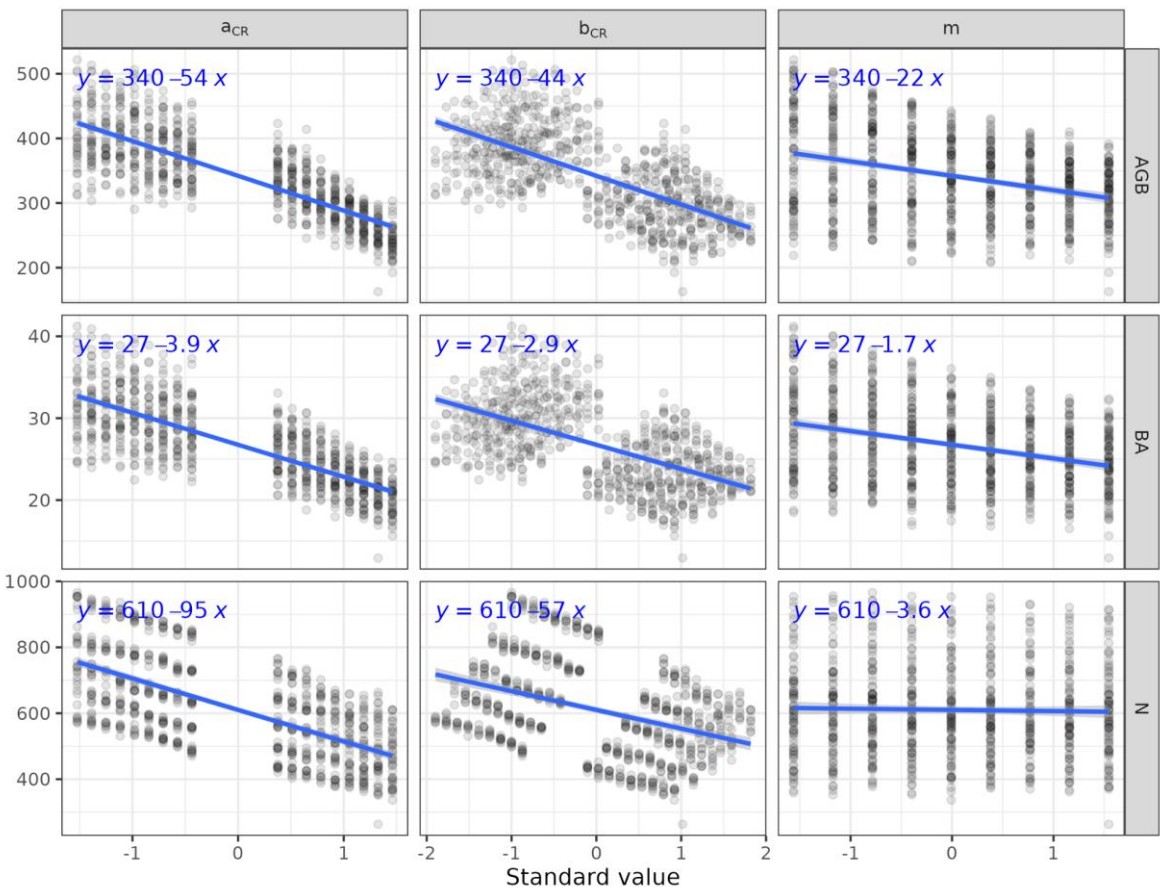

1364

Figure A13: Magnitude of sensitivity of forest structure to calibrated parameters. For each forest structure variable , the magnitude
of sensitivity to each parameter was assessed with regression assuming linear and monotonic relationships. The intercept ($a_{CR}$, left
column) and slope ($b_{CR}$, central column) of the crown radius allometry as well as the basal mortality rate ($m$, right column) had a
strong effect on aboveground biomass (AGB, Mg ha$^{-1}$, top line), basal area (BA, m$^2$ ha$^{-1}$, middle line), and number of stems (N, trees
ha$^{-1}$, bottom line). Parameters on the x-axis are expressed in standard values to ease their comparisons. The blue line and equation
represent the linear regression relating independently each variable to each parameter.

1371