# Peer review of "TROLL 4.0: representing water and carbon fluxes, leaf phenology,"

_EGUsphere, 2024_

## Author Response (AR1)

Dear Editors,

Please find enclosed the revised version of our manuscript entitled "TROLL 4.0: representing water and carbon fluxes, leaf phenology, and intraspecific trait variation in a mixed-species individual-based forest dynamics model – Part 2: Model evaluation for two Amazonian sites". We thank the reviewers for their thorough and constructive comments on our manuscript. We consider that our manuscript has greatly benefited from the meaningful comments from all the reviewers.

Below are our responses to the reviewers' comments, and in red in the author's track-changes file are our changes to the previously submitted version of the manuscript. We hereby confirm that this work has not been published or accepted for publication, and is not under consideration for publication, elsewhere. Submission for publication has been approved by all co-authors and all persons entitled to authorship have been named.

Yours sincerely,

*Sylvain Schmitt, on the behalf of all the co-authors*

**Reviewer 1:**

The author presents the potential TROLL 4 model performance in Amazon forest sites, this manuscript is clear but still needs some improvement before I recommend this to be published.

> We thank you for your thorough review of our manuscript and hope that the improvements we have made following your comments and those of the other reviewers will answer your questions.

Line 103: write typo: each tree belongs to a species

> We updated the sentence as "We assign a species label to each simulated tree and provide as input species-specific mean plant trait values and intraspecific trait variances and covariances." (l. 104-105).

Line 104: what's the meaning of recruitment, some explanations

> Tree recruitment is the process by which a new individual is added to a community. In the TROLL model, recruitment corresponds to the initialisation of a new tree with a diameter at breast height of 0.01m, which occurs in empty cells where there is at least one seed in the seed bank and sufficient ground light and topsoil water availability (see companion paper). We updated the relevant section as follows "New trees appear in the community through the process of tree recruitment, which is only possible in empty cells and with favourable light and water availability. Trees of a given species are recruited if there is at least one seed of that species in the local seed bank. Individual trait values of each recruited tree are randomly drawn from the intraspecific trait distribution." (l. 105-108).

Line 119: why do you select six parameters for calibration? These parameters are sensitive to what? Any sensitivity analysis?

> This manuscript evaluates the ability of TROLL 4.0 to simulate tropical forest functioning, structure and composition at two different Amazonian sites. We sought to minimize the number of site-specific calibrated parameters. In doing so, rather than maximizing the goodness-of-fit with data at each site, we performed a conservative assessment of the model performance and its transferability across sites. The six parameters we selected for calibration at each site are currently poorly informed by available data and/or to which the model is known to be sensitive based on sensitivity analyses performed on previous versions of the model (Maréchaux & Chave 2017; Fischer et al. 2019; Salzet et al. 2024). Values of all other parameters are derived from existing knowledge and data, and common to the two sites.

Specifically, previous sensitivity analyses of the TROLL model version 3 revealed the basal mortality $m$ has an important role in driving forest dynamics, and the intercept and slope of the crown radius allometry $a_{CR}$ and $b_{CR}$ mediate canopy packing. Moreover, novel developments in TROLL 4.0 were based on known or measurable ecological parameters and physical constants, such as leaf minimum conductance for water vapor (Duursma et al., 2019), but the three parameters of the new leaf phenology module $a_{T,0}$, $b_{T,0}$, and delta$_0$ are more empirical and not ecologically measurable. These parameters drive the timing and intensity of leaf litterfall peak during the dry season as shown in supplementary figure A5.

This is now better highlighted in the text:

"To provide a conservative assessment of the model's performance and its transferability across sites, we restricted the number of site-specific calibrated parameters to the ones that are currently poorly informed by available data, or to which the model is known to be sensitive based on sensitivity analyses performed on previous versions of the model (Maréchaux & Chave 2017; Fischer et al. 2019). At each site, we calibrated six parameters. These include three parameters related to forest structure: the reference background mortality rate m, and the intercept aCR and slope bCR of the crown radius scaling relationship (Table A1; Maréchaux and Chave, 2017; Fischer et al., 2019). Novel developments in TROLL 4.0 were based on known or measurable ecological parameters and physical constants but the three parameters of the new leaf phenology module $a_{T,0}$, $b_{T,0}$, and delta$_0$ (Table A1) are more empirical and not ecologically measurable." (l. 126-135).

A complete sensitivity analysis of TROLL 4.0 goes beyond the goals and scope of this first evaluation manuscript, and we leave it for a future contribution. However, to confirm our expert knowledge on chosen parameters, we have now further used the parameters space used for calibration to measure response variable sensitivity to each parameter with partial correlation coefficients (PCC) using the function *pcc* of the R package *sensitivity* with a thousand bootstrap draws to assess confidence intervals. As expected the intercept and slope of the crown radius allometry $a_{CR}$ and $b_{CR}$ had a strong effect on forest structure with a PCC around -0.4 for $a_{CR}$ and around 0.4 for $b_{CR}$ with number of stems, aboveground biomass (AGB), and basal area (BA). The basal mortality $m$ also had a strong effect on aboveground biomass (AGB) and basal area (BA) with a PCC around -0.2 but little to no effect on the number of stems (PCC around 0). Similarly, $a_{T,0}$ and $b_{T,0}$ had a strong positive effect on the peak day of litterfall (PCC close to 0.5) and negative effect on the ratio of the peak of litterfall (PCC around 0.3), but delta$_0$ had only a weak effect on the peak of litterfall. These new

results have been added in the new appendix figure Fig. A4 reported below, and described in material and methods and results:

- "To assess the model sensitivity to the chosen parameters, we used the calibration parameter spaces and measured response variable sensitivity to each parameter with partial correlation coefficients (PCC). Moreover, we used a sequential calibration scheme to reduce computation load based on the hypothesis that the second calibration of litterfall parameters does not interfere with the first of forest structure parameters. To assess this assumption, we explored the sensitivity of forest structure variables to forest litterfall parameters." (l. 275-279)

- "Finally, in agreement with results on previous versions of the model, forest structure showed high sensitivity to the explored parameters. Partial correlation coefficients (PCC) were around -0.4 for $a_{CR}$ and around 0.4 for $b_{CR}$ with number of stems, aboveground biomass, and basal area. The background mortality rate $m$ also had a strong effect on aboveground biomass and basal area with a PCC around -0.2 but little to no effect on number of stems (Fig. A4)" (l. 361-364)

- "revealing a strong positive effect of $a_{T,0}$ and $b_{T,0}$ on the peak day of litterfall and negative effect on the ratio of the peak of litterfall, and a weak effect of $delta_0$ on the peak of litterfall. As anticipated, litterfall calibration was independent of the forest structure calibration (Fig. A4)" (l. 433-435)

[Figure]

**Figure A4: Sensitivity of forest structure (left panel) and forest litterfall (right panel) to calibrated parameters. Forest structure and forest litterfall sensitivity to each parameter was assessed with partial correlation coefficients (PCC) using the function *pcc* of the R package *sensitivity* with a thousand bootstrap draws to assess confidence intervals. The intercept and slope of the crown radius allometry $a_{CR}$ and $b_{CR}$ had a strong effect on forest structure, i.e., the number of stems (red), aboveground biomass (AGB, light green), and basal area (BA, green). Basal mortality $m$ also had a strong effect on aboveground biomass (AGB) and basal area (BA) but little to no effect on the number of stems. $a_{T,0}$ and $b_{T,0}$ had a strong positive effect on the peak day of litterfall (blue) and negative effect on the ratio of the peak of litterfall (purple), but $delta_0$ had only a weak effect on the peak of litterfall. The forest structure variables, namely number of stems, aboveground biomass (AGB), and basal area (BA) showed little to no partial correlations to $a_{T,0}$, $b_{T,0}$ and $delta_0$.**

Line 137: I disagree with the parameter values you used in this model, for example the leaf minimum conductance from literature. I read this paper and found that this value depends on the species. Why do you set it as 5? Also please state why giving the values to other parameters? This is very important!!

> We agree that $g_0$ is expected to vary across species. However, implementing such variation would require either measured values for hundreds of tropical tree species, or robust relationships to infer it from other traits. To our best knowledge, and despite growing interest in such hydraulic traits, both remain beyond the current state of the art. We therefore made the parsimonious assumption of a single average value for $g_0$ across species. Such choice, as well as others, are explained in the companion paper:

"The parameter $g_0$ quantifies water fluxes through the leaf cuticle (cuticular conductance) and from stomatal leaks. Although it is increasingly recognized as a key parameter explaining tree water loss in drought conditions (Cochard, 2021; Martin-StPaul et al., 2017), its values and variation with other functional traits is poorly documented (Duursma et al., 2019; Slot et al., 2021; Nemetschek et al., 2024), and we here assumed a fixed value. Note that some previous studies have defined $g_0$ as cuticular conductance only, ignoring stomatal leak effects, and thus underestimating $g_0$." (l. 458-462 of the companion paper).

Parameters from supplementary table A1 are set at the community or ecosystem level and do not vary among species, and their values are either

- "Assumed", a value that is supposed,
- "Calibrated", a value that has been calibrated,
- "Constant", a physical constant,
- "Inferred", a value that has been derived from an existing dataset,
- or from the "Literature", a value prescribed from the literature,

as specified in the fifth column of Table A1.

Line 138 why other parameters are assumed site independent.

> As explained above, we minimized the number of site-specific parameters to perform a conservative test of the model transferability. In doing so, we believe we provide a more conservative evaluation of the model performance, which does not rely on computationally costly parameter fine-tuning.

Lines 164-165: Again, if these parameters are not used in your model previously in the same ecosystem, I think you should calibrate them instead of directly using the literature values.

> Soil data are an input to the TROLL model and were collected at each site as measured values. They are not parameters of the model *per se* and therefore do not require calibration.

Line 224: I don't think its an exhaustive search, more parameter sampling is needed

> Parameterizing a model is always a balancing act between computational burden and accuracy. No search is exhaustive, and one should be mindful of the limitations of this exercise. However, we maintain that the extent of parameters space we used here was sufficient to explore a wide range of resulting litterfall peaks as shown in supplementary figure A5, from no peak at all to a peak exceeding several times the intensity observed in the field. The step used between two parameters values for $delta_0$ was supported by the observed low sensitivity of litterfall to $delta_0$ in preliminary and sensitivity analyses (as shown above).

Line 229: I'm confused why convert SIF to GPP for validation there are several GPP data can be used

> SIF based GPP products are now being commonly used in model forcing, and we wanted to check their quality against (1) eddy flux tower data and (2) our model outputs. There are indeed several GPP products available, and we chose to use the one from Chen et al. (2022), which is derived from solar induced fluorescence (SIF). However, we are open to the inclusion of others at the suggestion of reviewer 1.

Please state the statistical indicators of consistency between simulations and observations in figures 1 to 10

> To compare simulations and observations, we chose to use the same metrics for all variables, regardless of their type, origin, spatial or temporal resolution, in order to facilitate comparison of the evaluation across variables (see Supplementary Table A2). We used the goodness of fit $R^2$ from linear regression with null intercept, the Pearson's correlation coefficient CC, the root mean square error of prediction RMSEP, the standard deviation of the error of prediction SD, three of which are classical metrics used in Taylor plots. To clarify this, we further added a sentence in the Material and Methods: "To compare simulations and observations, we used the same metrics for all variables, regardless of their type, origin, spatial or temporal resolution: the goodness of fit $R^2$ from linear regression with null intercept, the Pearson's correlation coefficient CC, the root mean square error of prediction RMSEP, the standard deviation of the error of prediction SD." (l. 346-349).

If the results are statistical significant or not in Figure 11 and 12

> Estimating statistical significance would require an appropriate null model, but constructing such a null model is non-trivial due to the complex underlying data structures. We focused on the different proportions of GPP and ET variation that can be predicted from the different drivers using the R2, and compared these between drivers and between simulations and observations.

Lines 473-475: I don't think so for basal area, I noticed there are obvious deviations between your model simulations and field observations.

> True, thanks, we modified the text as follows: "The calibration of three global parameters resulted in simulated number of stems across size classes and basal area or aboveground biomass in good agreement with observations from forest inventories above 10-cm dbh" (l. 541).

General Comments: I think the discussion part is too long and didn't combine with your results closely. Reorganizing this part is recommended.

> We strongly believe that the flow is the one we would like to convey in this paper, and leave it to the Editor to make a recommendation.

**Reviewer 2:**

This manuscript evaluates the new TROLL 4.0 model at two Amazon sites with trait, census, flux tower, and remote sensing data. Specifically, the study evaluates forest structure with census and lidar data, composition with species abundance, phenology with litter and LAI seasonality, and carbon and water fluxes with eddy covariance tower.

Calibrating and evaluating the species abundance and tropical phenology are very novel to me and it is good to see that GPP and ET seasonality generally matches the tower observations. Please see below for my comments.

> We thank you for your thorough review of our manuscript and hope that the improvements we have made following your comments and those of the other reviewers will answer your questions.

Overall, it is very surprising to me that the study does not evaluate demographics (growth/mortality) of the model at the two sites. Given TROLL 4.0 is an individual-level model to predict forest community and ecosystem dynamics, it is easy and critical to benchmark growth (e.g. growth-size relationship) and mortality, and biomass changes. I believe both sites have multiple census data for such comparison?

> We agree and thank reviewer 2 for this comment. We retrieved simulated individual-tree growth rates (cm $yr^{-1}$) and death rates (% $yr^{-1}$) over 10 years per 5-cm diameter at breast height (dbh) classes and compared them to the ones estimated from inventories of six 6.25-ha plots in Paracou from 2003 to 2013 (we do not have access to multiple census data in Tapajos). Overall, the individual-level growth rates simulated by TROLL match relatively well the ones estimated from the 10-year forest inventories (Fig. 6 below). The simulated mean growth rate across all classes is 0.18 cm $yr^{-1}$, slightly higher than the 0.13 cm $yr^{-1}$ observed at Paracou. This is due to higher growth rates in the small diameter class in the simulations compared to observations, despite a realistic reduction in growth rates of large trees (>50 cm), resulting in the expected bell-shaped growth-size relationship (Hérault et al. 2011). Furthermore, the variation in the simulated growth-size relationship was within the range of observations, which are large due to low measurement precision compared to growth rates and large inter- and intra-species variation (Schmitt, Hérault & Derroire, 2023). The simulated death rates were similar to those observed, with a mean simulated mortality rate across all classes of 1.73 % $yr^{-1}$ lower than the observed mean of 2.60 % $yr^{-1}$ due to underestimation of small and large tree mortality, but with close means per dbh class and largely overlapping confidence intervals. These new results have been added in the new

appendix figure Fig. 6 reported below, and described in the renamed sections "Forest structure, composition and dynamics":

- "Finally, we evaluated forest dynamics by retrieving simulated individual-tree growth rates (cm yr$^{-1}$) and death rates (% yr$^{-1}$) over 10 years per 5-cm diameter classes and comparing them to the ones estimated from field inventories of six 6.25-ha plots in Paracou from 2003 to 2013". (l. 296-298)

- "Forest dynamics simulated by TROLL 4.0 were consistent with the ones estimated from field inventories at Paracou (Fig. 6). Simulated individual-tree growth-size relationship were comparable to the ones retrieved from inventories (simulated mean of 0.18 cm yr$^{-1}$ against 0.13 cm yr$^{-1}$) with an expected bell-shaped relationship (Hérault et al. 2011) and similar high variation (Schmitt, Hérault & Derroire 2023). Simulated death rates also showed magnitude and variation similar to observed ones (simulated mean of 1.73 % yr$^{-1}$ against 2.60 % yr$^{-1}$ observed at Paracou but with consistent and overlapping ranges)." (l. 418-422)

- "Similarly, the multiple inventories at Paracou from 2003 to 2014 revealed a good ability of TROLL 4.0 to simulate forest dynamics with both bell-shaped growth-size relationship and tree mortality." (l. 544-545)

[Figure]

**Figure 6: Forest dynamics at Paracou and Tapajos, expressed in terms of individual-tree growth rate (top) and death rate (bottom) both per 5 cm-dbh classes across 10 years. The figures compare distributions simulated by TROLL 4.0 in blue and multiple field inventory observations from six 6.25-ha plots in Paracou from 2003 to 2013 in black. Simulated values and their confidence intervals correspond to ten repetitions of 10-year simulations starting from the**

**end-state of 600-year regeneration from bare ground with calibrated parameters at each site. Confidence intervals at 95 % are shown with shaded areas and are based on variations among plots (6 plots of 6.25 ha) for the observations.**

In addition, given the novelty of the version is to better represent plant water stress and phenology, I would recommend conducting a simulation at a drier site. For example, in the ED2.2 GMD paper (Longo et al. 2019), both Paracou and Brasilia are tested to evaluate model behavior and performance under different degrees of water stress.

> The ED2.2 GMD description paper (Longo et al. 2019a, Part 1; https://doi.org/10.5194/gmd-12-4309-2019) included the woody savanna site of Brasilia to showcase the ability of ED2.2 to simulate ecosystem responses to fire disturbances (Fig. 8 therein). Simulating fire regimes is out of the scope of TROLL 4.0. The ED2.2 GMD evaluation paper (Longo et al. 2019b, Part 2; https://doi.org/10.5194/gmd-12-4347-2019) focused exclusively on Paracou and Tapajos to evaluate model behaviour and performance, and in particular water fluxes, as we do here. This results in differences in water regimes and plant water stress and phenology. The same two sites are therefore denoted the "dry" site and the "wet" site in Longo et al. (2018). This is now hopefully better described in the Material and Methods section: "Both sites are covered by a high biomass and species rich lowland moist tropical forest, and they present contrasting soil characteristics and climate (Table 1), with a longer dry season in Tapajos than in Paracou resulting in 2,075 mm per year against 3,041 in Paracou. They thus differ in water regimes and resulting plant water stress and phenology. In addition, the two sites have been intensively monitored for several decades, mainly through repeated forest inventories and eddy flux tower measurements, fulfilling the requirement for in-depth model evaluation as previously used for such applications (Longo et al., 2019b)." (l. 120-124).

Some more detailed comments

L 44, 'canopy height distribution' is a vague term to me. How exactly is this distribution calculated? Based on the manuscript, I believe leaf/plant area density is more accurate?

> We refer here to the distribution of top canopy height across 1-m2 pixels, derived from a canopy height model obtained from Airborne Lidar Scanning or from the plot structure at the end of a 600-year simulation (Figure 2), and not to the temporal variation of leaf area index (Figure 7). We thus modified the sentence to clarify this as follows: "We assessed the model's ability to represent forest structure and composition using the lidar-derived spatial distribution of top canopy height and forest inventories combined with information on plant functional traits" (l. 40).

L 48-49 it would be helpful to report the magnitude of overestimation in addition to RMSEP, which includes both difference in mean and variability.

> We totally agree and added the error standard deviation: "However, TROLL 4.0 overestimated annual gross primary productivity at both sites (mean RMSEP=0.94±0.67 kgC m$^{-2}$ yr$^{-1}$) and evapotranspiration at one site (mean RMSEP=0.75±0.63 mm day$^{-1}$)" (l. 49-50).

L 119-120 related to my comments for the Part I manuscript, it would be helpful to show model sensitivity to these global parameters.

> As answered to the reviewer 1, we now measure the response variable sensitivity to each parameter with partial correlation coefficients (PCC). To do so, we use the function *pcc* of the

R package *sensitivity* with a thousand bootstrap draws to assess confidence intervals. As expected the intercept and slope of the crown radius allometry $a_{CR}$ and $b_{CR}$ had a strong effect on forest structure with a PCC around -0.4 for $a_{CR}$ and around 0.4 for $b_{CR}$ with number of stems, aboveground biomass (AGB), and basal area (BA). The basal mortality $m$ also had a strong effect on aboveground biomass (AGB) and basal area (BA) with a PCC around -0.2 but little to no effect on the number of stems (PCC around 0). Similarly, $a_{T,0}$ and $b_{T,0}$ had a strong positive effect on the peak day of litterfall (PCC close to 0.5) and negative effect on the ratio of the peak of litterfall (PCC around 0.3), but $delta_0$ had only a weak effect on the peak of litterfall. These new results have been added in the new appendix figure Fig. A4 reported below, and described in material and methods and results:

- "To assess the model sensitivity to the chosen parameters, we used the calibration parameter spaces and measured response variable sensitivity to each parameter with partial correlation coefficients (PCC). Moreover, we used a sequential calibration scheme to reduce computation load based on the hypothesis that the second calibration of litterfall parameters does not interfere with the first of forest structure parameters. To assess this assumption, we explored the sensitivity of forest structure variables to forest litterfall parameters." (l. 275-279)

- "Finally, in agreement with results on previous versions of the model, forest structure showed high sensitivity to the explored parameters. Partial correlation coefficients (PCC) were around -0.4 for $a_{CR}$ and around 0.4 for $b_{CR}$ with number of stems, aboveground biomass, and basal area. The background mortality rate $m$ also had a strong effect on aboveground biomass and basal area with a PCC around -0.2 but little to no effect on number of stems (Fig. A4)" (l. 361-364)

- "revealing a strong positive effect of $a_{T,0}$ and $b_{T,0}$ on the peak day of litterfall and negative effect on the ratio of the peak of litterfall, and a weak effect of $delta_0$ on the peak of litterfall. As anticipated, litterfall calibration was independent of the forest structure calibration (Fig. A4)" (l. 433-435)

[Figure]

**Figure A4: Sensitivity of forest structure (left panel) and forest litterfall (right panel) to calibrated parameters. Forest structure and forest litterfall sensitivity to each parameter was assessed with partial correlation coefficients (PCC) using the function *pcc* of the R package *sensitivity* with a thousand bootstrap draws to assess confidence intervals. The intercept and slope of the crown radius allometry $a_{CR}$ and $b_{CR}$ had a strong effect on forest structure, i.e., the number of stems (red), aboveground biomass (AGB, light green), and basal area (BA, green). Basal mortality *m* also had a strong effect on aboveground biomass (AGB) and basal area (BA) but little to no effect on the number of stems. $a_{T,0}$ and $b_{T,0}$ had a strong positive effect on the peak day of litterfall (blue) and negative effect on the ratio of the peak of litterfall (purple), but $delta_0$ had only a weak effect on the peak of litterfall. The forest structure variables, namely number of stems, aboveground biomass (AGB), and basal area (BA) showed little to no partial correlations to $a_{T,0}$, $b_{T,0}$ and $delta_0$.**

L 126, what are the physiological interpretations of a_T,o, b_T,o and detla_o? How are they related to different phenological strategies? This is related to my confusion in the phenological module in the companion manuscript.

> We have now further clarified the significance of those parameters in the companion paper, and provide a more explicit description of their effect in this evaluation part as follows: "The first term accounts for a decline in leaf drought tolerance with age, i.e. a reduced ability of old leaves to maintain turgor when the soil dries, where $a_{T,0}$ controls the ratio of the turgor loss point of old to mature leaves. The second term accounts for the height dependence of this susceptibility to decreasing water availability: it makes large trees susceptible to a (small) decrease in soil water availability $b_{T,0}$, while preventing them from constantly shedding their old leaves at a fast rate. Finally, $\delta_o$ controls the rate of leaf shedding in old leaves as they begin to lose turgor, but in the absence of water depletion. Overall, the parameters $a_{T,o}$, $b_{T,o}$ and $\delta_o$ control the intensity and timing of the peak of litterfall under drying soil conditions." (l. 141-147). Note that these three parameters are global and thus

independent of species functional strategies. However, the current implementation should induce some dependencies between species drought tolerance, their stature and their phenological strategies. Indeed, as explained in the companion paper, the tree height above which old leaves becomes susceptible to a small decrease in soil water availability is jointly dependent on $aT,o$, $bT,o$ and $\pi\_tlp$. As an illustration, for $aT,o$=0.2 and $bT,o$=0.02, it is 28m for a tree with a $\pi\_tlp$ of -1.5 MPa and 58m for a more tolerant tree with a $\pi\_tlp$ of -3 MPa. As a result, tree height is much more constrained for species with a low leaf drought tolerance, and a tall but drought vulnerable tree will exhibit a drought-deciduous habit, in agreement with the empirical literature. This is now better discussed in the companion paper. Relevance of such implementation could be better scrutinized and compared with alternative ones in future contributions as already highlighted in the discussion.

Line 133, Table 1, is Soil Organic Content/Bulk Density/CEC/pH used in TROLL 4.0?

> Sorry, we forgot to mention it. This is now included in the revised legend of Table 1: "Site overview with climate, vegetation and soil properties. Soil properties are those used as input of the pedotransfer functions implemented in TROLL 4.0" (l. 151-152).

Line 150-153, I like the efforts to capture inter-specific variation in height allometry. I wonder what's the correlation between a_h and h_lim with other functional traits. One editorial comment, superscript is not shown properly in the equation and sentences.

> Thank you. The correlations of $a_h$ and $h_{lim}$ with other functional traits was low and negligible (see figure below). We altered the rendering of superscripts for sigma parameters in the revised version of the manuscript.

[Figure]

**Species traits pairwise correlations before imputation. The number of species per trait for pairwise complete correlation are 125 for $\pi\_tlp$ (TLP), 192 for $dbh_{thres}$, 252 for ah, 252 for hlim, 2,028 for LMA, 2,044 for N, 2,044 for P, 2,053 for WSG, and 2,057 for LA.**

Line 162, Fig. A1 shows that over 2000+ traits are imputed based on 100+ trait data. I am a little concerned about the disproportionately large fraction of imputed data…

> Our Fig. A1 was misleading. We used traits of 114 species for simulations for which we had 107 to 110 raw values of $a_h$, $dbh_{thres}$, $h_{lim}$, and TLP and solely imputed 4 to 7 values. We thus clarified this in the methods and updated the figure A1 accordingly as follows:

" Overall, this procedure leads to a parameterization of 114 species for Paracou and 113 species for Tapajos, with imputed values for only 4 to 7 species for $a_h$, $dbh_{thres}$, $h_{lim}$, and $\pi\_tlp$ (Fig. A1)." (l. 182-183).

[Figure]

**Figure A1: Representativity of imputed functional traits values (red) against raw functional trait values (blue) from various datasets (see methods). Traits were imputed using predictive means matching for $a_h$, $dbh_{thres}$, hlim, and $\pi_{tlp}$ only. The number in each subplots represents the number of species with a trait value in the raw data and after imputation composing respectively the blue and red curves.**

Line 173-175, A minor comment, ERA5 rainfall data is likely not trustworthy at daily scale in the tropics. CHIRPS data might have better performance in my personal experience.

> Thank you for sharing your personal experience. We here used ERA5-Land given the good match with the local eddy-flux tower in Paracou (Schmitt et al. 2023). Following your comment, we further compared both CHIRPS and ERA5-Land against local eddy-flux tower measurements in Paracou extracted from FLUXNET 2015, at daily, 5-day, 15-day and monthly resolutions. CHIRPS and ERA5-Land had similar agreement to locally measured precipitations (figure below), with even higher correlations for ERA5-Land than CHIRPS. However, they both overestimated low precipitation events and underestimated high precipitation events, resulting in a low agreement for daily variations ($R^2$ of 0.10 and 0.22 for CHIRPS and ERA5-Land respectively), which quickly increases for 5-day ($R^2$ of 0.39 and 0.53) and 15-day variations ($R^2$ of 0.63 and 0.73). A similar assessment was not possible in Tapajos due to a lack of reliable rainfall data, so we decided to keep ERA5-Land for filling the precipitation data gaps in Tapajos. We thank the reviewer 2 for highlighting the alternative dataset and incentivizing us to further assess the reliability of ERA5-Land. These new results have been added in the new appendix figure Fig. A2 reported below, and described in material and methods: "ERA5-Land showed a better agreement with on-site precipitation data from FLUXNET 2015 at Paracou when compared to other products, like CHIRPS (Funk et al. 2015; Fig. A2)" (l. 197-198).

[Figure]

**Figure A2: Comparisons of CHIRPS (red) and ERA5-Land (blue) precipitation products against local eddy-flux tower measurements retrieved from FLUXNET 2015 in Paracou at daily, 5-day, 15-day, and monthly resolutions. CHIRPS and ERA5-Land had similar agreement to locally measured precipitations, with even higher correlations for ERA5-Land than CHIRPS. However, they both overestimated low precipitation events and underestimated high precipitation events, resulting in low agreement for daily variations ($R^2$ of 0.10 and 0.22), which quickly increases for 5-day ($R^2$ of 0.39 and 0.53) and 15-day variations ($R^2$ of 0.63 and 0.73). Although a similar assessment was not possible**

**in Tapajos due to a lack of local reliable rainfall data, we therefore decided to keep ERA5-Land for filling the precipitation data gaps in Tapajos.**

Line 210-213, This calibration approach assumes the forest dynamics is in equilibrium. I am fine with this assumption since we don't have much better alternatives when disturbance history is unknown. Meanwhile, this means the calibrated m includes the average disturbance rate of each site (i.e. it is not just ageing and biology).

> This is indeed a necessary hypothesis that we added when presenting the sites: "We assumed forest dynamics to be at equilibrium, as both sites are characterised by old-growth forests" (l. 124). We also now better specify what the background mortality rates actually include: "$m$ can be site-specific as it is used to simulate tree mortality events that are triggered by processes not explicitly represented in the model, such as site-specific disturbance regimes (e.g. Rau et al. 2022)" (l. 131-133).

Line 216, this equation basically assumes equal weights between (1) AGB, (2) total stem density and (3) stem density in each size class, which seems arbitrary to me. Why not check the best trait combinations for each metric and see whether these 'posterior' trait combinations overlap?

> We explored different weighting options and several indices. The equation assuming equal weights, despite being the simplest, entailed the best tradeoff among metrics. However, we reported in the table A4 below the 5 best simulations for each of stem distribution, number of stems, basal area and aboveground biomass, as well as the one with such equal weighing. This highlights the total overlap of posterior parameters combinations across metrics:

**Table A4: Calibrated parameters intervals for the 5 best 5 best simulations for each of stem distribution, number of stems, basal area and aboveground biomass, as well as the one with equal weighing among them. Values show median first followed by minimum and maximum values in brackets.**

| Site | Metric | RMSEP | $a_{CR}$ | $b_{CR}$ | $m$ |
|------|--------|-------|----------|----------|-----|
| Paracou | Number of stems | 5.75 [2-7.75] | 1.75 [1.75-1.8] | 0.3575 [0.3575-0.386] | 0.0475 [0.0375-0.05] |
| Paracou | Basal Area | 0.04 [0.03-0.07] | 1.85 [1.65-2] | 0.4715 [0.3505-0.5075] | 0.0325 [0.03-0.05] |
| Paracou | Stem distribution | 2.4 [1.38-2.7] | 1.85 [1.8-1.9] | 0.4145 [0.386-0.443] | 0.0425 [0.0325-0.05] |
| Paracou | All equally weighted | 0.16 [0.13-0.17] | 1.8 [1.8-1.9] | 0.386 [0.386-0.443] | 0.0325 [0.0325-0.0375] |
| Tapajos | Number of stems | 3 [0-3.5] | 2.5 [2.4-2.65] | 0.785 [0.728-0.9205] | 0.035 [0.03-0.04] |
| Tapajos | Aboveground Biomass | 0.13 [0.04-0.19] | 2.45 [2.35-2.5] | 0.835 [0.6495-0.885] | 0.045 [0.03-0.05] |
| Tapajos | Stem distribution | 2.54 [2.38-2.74] | 2.35 [2.3-2.35] | 0.6995 [0.671-0.6995] | 0.045 [0.0375-0.05] |
| Tapajos | All equally | 0.25 | 2.45 | 0.7565 | 0.04 [0.03-0.05] |

| | weighted | [0.18-0.25] | [2.35-2.5] | [0.6995-0.785] |

> Similarly, we explored different weights and different indices and this equation assuming equal weights, although the simplest, gave the best result. However, we have reported in the table below the best parameters set for each site and each metric, showing divergent posterior parameters depending on the calibration of the literfall peak day or ratio. As we were looking for the best representation of both the literfall peak day and the ratio at the same time, we decided to keep the equally weighted RMSEP, which gives parameter posteriors that are a compromise between the two.

**Best calibrated parameters of forest litterfall peak per site depending on the metric of evaluation.**

| Site | Metric | RMSEP | $a_0$ | $b_0$ | $delta_0$ |
|---|---|---|---|---|---|
| Paracou | Peak | 0.61 | 0.075 | 0.015 | 0.4 |
| Paracou | Peak | 0.61 | 0.100 | 0.015 | 0.4 |
| Paracou | Ratio | 0.17 | 0.200 | 0.015 | 0.3 |
| Paracou | Both equally weighted | 1.06 | 0.200 | 0.015 | 0.1 |
| Tapajos | Peak | 0.00 | 0.300 | 0.015 | 0.2 |
| Tapajos | Ratio | 0.06 | 0.100 | 0.050 | 0.4 |
| Tapajos | Both equally weighted | 0.33 | 0.200 | 0.015 | 0.2 |

Line 315-316, the m values make sense to me (2% baseline mortality + 1.5~2% disturbance rate). The big difference between a_CR and b_CR is interesting... These values mean the crown radius is *200%* higher for a 10cm tree at Tapajos compared with Paracou and the crown radius difference increases with tree size... Is this supported by canopy observations at the two sites? Given airborne lidar can identify crowns, does crown size show such a big difference between the two sites?

> The use of calibrated values in the allometry relating crown radius to dbh (equ. 17 within the companion paper) actually leads to a crown radius of 2.49 m and 2.03 m for a tree with a 10-cm dbh at Paracou and Tapajos respectively, i.e. a crown radius *22%* higher in Paracou than Tapajos. Although we had access to few crown measurements for our two sites in the TALLO database (Jucker et al. 2022), such variation falls well within the one reported in the resulting global analysis (Jucker et al. 2024, Fig. 1 therein). This is now highlighted in the text: "For tree with 10-cm dbh, the calibrated crown radius - dbh allometry (equ. 17 in Maréchaux et al., submitted companion paper) predicts a crown radius of 2.49 m at Paracou and 2.03 m at Tapajos a variation that falls well within the one reported globally (Jucker et al. 2024)" (l. 354-356).

Line 323: Fig. 1, given the large size-related variation in abundance, I suggest log-transform abundance before calculating correlations otherwise the correlation will definitely be high even if models might have large relative biases for big trees, which are important for ecosystem-level carbon/water fluxes.

> We compared the use of RMSEP computed either with the log-transformed or the untransformed stem density distribution. The use of log-transformed stem density distributions led to lower calibrated values of $a_{CR}$, $b_{CR}$ and higher value of $m$ in Paracou and Tapajos, indicating that large trees may have a relatively smaller crown radius relative to their dbh and smaller background mortality rates than the one previously driven by smaller trees using untransformed stem density distribution. However, the current untransformed stem density distribution gives the same parameter values as the one that gives the best simulated AGB and total number of stems, as shown in the previous answers. We have therefore decided to retain the untransformed distribution for parameter calibration for the time being.

**Intervals of parameters related to forest structure of the 5 best simulations per site, identified as the ones with the lower RMSEP of either untransformed or log-transformed stem density distribution. Values show median first, followed by minimum and maximum values in brackets.**

| Site | Stem density distribution | RMSEP | $a_{CR}$ | $b_{CR}$ | $m$ |
|---|---|---|---|---|---|
| Paracou | Untransformed | 2.4 [1.38-2.7] | 1.85 [1.8-1.9] | 0.4145 [0.386-0.443] | 0.0425 [0.0325-0.05] |
| Paracou | Log-transformed | 1.12 [1.11-1.14] | 1.8 [1.65-2] | 0.386 [0.3005-0.55] | 0.0325 [0.0325-0.035] |
| Tapajos | Untransformed | 2.54 [2.38-2.74] | 2.35 [2.3-2.35] | 0.6995 [0.671-0.6995] | 0.045 [0.0375-0.05] |
| Tapajos | Log-transformed | 1.22 [1.19-1.22] | 2.55 [2.5-2.7] | 0.9135 [0.8635-0.9635] | 0.0325 [0.03-0.045] |

Line 335-336, this (and Fig. 3) is likely because the model does not have light-driven plasticity. See my comments to the companion manuscript.

> We agree. We had mentioned it in the companion manuscript, but this was not clear enough. We now amended: "Another explanation could be a lack of light heterogeneity and associated trait variation in the understorey in simulations in comparison with observations (Montgomery and Chazdon, 2001), thus limiting the opportunities for recruitment and survival of small stems. Explorations of simulated micro-environmental variations within the canopy (de Frenne et al., 2019) and inclusion of trait ontogenetic shifts (Fortunel et al., 2019) and trait plasticity (Xu et al., 2017; Lamour et al., 2023) could further help understand and improve TROLL's ability to simulate forest structure and composition in the understory" (l. 551-556).

Line 339 Fig. 2, Given the model underestimates small trees in both basal area and abundance, I wonder why the model has a good match in understory density (fraction of voxels with leaves/branches) when compared with lidar data? I would expect the model has

lower density than observations... Or could the lidar data be problematic for the understory, especially at 1m resolution due to occlusion?

> Figure 2 does not show the fraction of voxels with positive plant area density at a given height, and in particular in the understory. Instead, for each 1-m$^2$ pixel of the ground, the canopy height in that pixel (i.e. the height of the highest voxel with positive PAD and located above this ground pixel, either simulated or inferred from lidar point clouds) was determined, and its distribution across 1-m2 pixels plotted in figure 2. This figure thus illustrates the canopy top height distribution, and notably the horizontal heterogeneity in canopy top height. The good match for low height values between simulated and lidar-derived distributions indicates that the model simulates the frequency of forest gaps with low canopy height correctly, but this is unrelated to the underestimation of small tree basal area and number of stems. What Figure 2 shows is now clarified in its legend to clarify this point:

"Canopy height distribution at Paracou and Tapajos. For each 1-m$^2$ pixel of the ground, the top canopy height in that pixel (i.e. the height of the highest voxel with positive plant area density, or PAD, and located this ground pixel) was determined, and its distribution across 1-m2 pixels plotted as the proportion of 1-m2 ground pixels (%, x-axis) with a given canopy height (m, y-axis, at 1-m resolution). The figure shows a comparison between distributions derived from PAD fields simulated by TROLL 4.0 (blue lines), and the ones derived from airborne laser scanning point clouds (black lines). Simulated values and their confidence intervals correspond to the end-state of simulations of ten 4-ha 600-year regeneration from bare ground for each site." (l. 383-388).

Investigating the variation of leaf area density across the canopy vertical profile would be valuable, but TLS lidar data from Smith et al. (2019) have proven not reliable enough to represent LAD variation at Tapajos (Figure 7) and UAV lidar data from Paracou might lack penetration and need adjustment to accurately represents understory LAD (G. Vincent pers. com.). We thus leave this evaluation and discussion for future contributions.

Line 355-356, species rank comparison is cool and interesting. I wonder whether the species in the observation and simulation are paired? i.e., is the most abundant species in the model also the most abundant in the observation? I am interested in a scatter plot of species rank in model vs observations. If they are close to a 1:1 line, that means the trait-based approach and 3D simulation can explain a large part of biodiversity.

> No. We compared the species rank-abundance distribution only and not species by species. We agree that this would be a very interesting study, but would require context from community ecology, and would not be a good fit for this manuscript. We are looking forward to exploring this question in a forthcoming paper. We do not expect the model to simulate species abundance closely matching observations. Field abundances result from complex interactions, including (i) community assembly, (ii) biogeographic processes, and (iii) disturbance history. While the model captures (i) through trait-based parameterization and resource competition, it omits (ii) and (iii), which may explain varying abundances among species with similar traits. Our use of homogeneous seed rain intentionally excludes (ii) and (iii) for a conservative assessment of (i). As expected, observed and simulated genus abundances differ (first figure below), but the slopes of species trait-abundance relationships

are similar (second figure below), with intercept differences driven by species richness discrepancies.

[Figure]

**Simulated mean genus abundances across simulation repetitions against observed mean genus abundance from inventories across plots. The black line shows the expected 1:1 relation and the blue line the linear relation with a goodness of fit R² of 0.002.**

[Figure]

Simulated mean genus abundances (blue) among simulation repetitions and observed mean species abundance (black) from inventories across plots against dbh$_{thres}$: maximum diameter in m, LA: leaf area in cm$^2$, LMA: leaf mass per area in g cm$^{-3}$, N$_{mass}$: leaf nitrogen content per dry mass in mg g$^{-1}$, P$_{mass}$: leaf phosphorus content per dry mass in mg g$^{-1}$, $\pi_{tlp}$: leaf water potential at turgor loss point in MPa, WSG: wood specific gravity in g cm$^{-3}$. The lines are the linear relation with a goodness of fit indicated above.

Line 382, litter mismatch might be because leaf aging is not considered

> Leaf aging is considered in TROLL 4.0 as shown with equation (56) in line 687-689 from the description companion paper. This litter mismatch is thus discussed in different ways (l. 598-655).

Line 410, Fig. 8. It seems there is no consensus of leaf age cohort seasonality between observations. How should we interpret this modeling results, which are constrained by the Wu et al. PhenoCam data right?

> Wu et al. (2016) PhenoCam data was not used to constrain TROLL outputs or the Yang et al. (2023) reanalysis. The comparison between Wu et al. (2016) and simulated outputs thus consists in an independent evaluation of TROLL's ability to simulate leaf demography and Yang et al (2023)'s reanalysis. The independence between these three sources is now recalled in the figure's legend. There is indeed a strong disagreement between observations from the PhenoCam from Wu et al. (2016) and the reanalysis of Yang et al. (2023) for seasonal variations in old and young leaves. We chose to kept both as we had no other products than Yang et al. (2023) at Paracou, but explained why we were more confident in locally observed PhenoCam data from Wu et al. (2016) in the discussion, highlighting the need for more research on this topic: "The seasonal dynamics of leaf cohorts remains poorly known in tropical forests and additional high-resolution optical imagery, e.g. by drones or phenological cameras, would be extremely useful to better document these patterns" (l. 654-655).

Line 425, Fig. 9. Could it be that the model is not sensitive enough to VPD (based on Fig. 11) If so, the model might overestimate GPP more in the midday/afternoon. It would be helpful to compare the dry/wet season average diurnal cycle of GPP between tower data and TROLL simulations.

> This is a possibility. The idea to explore the simulated vs observed dependencies of LUE to environmental drivers (see answer below) allowed us to quantify the limitations of GPP independently of the overriding effect of light variation across seasons. As suggested by the reviewer, the limitation of LUE at high values of daily maximal VPD (typically reached in the midday/afternoon) is underestimated in simulations in comparison to eddy flux-derived estimates. We already discussed this in the manuscript: "TROLL 4.0 may underestimate the stomatal control of transpiration during the dry season. [...]. Underestimation of stomatal control can result from the representation of stomatal conductance and its responses to soil water availability." (l. 688-671). We now further highlight that, the use of daily leaf predawn water potential, and not hourly leaf water potential in the water stress factors (equ. 39 and 40 in the companion paper) may contribute to explain the underestimation of stomatal control and the resulting overestimation of both transpiration and productivity. Variation of leaf water potential results from both the atmospheric demand (VPD) and soil water source (leaf predawn water potential) and specific tree functional strategies to respond to these two drivers, including tree vulnerability to cavitation and capacitance. Accordingly, the discussion section has been modified as follows:

"Underestimation of stomatal control can result from the representation of stomatal conductance and its responses to atmospheric dryness and soil water availability. In particular, the use of daily leaf predawn water potential to control leaf-level gas exchange, and not hourly variation of leaf water water potential, (see equ. 39 and 40 in Maréchaux et al. submitted companion paper) can explain the overestimated ecosystem-level fluxes during the dry season. More generally, the understanding of leaf- to ecosystem-level fluxes are active areas of research and alternative representations could be considered in the future as availability of data increases (Wolf et al. 2016; Anderegg et al. 2018; Sabot et al., 2022, Lamour et al., 2022; see sections 2.5.2, 2.5.3 and 4.1, and Appendix B in Maréchaux et al. submitted companion paper). Alternatively and/or concurrently, during the dry season, a lack of stomatal control can be due to an overestimation of soil water availability in the model." (l. 671-679).

"In addition, after removing the effect of absorbed light, simulated GPP showed less limitation to high values of VPD and temperature compared to eddy flux- or SIF-derived estimates. The response of leaf-level gas exchanges to the joint effect of atmospheric dryness and soil water availability shows no clear consensus across models (Powell et al, 2013; Trugman et al., 2018), and could be underestimated during the dry season in TROLL 4.0 simulations as discussed above for transpiration." (l. 704-708).

Line 441, Fig. 10, I am curious about the evaporation/transpiration partitioning as well as soil moisture dynamics in the model. These are new components of the model and they should be reported and discussed and they are important to understand the process-level accuracy of the model.

> Soil moisture dynamics is shown in appendix (figures A10 and A11 of the new version of the manuscript) and described in the results section of the manuscript: "TROLL 4.0 also captured the seasonality in RSWC of the top soil layer at Paracou and Tapajos (Fig. A6, Table S2, see Fig. A6 for absolute variation with varying depth), with a high RSWC in the wet season close to 100% and a sharp decrease in RSWC in the dry season, although overall smoother in simulations than field estimates" (l. 498-501). We added a new appendix figure A9 (see below) to represent the fortnightly evapotranspiration partitioning between canopy evaporation, soil evaporation, and tree transpiration and the corresponding description in the updated manuscript: "The partitioning of evapotranspiration between canopy evaporation, soil evaporation and tree transpiration (Fig. A9) showed that most of the evapotranspiration is due to tree transpiration in the dry season, while canopy evaporation is an important part of the total evapotranspiration in the wet season (Kunert et al., 2017)." (l. 495-498).

[Figure]

**Figure A9: Mean annual cycle of evapotranspiration partitioning between canopy evaporation (red), soil evaporation (green), and tree transpiration (blue) for Paracou (left) and Tapajos (right), derived from fortnightly means simulated with TROLL 4.0. Bands are the intervals of means across years, and the yellow vertical bands in the**

**background correspond to the site's climatological dry season. Simulated values correspond to 10 years of simulations starting from the end-state of 600-year regeneration from bare ground with calibrated parameters at each site.**

Line 457, Fig. 11, given the high collinearity between VPD and temperature and PAR, why not using multivariate regression or other techniques to separate partial sensitivities? The gross correlation/regression is not really informative (e.g. GPP increases with VPD) A good reference is the analysis in Bloomfield et al. 2023. Based on the current analysis, VPD limitation in the model is too weak. I wonder whether it is because of the usage of Penman-Monteith in calculating and constraining stomatal processes. It will be helpful to plot and analyze key intermediate variables in photosynthesis-stomata process (e.g. gs, VPDs, Ci, etc) to diagnose the problem.

> We thank the reviewer 2 for the reference to Bloomfield *et al.,* (2023). We computed the light use efficiency (LUE) in $mol_C \ mol_{photons}^{-1}$ normalizing gross primary productivity (GPP) by photosynthetic photon flux density (PPFD) and the fraction of absorbed photosynthetically active radiation (fAPAR), itself derived from leaf area index (LAI). This allows further comparison of observed and simulated sensitivity of LUE to vapour pressure deficit (VPD), temperature (T), and wind speed (WS) as shown in the new appendix figure A12 shown below. Based on this result it seems that VPD limitation of LUE (left column) in the model (blue) is matching the one observed with eddy flux towers (black). However, it is true that the limitation is lower for high values of VPD as compared to observations, especially in Tapajos. This is presented as "We additionally computed the light use efficiency (LUE in $mol_C$ $mol_{photons}^{-1}$ ) normalizing GPP by photosynthetic photon flux density (PPFD) and the fraction of absorbed photosynthetically active radiation (fAPAR) derived from leaf area index (LAI) to explore carbon flux environmental drivers independently of the overriding effect of light as in Bloomfield *et al.* (2023)." (l. 331-334) and described the comparison between observed and simulated relationships as follows: "Similarly, controlling for absorbed light, both eddy flux-derived and simulated LUE showed a negative logarithmic relationship with maximum VPD and a negative linear relationship with mean temperature at daily scale (Fig. A12). Limitations of LUE at high VPD and T values were however lower in simulations than in eddy flux- or SIF-derived estimates." (l. 510-513) in the revised version of the manuscript. And the discussion on the overestimated GPP during the dry season now reads:

"TROLL 4.0 tended to overestimate empirical GPP estimates, particularly during the dry season, in comparison to both eddy covariance- and SIF-derived GPP. GPP is driven by the photosynthetic activity of the canopy, which depends on multiple processes (Diao et al., 2023; Slot et al., 2024) and further work would be needed to precisely discriminate among them, while accounting for eddy covariance uncertainties (Cui and Chui, 2019). Absorbed light typically has an overriding effect on GPP and its variation across seasons in these light-limited rainforest systems (Yang et al. 2022, Guan et al. 2015), and simulated GPP is sensitive to the parameters that control light transmission and absorbance (light extinction coefficient, apparent quantum yield; Maréchaux & Chave, 2017). Both are assumed fixed and constant in simulations, but are known to vary with leaf angle distribution and leaf optical properties, depending on micro-environmental conditions and species (Long et al., 1993; Poorter et al., 1995; Meir et al., 2000; Kitajima et al., 2005). In addition, after removing the effect of absorbed light, simulated GPP showed less limitation to high values of VPD and temperature compared to eddy flux- or SIF-derived estimates. The response of leaf-level gas exchanges to the joint effect of atmospheric dryness and soil water availability shows no clear consensus across models (Powell et al, 2013; Trugman et al., 2018), and could be

underestimated during the dry season in TROLL 4.0 simulations as discussed above for transcription." (l. 696-708)

[Figure]

**Figure A12: Daily averages of light use efficiency as a function of daily maximum vapour pressure deficit, average temperature, and average wind speed for model-, satellite- and eddy flux-based estimates at Paracou (top) and Tapajos (bottom). Lines illustrate the linear regression of form y ~ log(x), and text the squared Pearson's R correlation coefficient. The light use efficiency (LUE) was obtained by normalizing gross primary productivity by photosynthetic photon flux density (PPFD) and the fraction of absorbed photosynthetically active radiation (fAPAR) itself derived from leaf area index.**

L. 480-482. I disagree, I think it is mainly due to lack of trait plasticity (check Xu et al. 2017 and Lamour et al. 2023)

Xu, X., Medvigy, D., Joseph Wright, S., Kitajima, K., Wu, J., Albert, L. P., Martins, G. A., Saleska, S. R., & Pacala, S. W. (2017). Variations of leaf longevity in tropical moist forests predicted by a trait-driven carbon optimality model. Ecology Letters, 20(9), 1097–1106. https://doi.org/10.1111/ele.12804

Lamour, J., Davidson, K.J., Ely, K.S., Le Moguédec, G., Anderson, J.A., Li, Q., Calderón, O., Koven, C.D., Wright, S.J., Walker, A.P., Serbin, S.P. and Rogers, A. (2023), The effect of the vertical gradients of photosynthetic parameters on the CO2 assimilation and transpiration of a Panamanian tropical forest. New Phytol, 238: 2345-2362. https://doi.org/10.1111/nph.18901

> This is one scenario. We slightly altered the text as follows: "Another explanation could be a lack of light heterogeneity and associated trait variation in the understorey in simulations in comparison to observations (Montgomery and Chazdon, 2001), thus limiting the opportunities for recruitment and survival of small stems. Explorations of simulated micro-environmental variations within the canopy (de Frenne et al., 2019) and inclusion of trait ontogenetic shifts (Fortunel et al., 2019) and trait plasticity (Xu et al., 2017; Lamour et

al., 2023) could further help understand and improve TROLL's ability to simulate forest structure and composition in the understory" (l. 551-556).

Line 499-504: If a homogeneous seed rain is likely to be the main cause of the model-data mismatch in species composition (and this mismatch is quite large), then it might be helpful to include some sensitivity tests to see the effect of a more heterogeneous seed rain.

> We understand the point of reviewer 2. However, the aim of our evaluation was not to have a maximum fine-tuning of the model to get the most realistic numerical representation of each forest. On the contrary: "To provide a conservative assessment of the model's performance and its transferability to multiple sites, we restricted the number of site-specific calibrated parameters to the ones that are currently poorly informed by available data, or to which the model is known to be sensitive based on sensitivity analyses performed on previous versions of the model (Maréchaux & Chave 2017; Fischer et al. 2019)" (l. 126-129). The uniform seed rain represents a minimal assumption for a conservative assessment of the community assembly mechanisms, independently of any external forcing of species composition through a heterogeneous seed rain. Further research is needed to explore whether and how to fit individualistic species responses

Line 520-522. If this is true, why does the model-data discrepancy only occur at one site? Are there not any Cecropia at Tapajos?

> The discrepancy also occurs at Tapajos, including also *Cecropia*, but the lower genus taxonomic resolution at this site hides this effect in figure 5. Lines 520-522 do not solely refer to Paracou: " The lack of light wood and high leaf area individuals can be related to the underestimated abundances of light demanding and pioneer species with fast growth (Chave et al., 2009), such as the ones of the genus *Cecropia"*.

Line 526. You can check the anomaly in trait/species abundance against seed trait to evaluate this hypothesis, if this is the main reason, there should be a correlation between the model-data mismatch with certain seed traits.

> We lack seed mass or other seed traits in our datasets and could not evaluate this hypothesis, but this is an interesting idea worthy of future explorations.

Line 592-94. With the current analysis, it is hard to tell whether the daily responses are realistic or not. Partial sensitivity comparison is critical

> We answered this comment above with new appendix figures A12 following Bloomfield *et al.,* (2023).

Line 620-636: in addition to the overestimate of GPP per se, the model also consistently overestimated the sensitivity of GPP to environmental factors, particularly in Tapajos. What are the potential causes of this overestimate?

> The discussion has been altered as follows: "TROLL 4.0 tended to overestimate empirical GPP estimates, particularly during the dry season, in comparison to both eddy covariance- and SIF-derived GPP. GPP is driven by the photosynthetic activity of the canopy, which depends on multiple processes (Diao et al., 2023; Slot et al., 2024) and further work would

be needed to discriminate among them, while accounting for eddy covariance uncertainties (Cui and Chui, 2019). Absorbed light typically has an overriding effect on the variation of GPP across seasons in these light-limited rainforests (Yang et al. 2022, Guan et al. 2015), and simulated GPP is sensitive to the parameters that control light transmission and absorbance (light extinction coefficient, apparent quantum yield; Maréchaux & Chave, 2017). Both are assumed fixed and constant in simulations, but are known to vary with leaf angle distribution and leaf optical properties, depending on micro-environmental conditions and species (Long et al., 1993; Poorter et al., 1995; Meir et al., 2000; Kitajima et al., 2005). Additionally, after removing the effect of absorbed light, simulated GPP showed a milder limitation to high values of VPD and temperature in comparison to eddy flux- or SIF-derived estimates. The response of leaf-level gas exchanges to the joint effect of atmospheric dryness and soil water availability shows no clear consensus across models (Powell et al, 2013; Trugman et al., 2018), and could be underestimated during the dry season in TROLL 4.0 simulations as discussed above for transpiration." (l. 696-708).

**Reviewer 3:**

This paper presented a comprehensive model calibration and evaluation of a new model TROLL 4.0. It is a thorough evaluation of model performance and the paper is well written. I do have a few concerns on the paper.

> We thank you for your thorough review of our manuscript and hope that the improvements we have made following your comments and those of the other reviewers will answer your concerns.

The authors used a sequential calibration scheme. Namely, it first calibrates the aboveground biomass and density. The best ensemble member is then used to calibrate the phenology leaf parameters. One concern on this scheme is that the interactions among carbon parameters and phenology leaf parameters could interfere with each other and thus the sequentially calibrated parameters might not give the best results.

> We thank the reviewer 3 for this very good comment. We used a sequential calibration scheme to reduce computing load and based on expert knowledge of developers that the second calibration of litterfall parameters should not interfere with the first of forest structure parameters. However, to assess this assumption, we further explored the sensitivity of forest structure variables from the first calibration to forest litterfall parameters from the second calibration with partial correlation coefficients (PCC) using the function *pcc* of the R package *sensitivity* with a thousand bootstrap draws to assess confidence intervals. As expected and shown below, forest structure variables, namely number of stems, aboveground biomass (AGB), and basal area (BA) showed little to no partial correlations to $a_{T,0}$, $b_{T,0}$, and $delta_0$. PCC showed a null effect for most with the exception of a weak effect of $a_{T,0}$ and $b_{T,0}$ on forest number of stems with a PCC between 0.1 and 0.2. We thus considered our sequential calibration scheme to be valid. These new results have been added in the new appendix figure Fig. A4 reported below, and described in material and methods and results:

- "To assess the model sensitivity to the chosen parameters, we used the calibration parameter spaces and measured response variable sensitivity to each parameter with partial correlation coefficients (PCC). Moreover, we used a sequential calibration scheme to reduce computation load based on the hypothesis that the second calibration of litterfall parameters

does not interfere with the first of forest structure parameters. To assess this assumption, we explored the sensitivity of forest structure variables to forest litterfall parameters." (l. 275-279)

- "Finally, in agreement with results on previous versions of the model, forest structure showed high sensitivity to the explored parameters. Partial correlation coefficients (PCC) were around -0.4 for $a_{CR}$ and around 0.4 for $b_{CR}$ with number of stems, aboveground biomass, and basal area. The background mortality rate $m$ also had a strong effect on aboveground biomass and basal area with a PCC around -0.2 but little to no effect on number of stems (Fig. A4)" (l. 361-364)

- "revealing a strong positive effect of $a_{T,0}$ and $b_{T,0}$ on the peak day of litterfall and negative effect on the ratio of the peak of litterfall, and a weak effect of $delta_0$ on the peak of litterfall. As anticipated, litterfall calibration was independent of the forest structure calibration (Fig. A4)" (l. 433-435)

[Figure]

**Figure A4: Sensitivity of forest structure (left panel) and forest litterfall (right panel) to calibrated parameters. Forest structure and forest litterfall sensitivity to each parameter was assessed with partial correlation coefficients (PCC) using the function *pcc* of the R package *sensitivity* with a thousand bootstrap draws to assess confidence intervals. The intercept and slope of the crown radius allometry $a_{CR}$ and $b_{CR}$ had a strong effect on forest structure, i.e., the number of stems (red), aboveground biomass (AGB, light green), and basal area (BA, green). Basal mortality $m$ also had a strong effect on aboveground biomass (AGB) and basal area (BA) but little to no effect on the number of stems. $a_{T,0}$ and $b_{T,0}$ had a strong positive effect on the peak day of litterfall (blue) and negative effect on the ratio of the peak of litterfall (purple), but $delta_0$ had only a weak effect on the peak of litterfall. The forest structure variables, namely number of stems, aboveground biomass (AGB), and basal area (BA) showed little to no partial correlations to $a_{T,0}$, $b_{T,0}$ and $delta_0$.**

Second, it is not clear to me how the authors capture the uncertainties in data. The authors only used the best fit from ensembles: however, there are always uncertainties in data and there is 'never' a single realization of parameter values that could give the 'best' prediction. It would be better to use an ensemble of parameter values that best fit into the range of plausible range from observations.

> The best simulation was used to reduce the computational load. In fact, the selection of the best set of parameters for forest structure allowed us to use the already run best 600-year simulation for the next steps, without having to run a simulation with the mean of best parameters. We agree that using an ensemble of parameter values would improve uncertainty estimates and remove biases and noise due to variation in the data . However, in our case and as shown in the new appendix table A3, the parameters from the best fit are almost always equal to the median of the parameter values of the five best fits. Thus, our approach remains valid as there is little uncertainty in the parameters.

**Table A3: Comparisons of forest structure and phenology parameter values from the five best fits, including minimum, maximum and median values, as well as the one of the best fit. Note that the median of the parameter values of the five best fits always equal the value of the best fit, except for _m_ at Paracou with a small difference of 0.0025 and $\square_0$ in both sites.**

| Site | Parameter | minimum | median | best | maximum |
|------|-----------|---------|--------|------|---------|
| Paracou | $a_{CR}$ | 1.80 | 1.80 | 1.80 | 1.90 |
| Paracou | $b_{CR}$ | 0.386 | 0.386 | 0.386 | 0.443 |
| Paracou | $m$ | 0.0325 | 0.0325 | 0.0350 | 0.0375 |
| Paracou | $a_{T,0}$ | 0.2 | 0.2 | 0.2 | 0.2 |
| Paracou | $b_{T,0}$ | 0.015 | 0.02 | 0.015 | 0.02 |
| Paracou | $\square_0$ | 0.1 | 0.4 | 0.1 | 0.5 |
| Tapajos | $a_{CR}$ | 2.35 | 2.45 | 2.45 | 2.50 |
| Tapajos | $b_{CR}$ | 0.6994 | 0.7565 | 0.7565 | 0.7850 |
| Tapajos | $m$ | 0.0300 | 0.0400 | 0.0400 | 0.0500 |
| Tapajos | $a_{T,0}$ | 0.2 | 0.2 | 0.2 | 0.3 |
| Tapajos | $b_{T,0}$ | 0.015 | 0.015 | 0.015 | 0.015 |
| Tapajos | $\square_0$ | 0.2 | 0.3 | 0.2 | 0.5 |

Third, it is not clear to me if the authors consider the potential correlation between parameters. The orthogonal design assumes independence among parameters. Based on this assumption, is it possible  that the calibrated parameter sets could be unrealistic?

> With respect to the forest structure parameters, we accounted for the non-independence of the crown allometry parameters $a_{CR}$ and $b_{CR}$ by first exploring their relationship using the TALLO dataset. We then used the residual of the relationship, $epsilon_{bCR}$, for calibration to

constraint their co-variation in agreement with observations. This is explained in the manuscript as follows:

"aCR and bCR are not independent, and we used the TALLO global database of crown radius (CR) and diameter (dbh) measurements (Jucker et al., 2022) to infer their relationship. To do so, we restricted the TALLO database to observations located within 10 km around sites from which we generated a thousand pairs of (aCR, bCR) values. Each pair of values was determined by randomly drawing 10 individuals per 10-cm diameter class to generate a size-balanced dataset to which the following model was fitted: $log(CR) \sim N[aCR +bCR \times log(dbh), \sigma\ 2]$. This resulted in the following linear relationship between the two parameters: $bCR = -0.39 +0.59 \times aCR + \epsilon bCR$ , with $\epsilon bCR$ the error around the relation. This relationship constrained the exploration of the three-dimensional parameter space, so we only had to calibrate $aCR$, $\epsilon bCR$ , and m." (l. 209-216)

A priori, we do not expect crown allometry to be biologically correlated with forest background mortality *m*, and the resulting calibrated values of *m*, as highlighted by the second reviewer, make biological sense. With respect to the litterfall parameters, these are new to this version and the link with existing data is less obvious, hence their calibration. While $a_{T,0}$ controls the decrease of leaf drought tolerance as leaves age, $b_{T,0}$ controls the tree height dependency of leaf sensitivity to water availability, and $delta_0$ controls the pace of leaf shedding in old leaves once they start losing turgor. There is no a priori reason to expect co-variation among those three and we thus chose the most parsimonious calibration design with no constraint on the co-variation. Note however, that, as explained above in answer to reviewer 2, while these three parameters are global and thus independent of species functional strategies, they should induce some dependencies between species drought tolerance, their stature and their phenological strategies in agreement with empirical literature, and this is now better discussed in the companion paper.

Please see below some specific comments:

TROLL 4.0: representing water and carbon fluxes, leaf phenology, and intraspecific trait variation in a mixed-species individual-based forest dynamics model – Part 2: Model evaluation for two Amazonian sites

Line 101: What is the abbreviation for TROLL, which has a meaning of ' a person who intentionally antagonizes others online by posting inflammatory, irrelevant, or offensive comments or other disruptive content'. This does not resonate with me well for a model name.

> TROLL was named after the German botanist Wilhelm Troll. As explained in Chave 1999, which describes the very first version of the model: "Most rainforest trees have a flat crown architecture of the Troll type (Hallé et al., 1978), probably due to their strategy of optimal light interception".

Line 126: $- 0.01 \times \hbar - bT,0$, not sure the meaning of this term. It would be better to explain it without referring to the companion paper.

> In agreement with a comment of reviewer 2, and as suggested here, we have now better explained the different terms of this equation in the evaluation manuscript: "The first term

accounts for a decline of leaf drought tolerance as leaf ages, i.e. a lower ability to maintain turgor as the soil dries for old leaves than mature leaves, with aT,0 controlling the ratio of old to mature leaf turgor loss point. The second term accounts for the tree height dependency of this susceptibility to decrease in water availability: it induces a susceptibility to a (small) decrease $b$T,0 in soil water availability for large trees, while preventing them from constantly shedding their old leaves at a fast pace. Finally, delta$_0$ controls the pace of old leaf shedding as they start losing turgor by modulating the old leaf turnover rate in absence of water depletion. Overall, the parameters $aT,o$, $bT,o$ and $\delta o$ control the intensity and the timing of the peak of litterfall under drying soil conditions." (l. 141-147)

Line 141: why species should have a maximum diameter at breast height and set it as 95th quantile of species diameter from observed data?  It is contradictory based on the definition and use of the 95th quantile of species diameter. My understanding of the system is that there could be a maximum height, but trees could continue to get 'fatter'.

> We fully agree, and our use of this stature trait is actually in agreement with this comment. However, this comment made us realize that there was a mismatch in variable name and abbreviation among the two companion manuscripts. What was named dbh$_{max}$ in the previous version of the evaluation manuscript (Part2) is actually named dbh$_{thres}$ in the companion manuscript (Part 1; see section 2.4.1 and 2.6.4 therein). Equ. 62 of the companion manuscript actually allows a tree to reach a higher diameter at breast height than dbh$_{thres}$, from which the carbon allocation to tree size increment only starts decreasing. Whether many trees will actually reach bigger size is mediated by the different sources of mortality (among which size-mediated treefall and the background mortality rate). We have now renamed dbh$_{max}$ dbh$_{thres}$, in agreement with the companion manuscript.

Line 150: can you give some scientific background on the equation without going to the modeling companion paper?

> As now better described in the manuscript: "We used a Michaelis-Menten model form for tree height $h$, which grows with diameter $dbh$ towards a plateau value $h_{lim}$ at a rate $a_h$ (Molto et al., 2014)." (l. 171-172).

Line 174: For Tapajos, rainfall data from FLUXNET 2015 is not reliable due to issues with rain gauges and the authors used rainfall data from the ERA5-Land reanalysis dataset. I understand the strategy here but it will be nice to know how much the coarse resolution ERA5 data create a difference compared to FLUXNET 2015 data based on site Paracou, as it might be useful to see how much difference between calibrated parameters across these two sites are from climate driver data source differences.

> As explained to reviewer 2, we here used ERA5-Land on the basis of a previous good evaluation of ERA5-Land against data from the local tower in Paracou (see Schmitt et al. 2023). However, following the comments of reviewers 2 and 3, we further compared both CHIRPS and ERA5-Land against local eddy-flux tower measurements in Paracou extracted from FLUXNET 2015, at daily, 5-day, 15-day and monthly resolutions. CHIRPS and ERA5-Land had similar agreement to locally measured precipitations (figure below), with even higher correlations for ERA5-Land than CHIRPS. However, they both overestimated the precipitation of low precipitation events and underestimated the precipitation of high precipitation events, resulting in a low agreement for daily variations ($R^2$ of 0.10 and 0.22 for

CHIRPS and ERA5-Land respectively), which quickly increases for 5-day ($R^2$ of 0.39 and 0.53) and 15-day variations ($R^2$ of 0.63 and 0.73). Although a similar assessment was not possible in Tapajos due to a lack of local reliable rainfall data, we therefore decided to keep ERA5-Land for filling the precipitation data gaps in Tapajos. These new results have been added in the new appendix figure Fig. A2 reported below, and described in material and methods: "ERA5-Land showed a better agreement with on-site precipitation data from FLUXNET 2015 at Paracou when compared to other products, like CHIRPS (Funk et al. 2015; Fig. A2)" (l. 196-198).

[Figure]

**Figure A2: Comparisons of CHIRPS (red) and ERA5-Land (blue) precipitation products against local eddy-flux tower measurements retrieved from FLUXNET 2015 in Paracou at daily, 5-day, 15-day, and monthly resolutions. CHIRPS and ERA5-Land had similar agreement to locally measured precipitations, with even higher correlations for ERA5-Land than CHIRPS. However, they both overestimated low precipitation events and underestimated high precipitation events, resulting in low agreement for daily variations ($R^2$ of 0.10 and 0.22), which quickly increases for 5-day ($R^2$ of 0.39 and 0.53) and 15-day variations ($R^2$ of 0.63 and 0.73). Although a similar assessment was not possible in Tapajos due to a lack of local reliable rainfall data, we therefore decided to keep ERA5-Land for filling the precipitation data gaps in Tapajos.**

Line 198: Might be good to point out the time to reach equilibrium in the model and how the climate driver is cycled.

> The time to reach equilibrium will depend on the variables, with forest fluxes quickly reaching equilibrium but forest structure and composition taking more time to stabilize. We usually refer to aboveground biomass (AGB) for equilibrium definition, and as shown in the

new appendix figure A3 (shown below), AGB reaches equilibrium around 400 years in TROLL spin-up simulations starting from bareground for both sites, but we used a total time of 600 years for safety. This is further referenced in material and methods "After 600 years of simulated forest dynamics the system reached a mature forest stage with stable forest structure (Fig. A3)" (l. 245). The climate driver is cycled yearly to keep within-year seasonal patterns but randomly drawing each year across the ten years available in data to avoid creating cycles as described line 207-208 in the first version of the manuscript: "Each simulation was forced each year by randomly drawing a year among the ten years of climatic data. In doing so, we avoided applying a periodic climatic forcing or any potential trend linked to global warming".

[Figure]

**Figure A3: 600-year spin-up simulations from bareground with calibrated parameters showing equilibrium reached by number of stems (bottom) aboveground biomass (AGB, top) at Paracou (red) and Tapajos (blue).**

**References**

Bloomfield, K. J., Stocker, B. D., Keenan, T. F., & Prentice, I. C. (2023). Environmental controls on the light use efficiency of terrestrial gross primary production. Global change biology, 29(4), 1037-1053.

Chave, J. (1999). Study of structural, successional and spatial patterns in tropical rain forests using TROLL, a spatially explicit forest model. Ecological modelling, 124(2-3), 233-254.

Duursma, R. A., Blackman, C. J., Lopéz, R., Martin‑StPaul, N. K., Cochard, H., & Medlyn, B. E. (2019). On the minimum leaf conductance: its role in models of plant water use, and ecological and environmental controls. *New Phytologist*, *221*(2), 693-705.

Fischer, F. J., Maréchaux, I., & Chave, J. (2019). Improving plant allometry by fusing forest models and remote sensing. *New Phytologist*, *223*(3), 1159-1165.

Funk, C., Peterson, P., Landsfeld, M., Pedreros, D., Verdin, J., Shukla, S., ... & Michaelsen, J. (2015). The climate hazards infrared precipitation with stations—a new environmental record for monitoring extremes. Scientific data, 2(1), 1-21.

Hallé, F., Oldeman, R. A., & Tomlinson, P. B. (2012). Tropical trees and forests: an architectural analysis. Springer Science & Business Media.

Hérault, B., Bachelot, B., Poorter, L., Rossi, V., Bongers, F., Chave, J., ... & Baraloto, C. (2011). Functional traits shape ontogenetic growth trajectories of rain forest tree species. Journal of ecology, 99(6), 1431-1440.

Jucker, T., Fischer, F. J., Chave, J., Coomes, D. A., Caspersen, J., Ali, A., ... & Zavala, M. A. (2022). Tallo: A global tree allometry and crown architecture database. *Global change biology*, *28*(17), 5254-5

Jucker, T., Fischer, F., Chave, J., Coomes, D., Caspersen, J., Ali, A., ... & Zimmermann, N. (2024). The global spectrum of tree crown architecture. *bioRxiv*, 2024-09.

Kunert, N., Aparecido, L. M. T., Wolff, S., Higuchi, N., dos Santos, J., de Araujo, A. C., & Trumbore, S. (2017). A revised hydrological model for the Central Amazon: The importance of emergent canopy trees in the forest water budget. Agricultural and Forest Meteorology, 239, 47-57.

Levis, C., Costa, F. R., Bongers, F., Peña-Claros, M., Clement, C. R., Junqueira, A. B., ... & Sandoval, E. V. (2017). Persistent effects of pre-Columbian plant domestication on Amazonian forest composition. Science, 355(6328), 925-931.

Li, L., Fang, Y., Zheng, Z., Shi, M., Longo, M., Koven, C. D., ... & Leung, L. R. (2023). A machine learning approach targeting parameter estimation for plant functional type coexistence modeling using ELM-FATES (v2. 0). Geoscientific Model Development, 16(14), 4017-4040.

Longo, M., Knox, R. G., Levine, N. M., Alves, L. F., Bonal, D., Camargo, P. B., ... & Moorcroft, P. R. (2018). Ecosystem heterogeneity and diversity mitigate Amazon forest resilience to frequent extreme droughts. New Phytologist, 219(3), 914-931.

Maréchaux, I., & Chave, J. (2017). An individual-based forest model to jointly simulate carbon and tree diversity in Amazonia: description and applications. *Ecological Monographs*, *87*(4), 632-664.

Molto, Q., Hérault, B., Boreux, J. J., Daullet, M., Rousteau, A., & Rossi, V. (2014). Predicting tree heights for biomass estimates in tropical forests–a test from French Guiana. Biogeosciences, 11(12), 3121-3130.

Powell, T. L., Galbraith, D. R., Christoffersen, B. O., Harper, A., Imbuzeiro, H. M., Rowland, L., ... & Moorcroft, P. R. (2013). Confronting model predictions of carbon fluxes with measurements of Amazon forests subjected to experimental drought. New Phytologist, 200(2), 350-365.

Salzet, G. (2024). *Etude de la durabilité de la filière forêt-bois en Guyane française : approche spatialisée par modélisation bioéconomique* (Doctoral dissertation, Université de Guyane).

Saltelli, A. (2019). A short comment on statistical versus mathematical modelling. Nature communications, 10(1), 3870.

Schmitt, S., Salzet, G., Fischer, F. J., Maréchaux, I., & Chave, J. (2023). rcontroll: An R interface for the individual‑based forest dynamics simulator TROLL. Methods in Ecology and Evolution, 14(11), 2749-2757.

Schmitt, S., Hérault, B., & Derroire, G. (2023). High intraspecific growth variability despite strong evolutionary legacy in an Amazonian forest. Ecology Letters, 26(12), 2135-2146.

Smith, M. N., Stark, S. C., Taylor, T. C., Ferreira, M. L., de Oliveira, E., Restrepo‑Coupe, N., ... & Saleska, S. R. (2019). Seasonal and drought‑related changes in leaf area profiles depend on height and light environment in an Amazon forest. New Phytologist, 222(3), 1284-1297.

Yao, Y. T., Joetzjer, E., Ciais, P., Viovy, N., Cresto Aleina, F., Chave, J., ... & Luyssaert, S. (2022). Forest fluxes and mortality response to drought: model description (ORCHIDEE-CANNHA r7236) and evaluation at the Caxiuanã drought experiment. Geosci Model Dev. 15 (20): 7809–7833.